# Systematic mapping of mitochondrial calcium uniporter channel (MCUC)-mediated calcium signaling networks

Hilda Delgado de la Herran [1,16], Denis Vecellio Reane [1,16], Yiming Cheng [1], Máté Katona [2], Fabian Hosp [3,15], Elisa Greotti [4,5,6], Jennifer Wettmarshausen[1], Maria Patron [7,8], Hermine Mohr [9], Natalia Prudente de Mello [1], Margarita Chudenkova[1], Matteo Gorza[1], Safal Walia[1], Michael Sheng-Fu Feng [1], Anja Leimpek [1], Dirk Mielenz [10], Natalia S Pellegata[9,11], Thomas Langer[7,8], György Hajnóczky[2], Matthias Mann[3,12], Marta Murgia [3,5✉] & Fabiana Perocchi [1,13,14✉]

## Abstract

The mitochondrial calcium uniporter channel (MCUC) mediates mitochondrial calcium entry, regulating energy metabolism and cell death. Although several MCUC components have been identified, the molecular basis of mitochondrial calcium signaling networks and their remodeling upon changes in uniporter activity have not been assessed. Here, we map the MCUC interactome under resting conditions and upon chronic loss or gain of mitochondrial calcium uptake. We identify 89 high-confidence interactors that link MCUC to several mitochondrial complexes and pathways, half of which are associated with human disease. As a proof-of-concept, we validate the mitochondrial intermembrane space protein EFHD1 as a binding partner of the MCUC subunits MCU, EMRE, and MCUB. We further show a MICU1-dependent inhibitory effect of EFHD1 on calcium uptake. Next, we systematically survey compensatory mechanisms and functional consequences of mitochondrial calcium dyshomeostasis by analyzing the MCU interactome upon EMRE, MCUB, MICU1, or MICU2 knockdown. While silencing EMRE reduces MCU interconnectivity, MCUB loss-of-function leads to a wider interaction network. Our study provides a comprehensive and high-confidence resource to gain insights into players and mechanisms regulating mitochondrial calcium signaling and their relevance in human diseases.

**Keywords** Calcium Signaling; Mitochondria; Mitochondrial Calcium Uniporter; Organelle; Proteomics
**Subject Categories** Organelles; Proteomics

## Introduction

For decades, mitochondria have been recognized as key players in $Ca^{2+}$-mediated signal transduction cascades, decoding the spatio-temporal dynamics of intracellular $Ca^{2+}$ signals (Hajnóczky et al, 1995; Spät et al, 2008; Kaftan et al, 2000). This property enables the organelle to regulate the metabolic state of the cell, its growth, fate, and overall survival (Rizzuto et al, 2012). Indeed, changes in cytosolic $Ca^{2+}$ concentration ($[Ca^{2+}]_{cyt}$) are promptly transferred into the mitochondrial matrix through an electrogenic pathway powered primarily by the mitochondrial membrane potential and mediated by MCUC, a highly selective calcium channel located at the inner mitochondrial membrane (IMM) (Kirichok et al, 2004; DeLuca and Engstrom, 1961; Vasington and Murphy, 1962). A transient elevation of matrix $Ca^{2+}$ concentration ($[Ca^{2+}]_{mt}$) is then efficiently coupled to the regulation of mitochondrial bioenergetics to match the metabolic demands of cells and tissues (Tsai et al, 2022; Fecher et al, 2019; Fieni et al, 2012; Paillard et al, 2017; Hajnóczky et al, 1995). However, when sustained, mitochondrial $Ca^{2+}$ (mt-$Ca^{2+}$) uptake can trigger oxidative stress and mitochondrial swelling, with consequent activation of cell death pathways (Rizzuto et al, 2012). Accordingly, dyshomeostasis of mt-$Ca^{2+}$ has already been implicated in numerous diseases including diabetes, neurodegeneration, stroke, heart failure, inflammation, muscular

[1]Institute for Diabetes and Obesity, Helmholtz Diabetes Center, Helmholtz Zentrum Munich, Munich, Germany. [2]Department of Pathology, Anatomy, and Cell Biology, MitoCare Center, Thomas Jefferson University, Philadelphia, PA, USA. [3]Department of Proteomics and Signal Transduction, Max Planck Institute of Biochemistry, Martinsried, Germany. [4]Neuroscience Institute, National Research Council of Italy, Padua, Italy. [5]Department of Biomedical Sciences, University of Padova, Padua, Italy. [6]Padova Neuroscience Center, University of Padova, Padua, Italy. [7]Institute for Genetics, Cologne Excellence Cluster on Cellular Stress Responses in Aging-Associated Diseases, Center for Molecular Medicine, University of Cologne, Cologne, Germany. [8]Max Planck Institute for Biology of Aging, Cologne, Germany. [9]Institute of Diabetes and Cancer, Helmholtz Center Munich, Munich, Germany. [10]Division of Molecular Immunology, University of Erlangen, Nikolaus-Fiebiger-Zentrum, FAU Erlangen-Nürnberg, Erlangen, Germany. [11]Department of Biology and Biotechnology, University of Pavia, Pavia, Italy. [12]Faculty of Health Sciences, Novo Nordisk Foundation Center for Protein Research, University of Copenhagen, Copenhagen, Denmark. [13]Institute of Neuronal Cell Biology, Technical University of Munich, Munich, Germany. [14]Munich Cluster for Systems Neurology, Munich, Germany. [15]Present address: Roche Pharma Research and Early Development, Large Molecule Research, Mass Spectrometry, Penzberg, Germany. [16]These authors contributed equally: Hilda Delgado de la Herran, Denis Vecellio Reane. ✉E-mail: mmurgia@mpi.de; fabiana.perocchi@tum.de

atrophy, and cancer (Garbincius and Elrod, 2022). Therefore, elucidating the molecular determinants of channel activity and MCUC-dependent control of mitochondrial functions represents an important milestone in cell physiology and pathophysiology.

Unbiased computational and experimental analyses that leveraged coevolution, co-expression, and organellar proteomics led to the discovery of MICU1 (Mitochondrial Calcium Uptake 1) (Perocchi et al, 2010) and MCU (Mitochondrial Calcium Uniporter) (De Stefani et al, 2011; Baughman et al, 2011), as Ca$^{2+}$-dependent regulator and pore-forming subunit of the uniporter, respectively. These findings paved the way for the characterization of additional binding partners, including the MCU-dominant negative beta subunit (MCUB) (Raffaello et al, 2013), a family of MICU1 paralogs and splice variants (MICUs) (Plovanich et al, 2013; Patron et al, 2019; Vecellio Reane et al, 2016; Patron et al, 2014), and EMRE (Essential MCU Regulator) (Sancak et al, 2013). However, although several channel-forming elements are currently known, MCUC-dependent Ca$^{2+}$ signaling networks in mitochondria have not yet been assessed. A few studies have applied affinity purification (AP) coupled with quantitative mass spectrometry (MS) for an unbiased mapping of MCU protein–protein interactions (PPIs) (Sancak et al, 2013; Austin et al, 2022; Antonicka et al, 2020) but failed to identify functional associations with other mitochondrial proteins besides the known MCUC subunits. Instead, recent evidence indicates that MCUC is part of a broader functional network and can dynamically interact with other mitochondrial complexes, for example with members of the respiratory chain complex I (RCCI) (Balderas et al, 2022), the ATP-dependent proteolytic complex (König et al, 2016; Tsai et al, 2017), and the mitochondrial contact site and cristae organization system (MICOS) (Tomar et al, 2023; Gottschalk et al, 2019). In this way, the organelle can regulate the stability, assembly and activity of the uniporter, which in turn influence mitochondrial membrane structure and energy synthesis, and compensate for mitochondrial dysfunctions. Nevertheless, a systematic and comprehensive analysis of the MCUC protein interaction landscape and signal transduction networks, both under resting conditions and perturbed mt-Ca$^{2+}$ homeostasis, is still lacking.

Here, we devised a biochemical strategy to characterize the MCUC interactome in human cells with high confidence and resolution under resting conditions and following genetic perturbations. Tandem affinity purifications (TAPs) coupled with quantitative and integrative liquid chromatography-tandem mass spectrometry (LC-MS/MS) analyses led us to map 139 statistically significant PPIs between 95 mitochondrial proteins in HEK293, including all currently known members of the uniporter complex. We were able to capture all previously described interactions between MCUC, RCCI, MICOS, and several mitochondrial proteases, as well as novel molecular links between mt-Ca$^{2+}$ signaling, organelle dynamics, biogenesis, and apoptosis. Among the MCUC binding partners, we also identified a handful of hitherto poorly characterized proteins, for example the EF-hand domain-containing protein D1 (EFHD1, also known as Swiprosin-2 or Mitocalcin) (Hou et al, 2016; Dütting et al, 2011). We corroborate a role for EFHD1 as an inhibitor of MCU-mediated mt-Ca$^{2+}$ uptake, a function that appears to be dependent on the presence of MICU1. Next, to investigate the molecular basis of compensatory mechanisms and functional consequences of MCUC remodeling upon loss-of-function (LOF) of its components, we

performed LC-MS/MS analyses of TAPs from MCU-tagged HEK293 cells upon silencing of EMRE, MCUB, MICU1 or MICU2. Upon EMRE knockdown (KD) the number of PPIs was dramatically reduced, possibly due to the inability of MCU to form active channels. In contrast, MCUB KD, which increases mt-Ca$^{2+}$ uptake, led to an expansion and greater interconnection of the MCU protein network and resulted in MCU forming high molecular weight (MW) macromolecular complexes, in both human cells and mouse tissues. Altogether, our dataset represents a rich, high-confidence resource that can be explored to discover novel molecular links between MCUC-mediated Ca$^{2+}$ entry and organelle biology, as well as providing new insights into unexplored mt-Ca$^{2+}$ signaling and disease associations.

# Results

## Systematic and unbiased identification of MCUC protein–protein interactions

We devised a biochemical workflow to achieve an unbiased and comprehensive characterization of MCUC-specific PPIs in human cells under resting conditions (Fig. 1). To this goal, MCU, MCUB, or EMRE proteins were fused at the C-terminus to a StrepII-HA-His$_6$ tag and stably integrated into Flp-In T-REx HEK293 cells (parental line) under the control of a tetracycline-inducible promoter. This allowed the targeted integration of each bait protein at a single transcriptionally active genomic locus that is the same in every cell line and ensures a near-physiological expression level of the bait proteins. The latter is essential to minimize false positive associations and accurately define the endogenous protein composition and interaction network of MCUC. By encompassing all three membrane-spanning subunits of MCUC as baits, we aimed to increase the coverage of copurifying proteins across all mitochondrial sub-compartments. As shown in Figs. 2A and EV1, we confirmed that tetracycline-dependent induction of baits did not affect the endogenous level of other MCUC subunits, except for EMRE. Upon expression of tagged MCU and EMRE, the level of endogenous EMRE was increased and decreased, respectively, which is consistent with previous findings indicating a tight regulation of its stability by MCU (Sancak et al, 2013; Tsai et al, 2017; König et al, 2016).

We employed a TAP strategy based on Strep and polyhistidine tags to enrich for interacting protein partners of MCUC. Immunoblot analyses validated that each bait could be efficiently purified from whole cell lysates (Fig. 2B). For a systematic and unbiased identification and quantification of proteins that co-precipitated with each bait, we performed LC-MS/MS analysis with at least four biological replicates. Importantly, we included TAPs from the parental line, hereby used as negative control (Ctrl-TAP) to discriminate specific interactions from spurious ones. As a result, we found that the relative abundance of MCU, MCUB, and EMRE was consistently and significantly higher in TAPs from bait-expressing cells than in control samples, when compared to the median intensity of all quantified proteins (Fig. 2C). Given our interest in characterizing the interaction partners of MCUC within mitochondria, we focused our analyses on mitochondrial preys. To define significant and specific MCUC interactors we filtered our dataset for mitochondrial proteins and ran a quantitative bait-prey

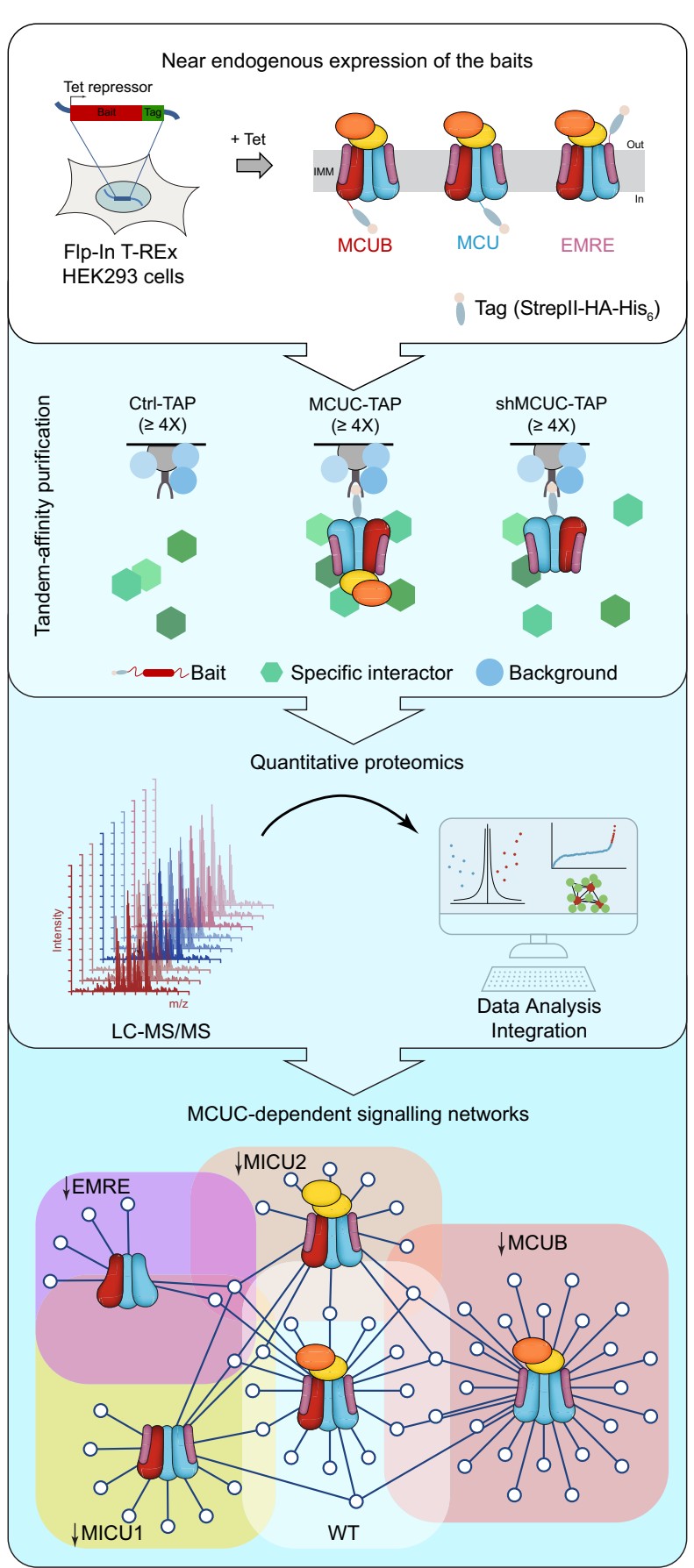

◄ **Figure 1. Proteomics approach to characterize the MCUC interactome.**

Tandem affinity purifications (TAPs) from Flp-In T-REx HEK293 cells that are wild-type (Ctrl) or expressing MCU, MCUB, and EMRE as baits. Tet tetracyclin, IMM inner mitochondrial membrane, StrepII streptavidin II tag, His$_6$ polyhistidine tag, HA hemagglutinin tag, LC-MS/MS liquid chromatography-tandem mass spectrometry.

co-enrichment analysis based on a two-tailed Welch's $t$ test, a within-group variance (s0) of 1, and a permutation-based false discovery rate (FDR) of either 0.05 ("high-confidence") or 0.10 ("medium-confidence") (Fig. 2D–F). In addition, Hein et al have previously demonstrated that the intensity profiles of interacting proteins are correlated (Hein et al, 2015). Therefore, as additional classifiers to identify MCUC interactors, we performed "local" and "global" correlation analyses, comparing the similarity of protein intensity profiles across either pairs of bait and control TAPs, or across all measured samples (Ctrl-TAPs, MCU-TAPs, MCUB-TAPs, and EMRE-TAPs), respectively.

With the term "interactor" or "prey" we refer to any protein that stably or transiently binds MCUC directly or indirectly. Indirect binding can occur when baits and preys are in close proximity without engaging in direct physical interactions. These associations might be facilitated by subunits of a protein complex or members of a pathway that directly interact with MCUC. Examples include connections between mitochondrial protease complexes and MCUC (König et al, 2016; Tsai et al, 2017, 2022) or between MCU and RCCI (Balderas et al, 2022). The joint analysis of all datasets yielded a total of 139 interactions among 95 mitochondrial proteins and successfully recovered all currently known subunits of the MCUC, namely MCU, EMRE, MCUB, MICU1, MICU2, and MICU3 (Fig. 2G; Dataset EV1). Among the MCUC interactors, 33 were found to be shared between at least two baits, and 11 were common to all three conditions. Interestingly, more than 50% of all preys were bait-specific, suggesting their involvement in the selective maturation and regulation of MCU, EMRE, or MCUB.

## The MCUC protein network identifies molecular links between Ca²⁺ and mitochondrial functions

To globally analyze MCUC-specific PPIs, we assessed the biochemical, functional, and evolutionary properties of all identified protein partners (Fig. 3; Dataset EV1–3). Roughly 50% of our interactome consisted of proteins with at least one predicted transmembrane domain, for example subunits of the outer (TOM) and inner (TIM) membrane protein translocases and the oxidative phosphorylation (OXPHOS) machinery. The other half were soluble proteins including components of the MICUs family, the tricarboxylic acid cycle (TCA cycle), and the mitochondrial DNA (mt-DNA) maintenance and expression system (Fig. 3A). Accordingly, the uniporter engaged in interactions with proteins localized in all four submitochondrial compartments (Fig. 3B). MCU and EMRE, which represent the minimal functional unit of MCUC (Kovács-Bogdán et al, 2014; Pittis et al, 2020), sampled largely similar environments although exposing their tags on opposite sides of the IMM. Interestingly, they also mediated functional associations with proteins located at the outer mitochondrial membrane (OMM) and involved in mitochondrial morphology, apoptosis, and dynamics, such as FIS1, BNIP3L, and PARK7. Besides subunits of MCUC, the identified interactors encompassed

a wide spectrum of mitochondrial functions, with several not yet known to be linked to Ca²⁺ signaling (Fig. 3C).

We then mapped the MCUC interactome as a protein network, whereby each node corresponds to a prey connected to a given bait through either evidence of co-enrichment or profile correlations (Fig. 3D). Reassuringly, each bait significantly interacted with itself and with all known MCUC members. To further corroborate the high quality and coverage of our dataset, we first searched the literature for experimental evidence of physical and functional associations between the uniporter and mitochondrial proteins in our network. Among shared interactions, we identified GHITM, also known as TMBIM5 or MICS1. This protein was recently characterized as a Ca²⁺/H⁺ exchanger of the IMM (Zhang et al, 2022), able to shape mt-Ca²⁺ cycling by inhibiting the activity of the m-AAA protease AFG3L2 (Austin et al, 2022; Patron et al, 2022). We also identified AFG3L2 and the YME1L proteolytic hub (YME1L1, PARL, and STOML2) (Wai et al, 2016), all previously implicated in MCUC processing, assembling and degradation (König et al, 2016; Tsai et al, 2017, 2022). To test the usefulness of the MCUC protein network as a resource for the identification of novel research directions and regulatory mechanisms, we also mined our dataset for associations between the uniporter and mitochondrial functions that are known to be controlled by Ca²⁺ but for which the molecular players remain poorly characterized. For instance, the MCUC interactome included several proteins (FIS1, KIF1B, GJA1, and MTUS1) involved in mitochondrial ultrastructural organization, shape and dynamics. This supports the notion that mt-Ca²⁺ signaling can affect the organelle's morphology through the regulation of mitochondrial fusion and fission (Zhao et al, 2015). In addition, our results raise the hypothesis that members of the MCUC could participate in the regulation of contact sites between inner and outer mitochondrial membranes through interactions with members of the MICOS complex such as MICOS10 (Rampelt et al, 2022). Notably, MICU1 was recently found to bind the MICOS complex and regulate mitochondrial membrane dynamics independently of matrix Ca²⁺ uptake, while MCU was shown to relocalize to the cristae junctions upon an increase of [Ca²⁺] in the IMS (Tomar et al, 2023; Gottschalk et al, 2019). Since mt-Ca²⁺ signaling is a central regulator of oxidative metabolism and ATP production (Hajnóczky et al, 1995; Jouaville et al, 1999), we also looked for molecular players coupling MCUC with mitochondrial bioenergetics. Remarkably, we found that the MCUC network was especially enriched in PPIs with components of RCCI (e.g., NDUFA3, NDUFB10, NDUFS8), RCCIV (e.g., COX6C, COX7C, COX18) and the TCA cycle (PDHB and PDHX). Supporting our findings, the RCCI assembly subunit NDUFA3 was recently shown to bind MCU, an interaction between complexes that regulates uniporter level and activity to maintain bioenergetic homeostasis (Balderas et al, 2022). Furthermore, our biochemical approach detected interactions between MCU, the J subunit of the ATP synthase complex (ATP5MPL), and the ATP synthase assembly factor, TMEM70. The latter was recently described to interact with the ATP synthase subunit c (Kovalčíková et al, 2019;

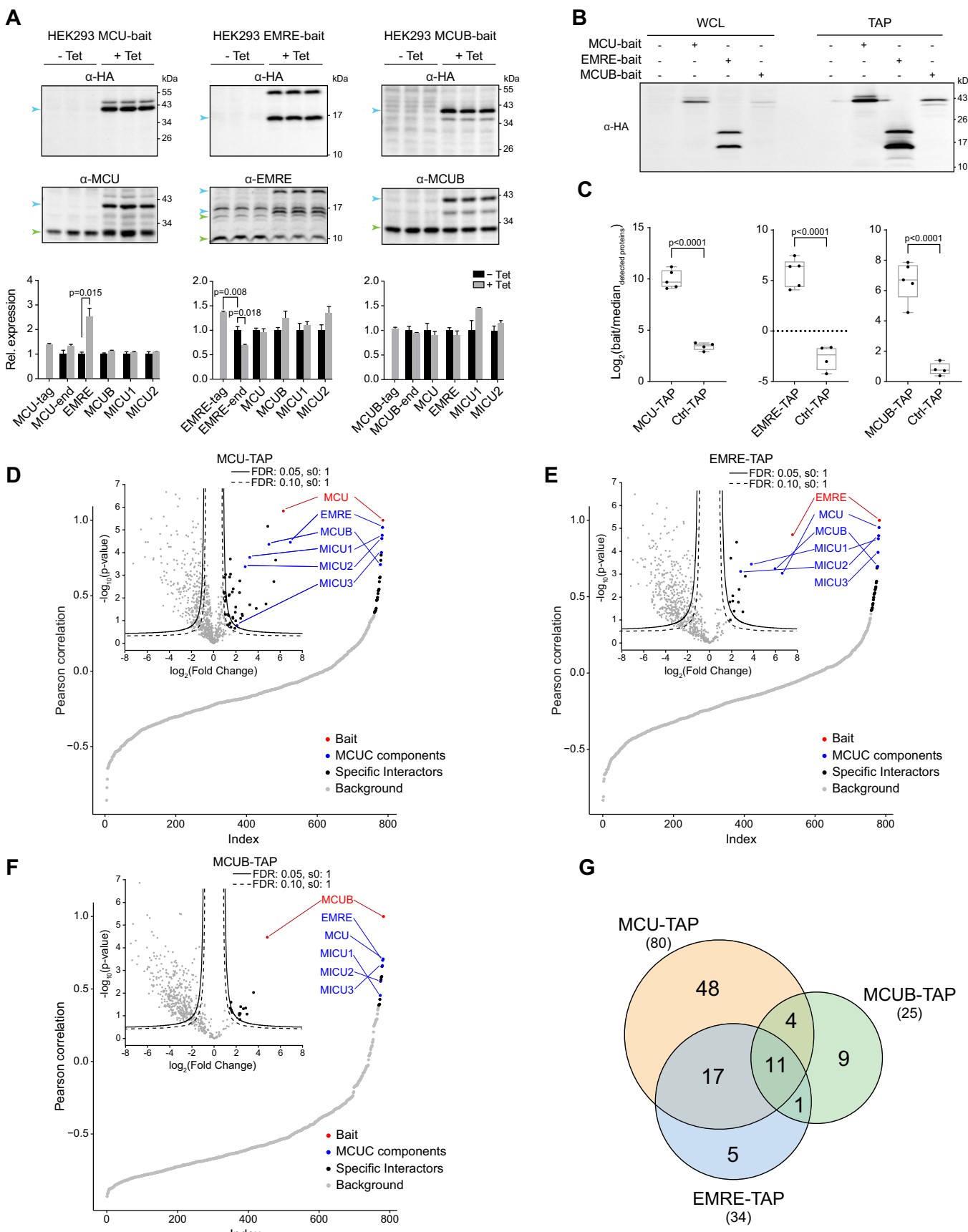

**Figure 2. Unbiased and systematic identification of endogenous MCU, MCUB, and EMRE protein interactors.**

(A) Protein expression of known MCUC subunits upon bait induction. Upper: Immunoblot analysis of whole cell lysates before (-Tet) and after ( + Tet) tetracycline-driven expression of MCU, EMRE, and MCUB. Green and blue arrows refer to endogenous and exogenous protein levels, respectively. Lower: Quantification of MCUC protein expression (-tag and -end refer to exogenous and endogenous protein level, respectively) normalized to ACTIN (loading control) and relative to -Tet condition (mean ± SEM; $n = 3$ independent experiments); Student's $t$ test. (B) Bait enrichment after TAP from whole cell lysates (WCL) of HEK293 cells after tetracycline-driven expression of the bait. (C) Intensity-based absolute quantification of baits over the median intensity of all detected proteins in biological replicates of TAPs (mean ± SEM; $n \geq 4$ independent experiments); Student's $t$ test. The line in the middle of the box is plotted at the median, the boxes extend from the 25th to the 75th percentile, and the whiskers extend to the minimum and maximum values. (D–F) Global Pearson's correlation rank and Hawaii plots of mitochondrial proteins enriched in MCU-bait (D), EMRE-bait (E), and MCUB-bait (F) TAPs ($P$ value, two-tailed Welch's $t$ test). Continuous and dashed lines indicate specific interaction partners defined by permutation-based FDR thresholds of 0.05 or 0.10, respectively. (G) Overlap of MCUC interactors from all three baits. See also Fig. EV1 and Dataset EV1. Source data are available online for this figure.

Bahri et al, 2021), which was previously shown to bind MCU in trypanosomes and human cells (Huang and Docampo, 2020). To the same goal, during $Ca^{2+}$ signaling the activation of SLC25A25 allows matching cellular energy demand and supply by regulating the matrix adenyl nucleotide pool (del Arco et al, 2016). However, no previous evidence of an interaction between the uniporter and SLC25A25 has been reported. Instead, we found SLC25A25 to be significantly co-enriched with the MCU and EMRE baits possibly allowing them to co-localize in the same $Ca^{2+}$ microdomain and co-activate in response to rises in $Ca^{2+}$ concentration ($[Ca^{2+}]$).

Next, we surveyed properties of mt-$Ca^{2+}$ signaling by analyzing the expression and evolutionary profiles of all MCUC interactors, as well as their involvement in human diseases (Fig. 3E–H; Datasets EV2 and EV3). First, we quantified the relative protein level of each MCUC binding partner across healthy human tissues using the GTEx (Genotype-Tissue Expression) database (Jiang et al, 2020). Interestingly, we observed a highly heterogeneous tissue distribution of MCUC interactors, which is consistent with the wide range of biological processes linked to mt-$Ca^{2+}$ in our network (Fig. 3E). These proteins encompass a wide range of functions, from fundamental mitochondrial processes like OXPHOS and mt-DNA replication, to more specialized roles, such as regulation of organelle shape and dynamics, as well as activation of cell death pathways. As expected, based on their role as the minimal unit of the uniporter, MCU and MICU1 showed a ubiquitous expression compared to tissue-specific regulators such as MICU2, MICU3 and MCUB. EMRE was often undetected due to its small size and highly hydrophobic nature, which makes it difficult to detect by MS analyses of whole proteomes. Next, we assessed the evolutionary conservation of each interactor across eukaryotes by mapping orthology relationships across 120 eukaryotic species inferred from Protphylo (Cheng and Perocchi, 2015). Comparative genomics analyses have been instrumental in identifying MCUC components (Perocchi et al, 2010; Baughman et al, 2011; De Stefani et al, 2011), which are present in all major eukaryotic groups, but mostly absent in protists and fungi (Pittis et al, 2020). However, while coevolution can be used to predict functional associations, functionally related proteins do not necessarily co-evolve, calling for complementary approaches, like ours, to reach a comprehensive map of mt-$Ca^{2+}$ signaling networks. Namely, most of the proteins in our network did not exhibit similar phylogenetic profiles to MCU (Fig. 3F). Overall, they showed a patchy evolutionary conservation, with mitochondrial functional modules of ancient origin being highly conserved, while others present only in higher vertebrates. The former group could represent processes that were placed under the control of mt-$Ca^{2+}$ early in evolution; the latter would possibly

spotlight species-specific functions linked to $Ca^{2+}$ signaling after the acquisition of MCU. Finally, we found that half of the MCUC interactors were linked to over 300 different disease phenotypes, spanning from neurological and metabolic disorders to cancer (Fig. 3G), including disorders related to mt-DNA homeostasis, whose pathophysiology has not yet been connected to dysfunctions in mt-$Ca^{2+}$ signaling (Fig. 3H). Altogether, our analysis provides a high-confidence and comprehensive dataset of endogenous MCU, MCUB, and EMRE protein binding partners.

## Validation of newly identified MCUC interactors

To corroborate the quality of our resource, we experimentally validated a subset of specific interactions with different degrees of co-enrichment and correlation to MCU, EMRE, and MCUB baits. MICU3, which was not previously identified as a bona fide binding partner of MCUC in HEK293 cells (Sancak et al, 2013; Antonicka et al, 2020; Austin et al, 2022), resulted as a high-confidence interactor of all three baits in our dataset. Indeed, quantitative proteomic analyses of mitochondria have confirmed that MICU3 is expressed in HEK293 cells (Morgenstern et al, 2021), albeit in low amount when compared to other MCUC components (Fig. 4A). MICU3 is a mitochondrial protein located in the IMS and loosely attached to the IMM, which can form cysteine-mediated disulfide bonds with MICU1 and acts as an enhancer of MCUC thanks to its ability to sense $[Ca^{2+}]$ (Fig. EV2A–I) (Patron et al, 2019). To assess whether MICU3 also played a functional role in the regulation of mt-$Ca^{2+}$ uptake in HEK293 cells, we stably expressed either control (pLKO) or shRNAs targeting different regions of MICU3 mRNA (Fig. 4B). Notably, mitochondria of digitonin-permeabilized sh-MICU3 cells showed reduced $Ca^{2+}$ uptake capacity compared to pLKO upon consecutive addition of exogenous $Ca^{2+}$ (Fig. 4C). The $Ca^{2+}$ clearance phenotype was strongly correlated to MICU3 protein level and therefore not likely due to an off-target effect. Next, we investigated the link between MCU, Tafazzin (TAZ), and the lipid transfer protein PRELI domain-containing protein 1 (PRELID1). Both proteins participate in the biosynthesis of cardiolipin (CL) (Tatsuta and Langer, 2017), which is required for the assembly and stability of numerous IMM protein complexes, including MCUC. Whereas cardiolipin metabolism was shown to affect MCU-dependent $Ca^{2+}$ uptake (Gottschalk et al, 2019; Ghosh et al, 2020), the functional interactions involved have not been reported yet. Notably, we observed that PRELID1 KD resulted in a dramatic reduction of MCU protein level at both monomeric (Fig. EV2J) and oligomeric states (Fig. 4D), confirming its role in the regulation of MCU stability. As a consequence of

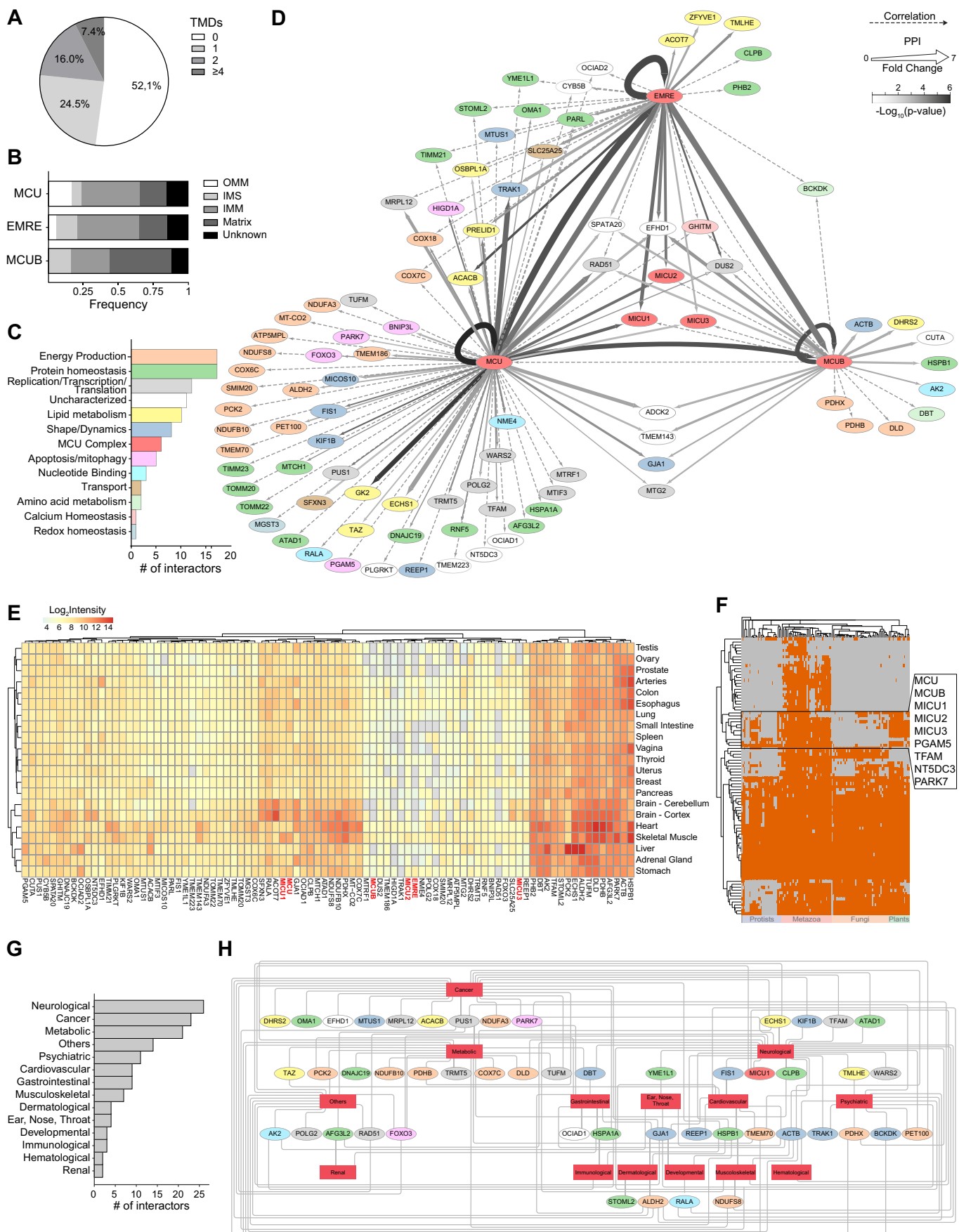

**Figure 3. Systems-wide analysis of the MCUC protein network.**

(A) Percentage of MCUC interactors with transmembrane domains (TMDs). (B) Distribution of bait-specific interactors across submitochondrial compartments. OMM outer mitochondrial membrane, IMS intermembrane space, IMM inner mitochondrial membrane, Unknown missing information or multi-localized proteins. (C) Distribution of MCUC interactors into functional categories. (D) Network of MCUC protein–protein interactions (PPIs) defined by quantitative co-enrichment (solid line), local and global correlation analyses (dashed line). Color and thickness of solid lines indicate statistical significance (*P* value, two-tailed Welch's *t* test) and enrichment (fold change), respectively, whereas the color of each node refers to the protein functional annotation. (E) Heatmap of relative protein expression for MCUC interactors in human tissues. (F) Phylogenetic profiles of MCUC interactors across 120 eukaryotic organisms and *E. coli*. (G) Absolute frequency histogram for known disease classes associated to MCUC interactors. (H) Gene-disease network analysis of MCUC interactors. See also Datasets EV1, EV2, and EV3. Source data are available online for this figure.

impaired MCUC assembly, histamine-stimulated si-PRELID1 HeLa cells showed a marked decrease in $[Ca^{2+}]_{mt}$ (Fig. 4E), without any obvious effect on $[Ca^{2+}]_{cyt}$ transients (Fig. EV2K) and mitochondrial membrane potential (Fig. EV2L).

As a third candidate to test we chose EFHD1, a poorly characterized EF-hands mitochondrial protein (Fig. 4F), whose role in mt-$Ca^{2+}$ homeostasis remains controversial (Meng et al, 2023; Hou et al, 2016; Eberhardt et al, 2022). EFHD1 was recently shown to bind MCU in clear cell renal cell carcinoma (ccRCC) (Meng et al, 2023), and it therefore provides a great example of the predictive power of our resource. Indeed, we identified EFHD1 as a high-confidence and common interactor of all three baits suggesting a strong functional connection to MCUC. To test this hypothesis, we first used aequorin as a luminescent $Ca^{2+}$ sensor to quantify $[Ca^{2+}]_{mt}$ and $[Ca^{2+}]_{cyt}$ transients in HeLa cells upon either stable or transient KD of EFHD1 by shRNA and siRNA treatment, respectively. Out of five distinct shRNAs targeting different regions of EFHD1 mRNA, two (sh1 and sh5) strongly decreased EFHD1 protein level (Fig. 4G) and caused a significant up-regulation of $[Ca^{2+}]_{mt}$ in response to histamine stimulation (Fig. 4H). Likewise, we observed higher $[Ca^{2+}]_{mt}$ upon transfection of HeLa cells with two distinct EFHD1-targeting siRNAs (si1, si2) compared to control (Scr) (Figs. EV3A and 4I). In both experimental set-ups, histamine-stimulated $[Ca^{2+}]_{cyt}$ responses remained unaffected by the silencing of EFHD1 (Fig. EV3B,C). We corroborated these results by quantitative and simultaneous single-cell analyses of $[Ca^{2+}]_{mt}$ and $[Ca^{2+}]_{cyt}$ using a mitochondrial matrix-targeted RCaMP and Fura-2 AM, respectively, as fluorescent $Ca^{2+}$ probes. The measurements were performed in the absence of extracellular $Ca^{2+}$ to study the ER-to-mitochondria $Ca^{2+}$ transfer without the involvement of SOCE. Histamine stimulation in si-EFHD1 but not Scr cells triggered a significant increase in both peak and area under the curve (AUC) for the $[Ca^{2+}]_{mt}$ response, without affecting $[Ca^{2+}]_{cyt}$ transients (Fig. 4J). Similar results were observed in the presence of 1.8 mM $Ca^{2+}$ in the extracellular medium, when both agonist-induced ER $Ca^{2+}$ release and SOCE were activated (Fig. EV3D). To evaluate whether the histamine-stimulated mt-$Ca^{2+}$ phenotype was due to EFHD1-dependent changes in the propagation of $Ca^{2+}$ signals from the ER to the mitochondria, we also quantified $[Ca^{2+}]_{mt}$ and $[Ca^{2+}]_{cyt}$ transients upon opening of SOCE channels at the plasma membrane. To this goal, we pre-treated cells with thapsigargin, an inhibitor of the ER-localized $Ca^{2+}$ ATPase (SERCA) pump, to deplete ER stores and activate SOCE. The addition of 1.5 mM of $Ca^{2+}$ to the extracellular medium of si-EFHD1 HeLa cells resulted in $[Ca^{2+}]_{mt}$ transients of higher peak and amplitude compared to Scr, without a concomitant difference in $[Ca^{2+}]_{cyt}$ dynamics (Fig. 4K), consistent with the results obtained with histamine

stimulation. Next, to understand whether the regulation of mt-$Ca^{2+}$ uptake by EFHD1 was dependent on $Ca^{2+}$ binding, we generated HeLa cells expressing either wild-type (WT) EFHD1 or a mutant in both EF-hand domains (EFHD1$_{EF1+2}$). Although the overexpression of WT EFHD1 did not affect histamine-stimulated $[Ca^{2+}]_{mt}$ response, EFHD1$_{EF1+2}$ cells phenocopied the effect of EFHD1 KD, causing an increase in $[Ca^{2+}]_{mt}$ (Fig. 4L). Importantly, we show that the overexpression of WT EFHD1, but not EFHD1$_{EF1+2}$, was sufficient to rescue the $Ca^{2+}$ phenotype triggered by EFHD1 loss-of-function (Fig. EV3E). This suggests that the binding of $Ca^{2+}$ to EFHD1 is required to exert an inhibitory function on MCUC.

Altogether, we corroborate a role for MICU3 in the modulation of MCUC activity in HEK293 cells, whereby all MICUs co-exist; we spotlight PRELID1 as a candidate protein linking membrane phospholipid metabolism and $[Ca^{2+}]_{mt}$ homeostasis; we identify EFHD1 as a negative modulator of mt-$Ca^{2+}$ uptake.

## EFHD1 is an IMS protein that regulates MCUC activity through MICU1 and affects the viability of breast and cervical cancer cells

To understand the mechanism of EFHD1-dependent inhibition of mt-$Ca^{2+}$ uptake, we measured $Ca^{2+}$ transients in mitochondria of digitonin-permeabilized mt-AEQ HeLa cells upon EFHD1 KD. The addition of exogenous $Ca^{2+}$ triggered a dramatic increase of $[Ca^{2+}]_{mt}$ compared with control pLKO cells, and it was dependent on MCU activity given it was fully abrogated by pre-treatment with the ruthenium red derivative Ru360 (Fig. 5A). Because the impairment of IMM polarization and OXPHOS can also affect mt-$Ca^{2+}$ uptake, we measured bioenergetics and mitochondrial membrane potential in EFHD1 knockdown HeLa cells. The oxygen consumption rate (OCR) of sh-EFHD1 cells at resting conditions, and after treatment with the RCCV inhibitor oligomycin and the uncoupler carbonyl cyanide m-chlorophenyl hydrazone (CCCP) was comparable to control cells, demonstrating that ATP-coupled respiration was unaffected and membrane potential was preserved (Fig. 5B). Similarly, when directly assessing glycolytic function, sh-EFHD1 and pLKO cells were indistinguishable (Fig. 5C) and the mitochondrial membrane potential measured with the potentiometric probe TMRM under both stable shRNA and transient siRNA-mediated silencing of EFHD1 was not affected (Fig. EV4A). These results are consistent with our findings that the basal $[Ca^{2+}]_{mt}$ was not affected by EFHD1 knockdown (Fig. EV4B). Next, we tested whether the inhibitory effect of EFHD1 on MCU-dependent $Ca^{2+}$ uptake was due to potential regulation of MCUC abundance or assembly, as previously shown for other uniporter components (Garbincius and Elrod, 2022). However, we found that neither stable nor transient silencing of

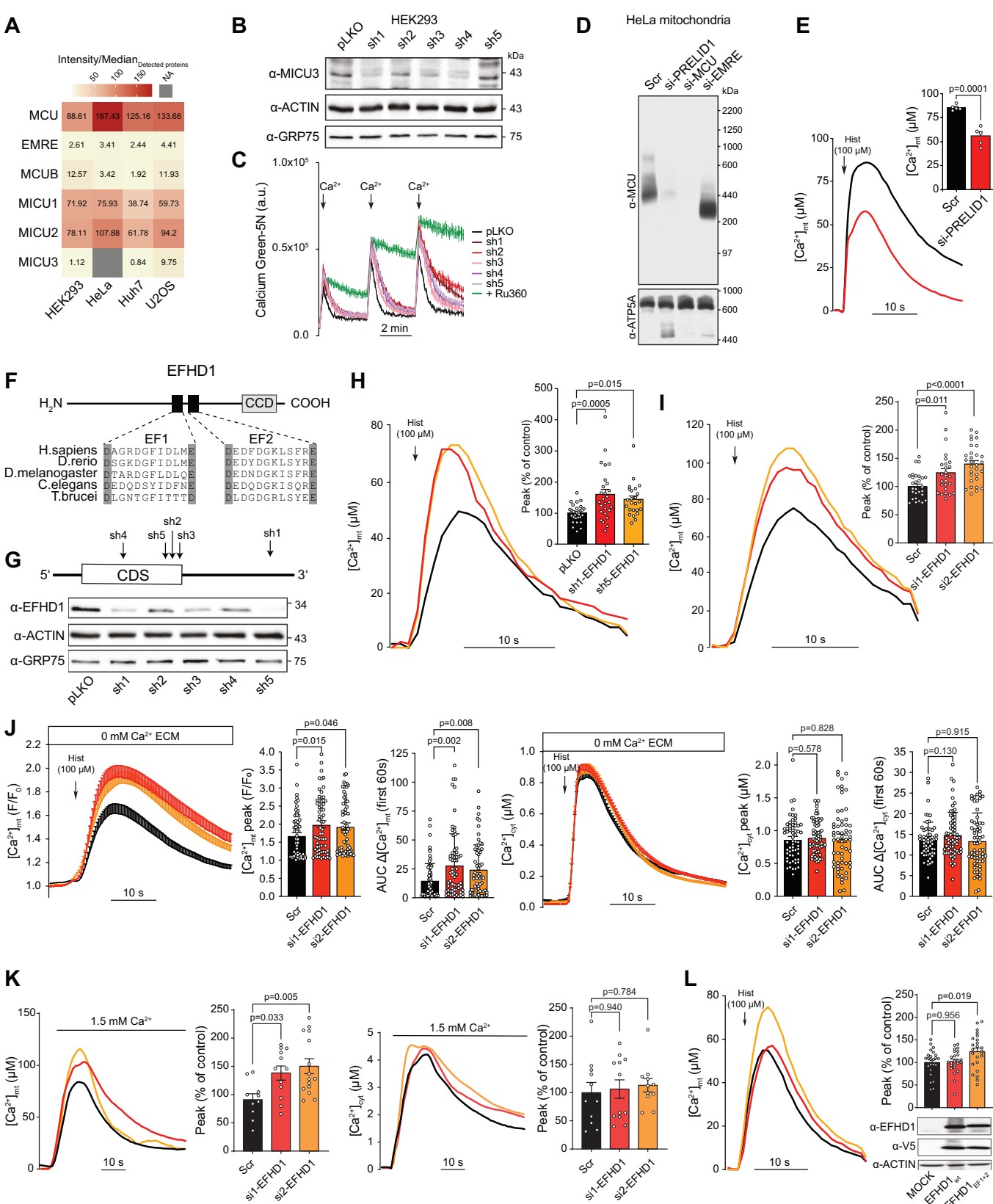

◄ **Figure 4. MICU3, PRELID1, and EFHD1 loss-of-function affect mt-Ca²⁺ homeostasis.**

(A) MS-based quantification of MCUC components in mitochondria isolated from HEK293, HeLa, Huh7 and U2OS human cell lines as in MitoCoP (Morgenstern et al, 2021). The numbers in the heatmap represent the relative expression levels of MCUC components over the median intensity of all detected proteins in the specific proteome of each cell line. NA not assigned. (B) Immunoblot analysis of MICU3, ACTIN (loading control for cytosol), and GRP75 (loading control for mitochondrial) protein level in whole cell lysates from HEK293 cells stably expressing either sh-MICU3 RNAs (sh1-5) or an empty vector (pLKO). (C) Average kinetics of $Ca^{2+}$ clearance by mitochondria of digitonin-permeabilized sh-MICU3 and pLKO HEK293 cells (arrow, injection of 40 μM CaCl₂). Ru360 (10 μM) is used as a positive control for MCU-dependent $Ca^{2+}$ uptake inhibition (mean ± SEM; $n = 4$ biological replicates). (D) BN-PAGE analysis of mitochondria isolated from HeLa cells transfected with siRNAs against PRELID1, MCU, EMRE, and compared to control (Scr). ATP5A is used as a loading control. (E) Representative traces and quantification of $[Ca^{2+}]_{mt}$ transients upon histamine (Hist) stimulation in PRELID1-silenced HeLa mt-AEQ cells (mean ± SEM; $n = 5$ biological replicates); Student's $t$ test. (F) Domain structure of EFHD1 highlighting two evolutionarily conserved EF-hand domains (EF1 and EF2) and a coiled-coil domain (CCD). (G) Immunoblot analyses of EFHD1 protein level in whole cell lysates from shRNA-mediated EFHD1 knockdown (sh1-5) and control (pLKO) cells. CDS coding sequence. GRP75 is used as a loading control for mitochondrial proteins. (H, I) Representative traces and quantification of $[Ca^{2+}]_{mt}$ transients upon histamine (Hist) stimulation in HeLa mt-AEQ cells expressing either EFHD1-targeting shRNAs (H) or siRNAs (I) (mean ± SEM; $n > 23$ biological replicates for (H) and $n > 26$ biological replicates for (I); one-way ANOVA with Dunnett's multiple comparisons test. (J) Average time courses and quantification of histamine-induced $[Ca^{2+}]_{mt}$ (left panel) and $[Ca^{2+}]_{cyt}$ responses (right panel) in si-EFHD1 HeLa cells. Peak and area under the curve (AUC) are calculated for the first 60 s of histamine (Hist) stimulation (mean ± SEM; Scr, $n = 60$; si1-EFHD1, $n = 65$; si2-EFHD1, $n = 63$ cells from three independent experiments); Student's $t$ test. (K) Representative traces and quantification of $[Ca^{2+}]_{mt}$ (left panel) and $[Ca^{2+}]_{cyt}$ (right panel) responses upon 1.5 mM $Ca^{2+}$ induced SOCE activation in si-EFHD1 HeLa mt-AEQ cells (mean ± SEM; $n \geq 10$ biological replicates); one-way ANOVA with Dunnett's multiple comparisons test. (L) Representative traces and quantification of $[Ca^{2+}]_{mt}$ transients upon histamine (Hist) stimulation in HeLa mt-AEQ cells transfected with a C-terminal V5-tagged WT or mutant EFHD1 in both EF-hand domains (EFHD1$_{EF1+2}$; EF1: D231A, E242K; EF2: D421A, E432K). Indel shows immunoblot analysis of EFHD1 protein expression. ACTIN is used as a loading control (mean ± SEM; $n > 24$ biological replicates); one-way ANOVA with Dunnett's multiple comparisons test. See also Figs. EV2–EV3. Source data are available online for this figure.

EFHD1 in HeLa cells affected the expression of known MCUC subunits (Fig. EV4C). Similarly, BN-PAGE of mitochondria from sh-EFHD1 HeLa cells did not reveal a significant change in the macromolecular profile of the complex compared to control cells (Fig. EV4D), indicating that both stability and assembly of MCUC are preserved in the absence of EFHD1.

To gain insights into the functional relationship between EFHD1 and MCUC, we characterized EFHD1 localization and topology. First, we performed a proteinase K (PK) assay on crude mitochondria isolated from HeLa cells (Fig. 5D). Proteins that are localized to the OMM and are exposed to the cytosol, such as TOM20, were immediately digested upon addition of PK, even in the absence of digitonin, while IMS-facing (TIM23) or matrix-localized (HSP60 and CypD) proteins were protected from PK until subsequent permeabilization of the OMM or IMM, respectively. EFHD1 showed a protein digestion profile comparable to that of TIM23 and consistent with an IMS localization. We obtained similar results when employing a PEGylation assay, in which OMM and IMM were selectively permeabilized by increasing concentrations of digitonin in the presence of maleimide functionalized polyethylene glycol (mPEG) (Fig. EV4E). This reagent selectively adds a ~5 kDa polyethylene glycol (PEG) polymer chain on free thiol groups of cysteines and is small enough to cross the OMM and directly react with IMS proteins. As it was reported for MICU1 (Tsai et al, 2016), EFHD1 was pegylated even in the absence of digitonin, in contrast to the mitochondrial matrix protein, SOD2. Consistent with the lack of a predicted transmembrane domain, we found that EFHD1 was only associated with, but not spanning the IMM, because it was recovered in the soluble fractions of mitochondria from HeLa and HEK293 cells upon carbonate extraction, at both low and high pH, as MICU1 (Fig. 5E). The presence of EF-hand domains, EF-hand-dependent mt-Ca²⁺ uptake regulation, and interaction with MCUC, together with evidence of an IMS localization, raised the hypothesis that EFHD1 functions is linked to other MICUs. Although EFHD1 was unable to oligomerize through cysteine-mediated disulfide bonds (Fig. EV4F) and to affect the formation of MICU1-MICU2 heterodimers, just like MICU2, it was cross-stabilized by MICU1 (Fig. 5F). Thus, we speculated that MICU1 could mediate the binding of EFHD1 to

MCUC by affecting its intra-protein stability. Indeed, upon MICU1 silencing we found that the interaction between EFHD1 and MCU-tag was dramatically impaired, with a fourfold reduction in fold change and a loss of significance compared to control (Fig. 5G).

Since the sustained increase of $[Ca^{2+}]_{mt}$ has been shown to promote cell death, we speculated that high EFHD1 expression would protect cancer cells from pro-apoptotic triggers. As shown in Fig. 5H, the KD of EFHD1 in HeLa cells significantly decreased cell viability and sensitized cells to sub-lethal doses of apoptotic inducers such as C2-ceramide and paclitaxel (Fig. 5I). We next set out to evaluate the relevance of EFHD1 more broadly in cancer biology. Indeed, expression profiling of a subset of tumor types, such as clear cell renal cell carcinoma (ccRCC) and colorectal cancer, has recently proposed the expression level of EFHD1 as a prognostic factor and biomarker (Meng et al, 2023). However, a comprehensive assessment of its molecular signature in cancer cells and tissues is currently lacking. To this end, we analyzed publicly available cancer cell lines, primary tumors and patient datasets from the Cancer Cell Line Encyclopaedia (CCLE) of the Dependency Map Consortium (DepMap) (Ghandi et al, 2019), Gene Expression Omnibus of the National Centre for Biotechnology Information (NCBI-GEO), Cancer Genome Atlas (TCGA), Therapeutically Applicable Research to Generate Effective Treatments (TARGET), and Genotype-Tissue Expression (GTEx) repositories (Jiang et al, 2020). Compared to known components of MCUC such as MCU, EMRE, MICU1, MICU2, and MCUB, EFHD1 showed great variability in gene expression across 1450 cancer cell lines from 29 different lineages (Fig. EV5A) but was consistently upregulated in breast, uterine, ovarian, and cervical cancer cells, both at the RNA and protein (Fig. EV5B) level. Moreover, the expression of EFHD1 in primary breast tumor biopsies was significantly higher compared to the adjacent healthy tissues (Fig. EV5C), and breast cancer patients with the highest EFHD1 expression exhibited a decreased response to chemotherapy (Fig. EV5D) and a lower probability of survival (Fig. EV5E). Consistently, downregulation of EFHD1 in two breast cancer cell types, HCC1500 and EFM19, with the highest EFHD1 protein expression compared to the median of all CCLE lines (5.99-fold

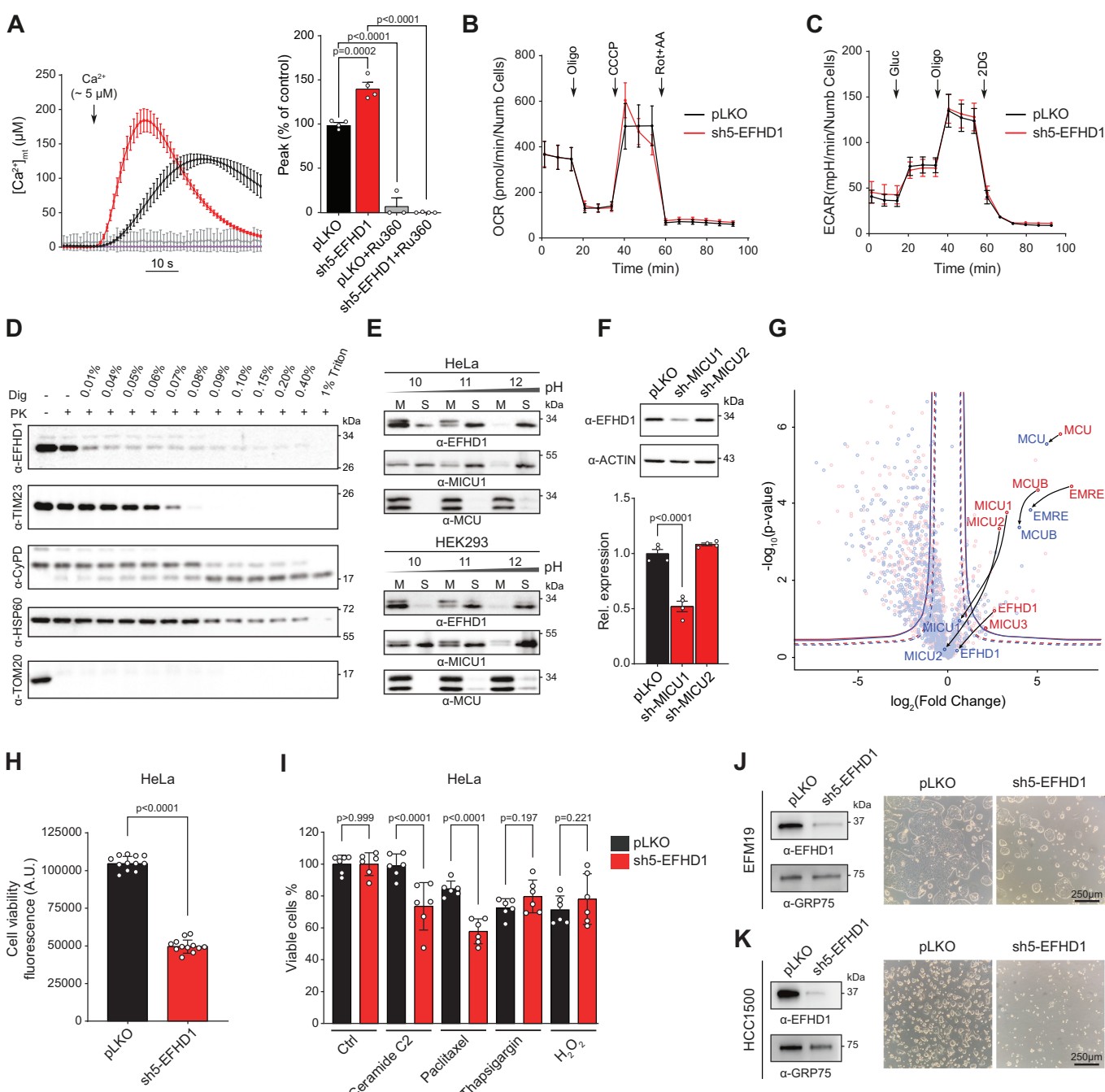

and 5.91-fold, respectively), resulted in a dramatic decrease in cell viability and proliferation (Fig. 5J,K) that was not accompanied by a change in invasive potential (Fig. EV5F).

## Genetic perturbation of MCUC greatly remodels the MCU protein network

Mapping of the MCUC network at resting conditions clearly highlighted that the molecular composition of MCUC and the functional associations between mitochondria and Ca²⁺ are far more complex than previously anticipated (Sancak et al, 2013). Moreover, to what extent genetic perturbations of MCUC impact function,

structure, and regulation of mitochondrial processes has not been systematically assessed so far. To this goal, we sought to map the remodeling of MCUC interactome upon LOF of EMRE, MICU1, MICU2, and MCUB. We generated isogenic MCU-tagged HEK293 cell lines stably expressing either short hairpin RNAs (shRNAs) targeting MICU1, MICU2, EMRE and MCUB, or pLKO as a control. As shown in Fig. 6A, we achieved an almost complete KD of each target. As previously reported (Kamer and Mootha, 2014), we observed that MICU1 KD resulted in the concomitant reduction of MICU2 and EMRE protein levels in whole cell lysates, whereas MICU1 expression was not affected by silencing of EMRE, MICU2 or MCUB. Instead, neither MICU2 nor MCUB downregulation altered

**Figure 5.  EFHD1 interaction with MCUC depends on MICU1 and affects the viability of breast and cervical cancer cells.**

(A) Quantification of $[Ca^{2+}]_{mt}$ upon addition of $Ca^{2+}$ to permeabilized control (pLKO) and sh-EFHD1 mt-AEQ-HeLa cells (mean ± SD; $n \geq 3$ biological replicates); one-way ANOVA with Dunnett's multiple comparisons test. (B) Normalized oxygen consumption rate (OCR) in control and sh-EFHD1 HeLa cells in response to oligomycin (Oligo, 1.5 µM), CCCP (1.5 µM), and Rotenone/Antimycin A (Rot/AA, 4 µM/2 µM), (mean ± SD; $n \geq 10$ biological replicates). (C) Normalized extracellular acidification rate (ECAR) upon addition of glucose (Gluc, 10 mM), oligomycin (Oligo, 1.5 µM), and 2-deoxyglucose (2-DG, 100 mM), (mean ± SD; $n \geq 10$ biological replicates). (D) Immunoblot analysis of EFHD1 in mitochondria from HeLa cells treated with Proteinase K (PK) at increasing digitonin (Dig) concentrations. TOM20, TIM23, Cyclophilin D (CyPD), and HSP60 are used as positive controls for OMM, IMS, and matrix proteins, respectively. (E) Immunoblot analysis of EFHD1 in soluble (S) and membrane (M) fractions of mitochondria from HeLa and HEK293 cells after alkaline carbonate extraction at pH 10, pH 11, and pH 12. MICU1 and MCU are used as positive controls for membrane-associated and integral membrane proteins, respectively. (F) EFHD1 abundance in whole cell lysates from control, sh-MICU1, and sh-MICU2 HEK293 MCU-flag cells. ACTIN is used as a loading control (mean ± SEM; $n = 4$ independent experiments); one-way ANOVA with Dunnett's multiple comparisons test. (G) Volcano plot of mitochondrial proteins enriched in MCU-TAP upon sh-MICU1 (blue) compared to pLKO control (red). Continuous and dashed lines indicate specific interaction partners defined based on a permutation-based FDR of either 0.05 or 0.10, respectively. (H) Viability of HeLa cells after stable EFHD1 KD compared to control (pLKO) (mean ± SD; $n = 12$ biological replicates); Student's $t$ test. (I) Percentage of viable cells in HeLa pLKO and EFHD1 KD after treatment with apoptotic inducers (C2-ceramide, 40 µm; paclitaxel, 50 nM; thapsigargin, 500 nM; $H_2O_2$, 0.5 mM), compared to untreated cells (mean ± SD; $n = 6$); Fisher's LSD test. (J, K) Immunoblot analysis of EFHD1 (left) and representative images (right) of EFM19 (E) and HCC1500 (F) cells upon shRNA-mediated silencing of EFHD1. GRP75 is used as a loading control. See also Figs. EV4–EV5. Source data are available online for this figure.

the overall stability of other MCUC members. We then performed TAP-MS analyses upon tetracycline-inducible expression of StrepII-HA-His$_6$ tagged MCU in stable pLKO and shRNA-expressing HEK293 cells in at least 4 biological replicates (Dataset EV4). Principal component analysis showed great reproducibility across biological replicates and a clear separation between the interactome of MCU in control (pLKO) versus EMRE and MCUB KD, while clustering together with MICU1 and MICU2 KD, suggesting there are minimal changes to the MCU interaction network upon MICUs KD (Fig. 6B). By quantitative bait-prey co-enrichment analysis and local correlation of protein intensity profiles, we mapped PPIs between MCU and 245 mitochondrial proteins (Dataset EV4). As expected, we observed that the interaction of MICU2 with MCU was dependent on MICU1 (Plovanich et al, 2013; Kamer and Mootha, 2014; Patron et al, 2014) and that silencing of EMRE resulted in the loss of MICUs (Sancak et al, 2013) as significant binding partners (Fig. 6C). Interestingly, although the MICU1-MCU interaction was not affected by MICU2 KD, MICU3 was not significantly co-enriched, suggesting that in absence of MICU2, the MICUs dimers are mostly formed by MICU1. On the contrary, the interaction of MCU with MCUB was always preserved, except upon silencing of MCUB, indicating that MCU and MCUB form a stable complex. Strikingly, the global analysis of all mapped MCUC interactomes spotlighted a dramatic remodeling of PPIs upon LOF of EMRE and MCUB (Fig. 6D,E). Downregulation of EMRE resulted in more than tenfold lower prey recovery. However, although the loss of EMRE leads to non-functional MCU channels (Sancak et al, 2013; Liu et al, 2020; König et al, 2016), we also identified novel and significantly enriched interactions between MCU and proteins of the TCA cycle that are known to be regulated by $Ca^{2+}$, for example, the pyruvate dehydrogenase complex (Fig. 6F). It is tempting to speculate that such PPIs could mediate a compensatory metabolic response aimed at preserving mitochondrial energy production. This could explain why, despite both MCU and EMRE knockout (KO) mouse models displaying a complete loss of mt-$Ca^{2+}$ uptake, only MCU KO mice showed impaired exercise performance (Liu et al, 2020; Pan et al, 2013). In the opposite direction, MCUB KD, which leads to increased $[Ca^{2+}]_{mt}$, neither affected MCUC protein stability nor the interaction of EMRE and the MICUs with MCU. However, it substantially remodeled the MCU interactome, mostly by allowing the gain of novel interactions. These involved mitochondrial proteins that regulate stress response pathways, cell death, and mitochondrial translation. To corroborate these results, we tested whether a wider protein interaction network would also be reflected in the formation of MCU-containing

complexes with higher MW upon sh-MCUB. Blue native-polyacrylamide gel electrophoresis (BN-PAGE) analysis of mitochondria isolated from pLKO MCU-tagged HEK293 cells identified the expected bands around 480 kDa and above 720 kDa and showed that silencing of MICU1 or EMRE caused a shift toward a lower MW (Plovanich et al, 2013; Sancak et al, 2013), while MICU2 LOF did not affect the overall assembly (Plovanich et al, 2013) (Fig. 6G). Instead, stable (shRNA) and transient (siRNA) MCUB KD resulted in the shift of MCU-containing complexes toward a higher MW, both in HEK293 (Fig. 6G) and HeLa cells (Fig. 6H). We observed the same phenotype in mitochondria isolated from the brain of MCUB KO mice, where we and others (Samaras et al, 2020; Hansen et al, 2024) could detect a clear expression of MCUB (Fig. EV6A–C). Moreover, two-dimensional blue native/SDS analysis of whole brain mitochondria from MCUB KO confirmed that MCU-containing complexes were shifted toward a higher MW (Fig. 6I). In summary, our results suggest that the extent to which MCU engages in PPIs within the organelle is determined by channel activity and $[Ca^{2+}]_{mt}$ and that MCUB can act as a protein barrier (Fig. 6J), by preventing PPIs between MCU and several mitochondrial complexes and pathways.

## Discussion

Proteins typically exert their function by functionally and physically interacting. To this end, several biochemical approaches have so far been exploited to map the protein interactome landscape of the uniporter in HEK293 cells (Antonicka et al, 2020; Sancak et al, 2013; Austin et al, 2022), mostly using MCU as a bait and upon its overexpression, a condition known to affect mitochondrial physiology and cell survival (De Stefani et al, 2011; D'Angelo et al, 2023; Mammucari et al, 2015). As a result, MCU, MCUB, EMRE, MICU1 and MICU2 were proposed to represent all the subunits of the so-called "holocomplex" in this cell type (Sancak et al, 2013). Compared to those analyses, we were able to expand the "holocomplex" by mapping 139 statistically significant PPIs between known members of the uniporter and an additional 89 mitochondrial proteins localized in all four submitochondrial compartments. Key to our approach was the use of all three membrane-spanning subunits of the complex as baits and the integration of complementary computational strategies to systematically analyze MCUC copurifying proteins. Of utmost importance, our study provides the first snapshot of the MCUC signaling network at near-endogenous conditions. Besides being well

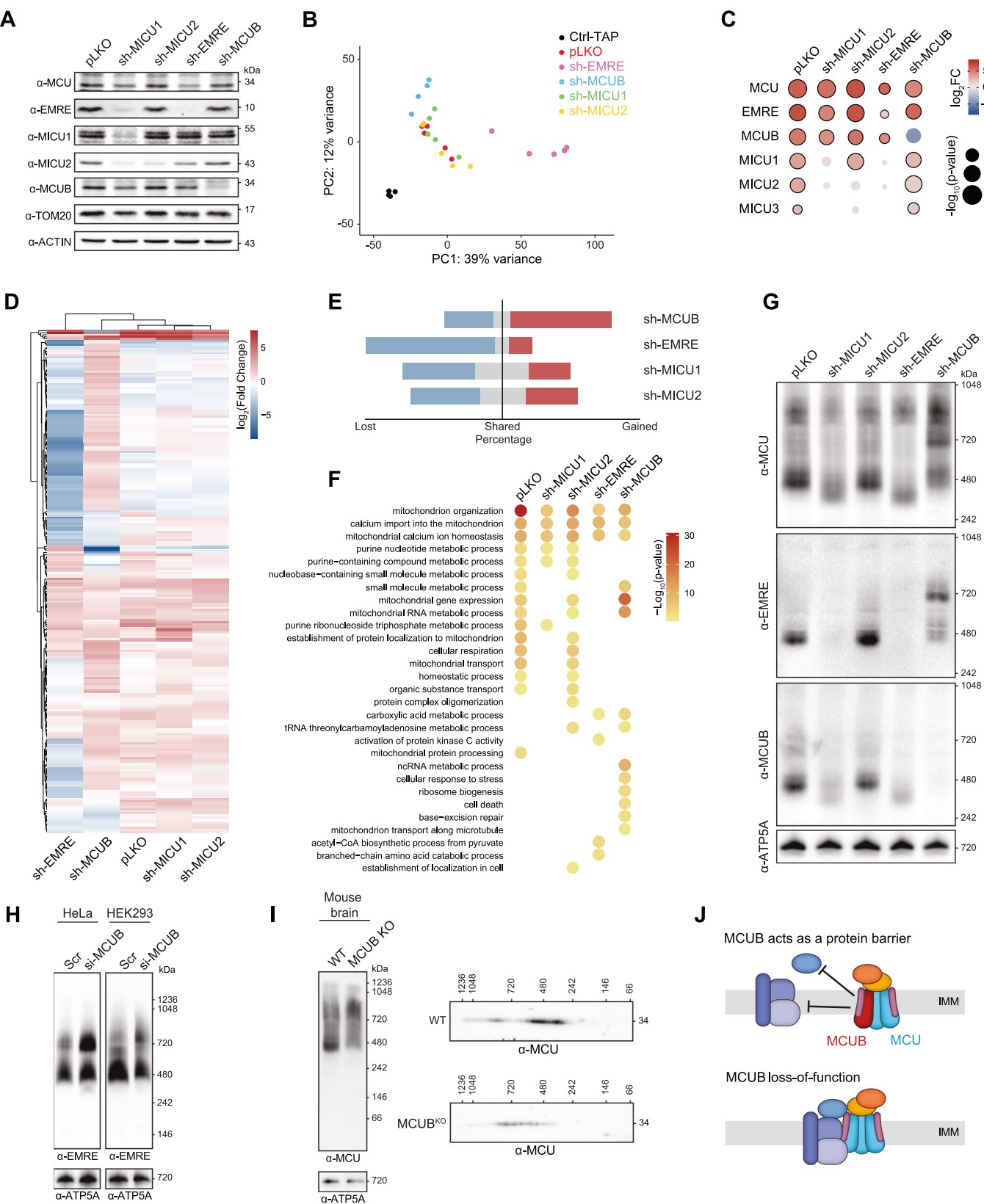

**Figure 6.   Remodeling of the MCU interactome upon genetic perturbation of the uniporter complex.**

(A) Immunoblot analysis of known MCUC components in Flp-In T-REx HEK293 cells expressing MCU as a bait and infected with either control (pLKO) or shRNAs targeting MICU1, MICU2, EMRE and MCUB. ACTIN and TOM20 are used as loading controls. (B) Principal Component (PC) Analysis of TAPs from MCU-tagged and parental cell lines. (C) Quantitative bait-prey co-enrichment analysis of the previously known MCUC components upon genetic perturbation. Dots with a black border indicate significant interactions based on FDR cutoff = 0.10 (*P* value, two-tailed Welch's *t* test). (D) Heatmap showing differences in the $Log_2$ fold change among MCU interactors across all tested conditions. Hierarchical clustering was performed using the Euclidean distance with complete linkage. (E) Percentage of gained (red), shared (gray), and lost (blue) interactors in each condition compared to MCU-TAPs and calculated based on the union of TAPs from MCU pLKO and shRNAs. (F) Pathway enrichment analysis of the significant interactors for pLKO and shRNAs conditions (Fisher's one-tailed test with g:SCS multiple-testing correction algorithm (Raudvere et al, 2019)). (G) BN-PAGE analysis of mitochondria isolated from Flp-In T-REx HEK293 cells expressing MCU-tagged and either pLKO or shRNAs. ATP5A is used as a loading control. (H) BN-PAGE analysis of MCUC assembly in mitochondria isolated from control (Scr) and si-MCUB HeLa or HEK293 cells. ATP5A is used as a loading control. (I) Blue-Native/SDS-PAGE analysis of MCUC in isolated mitochondria from WT and MCUB KO mouse brains. ATP5A is used as a loading control. (J) Model of MCUB as a protein barrier: Incorporation of MCUB into the MCUC obstructs protein–protein interactions between MCU and various mitochondrial complexes and pathways. Loss-of-function of MCUB enhances the number of PPIs involving MCU and facilitates interaction with different mitochondrial functions. See also Fig. EV6, Datasets EV4 and EV5. Source data are available online for this figure.

supported by recent literature and providing novel insights, our results identify proteins that regulate the MCUC activity as well as functional associations between MCU-mediated mt-$Ca^{2+}$ uptake and numerous mitochondrial complexes and pathways. As an example, we recovered functional interactions between MCU, EMRE, the YME1L proteolytic hub, and the m-AAA protease AFG3L2 that are required for MCUC biogenesis and assembly (König et al, 2016; Tsai et al, 2022, 2017). We also mapped several molecular links between the MCUC, energy production, and cell death activation, mitochondrial functions that are known to be regulated by calcium (Rizzuto et al, 2012). As MCUC interactors, we also identified proteins involved in mt-DNA maintenance and replication, mRNA transcription and protein translation, as well as enzymes of the beta-oxidation pathways (e.g., ECHS1, ACACB, ACOT7). Interestingly, the impairment of mt-$Ca^{2+}$ uptake was already associated with the rewiring of energy production from glycolysis to fatty acid oxidation in the skeletal muscle (Kwong et al, 2018; Gherardi et al, 2018; Huo et al, 2023), but the molecular mechanisms underlying such metabolic flexibility remain unknown.

The high coverage and sensitivity of our approach were further corroborated by the identification of MICU3 as a bona fide component of MCUC in HEK293 cells. None of the previous PPIs analyses (Sancak et al, 2013; Antonicka et al, 2020; Austin et al, 2022) recovered MICU3 within the set of MCU copurifying proteins, possibly due to its low expression level in HEK293 cells (Morgenstern et al, 2021), rather than its tissue-specific expression (Plovanich et al, 2013; Patron et al, 2019). Our MCUC interactome represents also a great resource to discover additional regulatory mechanisms of mt-$Ca^{2+}$ signaling. For example, we identified EFHD1 as a high-confidence binding partner of all three baits. So far, EFHD1 has been linked to mitochondrial flashes (Hou et al, 2016; Eberhardt et al, 2022), ROS signaling (Eberhardt et al, 2022), bioenergetics (Ulisse et al, 2020), pro-B immune cell development and maturation (Stein et al, 2017), as well as cell survival and proliferation (Meng et al, 2023). However, its function in mitochondria and the mechanisms responsible for such a pleiotropic effect in different cell and tissue types remain poorly understood. Its involvement in $[Ca^{2+}]_{mt}$ homeostasis has been explored in previous reports but with some inconsistencies (Hou et al, 2016; Meng et al, 2023; Eberhardt et al, 2022). Whereas EFHD1 silencing in HeLa cells was not associated to a defect in $[Ca^{2+}]_{mt}$ upon either osmotic stress or histamine stimulation (Hou et al, 2016), a slight decrease in $[Ca^{2+}]_{mt}$ was observed after EFHD1 KO in cardiomyocytes (Eberhardt et al, 2022), potentially due to a

drop in the mitochondrial membrane potential. On the other hand, the overexpression of EFHD1 in ccRCC was found to significantly reduce $[Ca^{2+}]_{mt}$ (Meng et al, 2023). To solve this conundrum, we performed complementary and quantitative measurements of intracellular $Ca^{2+}$ dynamics upon either stable or transient KD of EFHD1 in HeLa cells, employing both luminescence- and fluorescence-based $Ca^{2+}$ assays. Collectively, our results indicate that EFHD1 functions as an inhibitor of MCU-mediated $[Ca^{2+}]_{mt}$ uptake. This role is neither due to changes in upstream $Ca^{2+}$ signaling pathways nor in oxidative metabolism, but seems to be dependent on its interaction with the MCUC, observation supported by its cross-stabilization with MICU1. This possibly allows EFHD1 to sense changes in $[Ca^{2+}]_{mt}$ and regulate MCUC activity accordingly, as performed by other EF-hand-containing proteins such as the MICUs (Csordás et al, 2013; Plovanich et al, 2013; Patron et al, 2019). Importantly, the heterogeneous expression pattern of EFHD1 among human tissues and cancer cell lines could also explain why EFHD1 loss- or gain-of-function would affect $[Ca^{2+}]_{mt}$ homeostasis to different extents in different cell types. Accordingly, in HeLa cells that already express EFHD1 at a high level, we failed to observe an alteration in $[Ca^{2+}]_{mt}$ upon EFHD1 overexpression. Instead, its overexpression in cells like ccRCC that show a 30-fold lower level of endogenous EFHD1 compared to HeLa cells was found to significantly reduce $[Ca^{2+}]_{mt}$ (Meng et al, 2023). As a selective determinant of cell survival and cancer progression, EFHD1 surely represents an important therapeutic target for further investigations.

One of the most characteristic features of MCUC is the deep plasticity to adapt to and compensate for mitochondrial dysfunctions. Indeed, the cross-regulation of protein level and stability among its components represents one of the main mechanisms behind compensatory adaptation observed upon chronic loss or gain of mt-$Ca^{2+}$ uptake in cells and mouse models (Pan et al, 2013; Liu et al, 2020). For example, the reduction of $[Ca^{2+}]_{mt}$, by increasing MCUB:MCU (Huo et al, 2020; Lambert et al, 2019) or MICU1:MCU ratio (Paillard et al, 2022), allows to protect mitochondria from $Ca^{2+}$ overload and associated cell death. Conversely, enhancing uniporter activity counteracts the bioenergetic deficit due to OXPHOS impairment and mitochondrial cardiomyopathies upon TFAM deletion (Balderas et al, 2022), as well as sustained adaptive thermogenesis in the brown adipose tissue upon adrenergic stimulation (Xue et al, 2022). To gain insights into the remodeling of MCUC interactome upon genetic

perturbation, we also performed TAPs followed by LC-MS/MS analysis from MCU-tagged cells where we stably silenced MICU1, MICU2, EMRE or MCUB. Our integrative analysis of the MCUC PPIs upon loss- or gain-of-function in mt-$Ca^{2+}$ uptake, demonstrates that the MCU interactome is a flexible and adaptable network to perturbations and spotlights MCUB as a protein barrier. MCUB was initially described as a dominant negative regulator of $[Ca^{2+}]_{mt}$, due to critical substitution in its highly conserved transmembrane domain, which could impact the permeation of $Ca^{2+}$ through the channel (Raffaello et al, 2013). We observed that silencing of MCUB led to a more extended MCUC network and was accompanied by an increased formation of high MW complexes of MCU, both in vitro and in vivo. These findings are also consistent with recent evidence showing that MCUB expression in cardiomyocytes leads to a reduction in MCUC size and disrupts its interaction with MICU1 and MICU2 (Lambert et al, 2019; Huo et al, 2020). However, our results indicate that the formation of MCU complexes with higher MW is not simply due to changes in PPIs between the known MCUC members, but to a wider remodeling of the MCU interactome.

In summary, our MCUC interaction map under resting conditions and genetic perturbation of the different MCUC components represents a great tool to (1) identify novel candidate proteins and pathways involved in $[Ca^{2+}]_{mt}$ signal transduction cascades; (2) provide new insights into both players and mechanisms regulating tissue-specific MCUC activity; (3) uncover novel genetic underpinnings and pharmacological targets to develop new therapeutical strategies for several human diseases.

## Methods

### Cell lines

HeLa and HEK293 cells were cultured in Dulbecco's modified Eagle's medium (DMEM) supplemented with 10% FBS. HeLa cells stably expressing a mitochondrial matrix-targeted GFP-aequorin (mt-AEQ) (Arduino et al, 2017) were cultured in DMEM supplemented with 10% FBS and 100 µg/mL geneticin (Thermo Fisher Scientific, 10131027). EFM19 cells (German Collection of Microorganisms and Cell Cultures GmbH) and HCC1500 cells (American Type Culture Collection) were cultured in RPMI1640 media supplemented with 10% FBS. All cell lines were kept at 37 °C in an incubator with 5% $CO_2$.

The Flp-In T-REx system was used to generate isogenic, stable HEK293 cell lines exhibiting tetracycline-inducible expression of tagged bait proteins for tandem affinity purification (TAP). The open reading frame (ORF) of each bait protein (MCU, MCUB, and EMRE) was cloned into the pcDNA5/FRT/TO Flp-In expression vector, in frame with a C-terminal StrepII$_2$-HA-His$_6$-tag and under the control of a tetracycline-regulated CMV/TetO2 promoter. The resulting MCU, EMRE, or MCUB Flp-In expression vectors were co-transfected with the Flp recombinase vector, pOG44, into Flp-In T-REx HEK293 cells, which contained a single integrated FRT site and stably expressed the lacZ-ZeocinTM fusion gene (pFRT/lacZeo) and the tetracycline repressor (pcDNA6/TR). This allowed the targeted integration of each bait protein at a single transcriptionally active genomic locus that is the same in every cell line. Flp-In T-REx HEK293 cells were grown in DMEM with 10% FBS, containing blasticidin (15 µg/mL) and Zeocin (100 µg/mL). Stable transfectants were selected and maintained in DMEM with 10% FBS containing blasticidin (15 µg/mL) and 100 µg/mL hygromycin. The expression of tagged bait proteins was induced by

**Reagents and tools table**

| Reagent/resource | Reference or source | Identifier or catalog number |
|---|---|---|
| **Experimental models** | | |
| C57BL/6N mice | Charles River Laboratories | C57BL/6N |
| C57BL/6N Mcub KO mice | Feno et al, 2021 | – |
| EFM19 cells | DSMZ | ACC 231 |
| EFM19 cells pLKO | Generated for this study | – |
| EFM19 cells sh5-EFHD1 | Generated for this study | – |
| Flp-In™-293 cells | Thermo Fisher Scientific | R75007 |
| Flp-In™-293 cells MCU-TAP | Generated for this study | – |
| Flp-In™-293 cells EMRE-TAP | Generated for this study | – |
| Flp-In™-293 cells MCUB-TAP | Generated for this study | – |
| Flp-In™-293 cells MCU-TAP sh-EMRE | Generated for this study | – |
| Flp-In™-293 cells MCU-TAP sh-MICU1 | Generated for this study | – |
| Flp-In™-293 cells MCU-TAP sh-MICU2 | Generated for this study | – |
| Flp-In™-293 cells MCU-TAP sh-MCUB | Generated for this study | – |
| HCC1500 cells | ATCC | CRL-2329 |
| HCC1500 cells pLKO | Generated for this study | – |
| HCC1500 cells sh5-EFHD1 | Generated for this study | – |
| HEK293 cells | ATCC | CRL-1573 |
| HEK293 cells pLKO | Generated for this study | – |
| HEK293 cells sh1-MICU3 | Generated for this study | – |

| Reagent/resource | Reference or source | Identifier or catalog number |
|---|---|---|
| HEK293 cells sh2-MICU3 | Generated for this study | – |
| HEK293 cells sh3-MICU3 | Generated for this study | – |
| HEK293 cells sh4-MICU3 | Generated for this study | – |
| HEK293 cells sh5-MICU3 | Generated for this study | – |
| HeLa cells | ATCC | CCL-2 |
| mt-AEQ HeLa cells | Generated previously in the lab. | – |
| mt-AEQ HeLa cells pLKO | Generated for this study | – |
| mt-AEQ HeLa cells sh1-EFHD1 | Generated for this study | – |
| mt-AEQ HeLa cells sh2-EFHD1 | Generated for this study | – |
| mt-AEQ HeLa cells sh3-EFHD1 | Generated for this study | – |
| mt-AEQ HeLa cells sh4-EFHD1 | Generated for this study | – |
| mt-AEQ HeLa cells sh5-EFHD1 | Generated for this study | – |
| mt-AEQ HeLa cells plx304-EFHD1-WT | Generated for this study | – |
| mt-AEQ HeLa cells plx304-EFHD1-MUT | Generated for this study | – |
| **Recombinant DNA** | | |
| Gateway™ pDONR™221 Vector | Thermo Fisher Scientific | 12536017 |
| pLX304 | Addgene | #25890 |
| pLKO.1 puro | Addgene | #8453 |
| pCMV-dR8.91 | Addgene | #202687 |
| pCMV-VSV-G | Addgene | #8454 |
| sh-MCU pLKO.1 puro | Sigma-Aldrich | sh-MCU, TRCN0000133861 |
| sh-EMRE pLKO.1 puro | Sigma-Aldrich | sh-EMRE, TRCN0000145067 |
| MICU1 pLKO.1 puro | Sigma-Aldrich | sh-MICU1, TRCN0000053370 |
| MICU2 pLKO.1 puro | Sigma-Aldrich | sh-MICU2 TRCN0000055848 |
| MCUB pLKO.1 puro | Sigma-Aldrich | sh-MCUB, TRCN0000128550 |
| MICU3 pLKO.1 puro | Sigma-Aldrich | sh1-MICU3, TRCN0000056083 |
| MICU3 pLKO.1 puro | Sigma-Aldrich | sh2-MICU3, TRCN0000056084 |
| MICU3 pLKO.1 puro | Sigma-Aldrich | sh3-MICU3, TRCN0000056085 |
| MICU3 pLKO.1 puro | Sigma-Aldrich | sh4-MICU3, TRCN0000056086 |
| MICU3 pLKO.1 puro | Sigma-Aldrich | sh5-MICU3, TRCN0000056087 |
| EFHD1 pLKO.1 puro | Sigma-Aldrich | sh1-EFHD1, TRCN0000056183 |
| EFHD1 pLKO.1 puro | Sigma-Aldrich | sh2-EFHD1, TRCN0000056184 |
| EFHD1 pLKO.1 puro | Sigma-Aldrich | sh3-EFHD1, TRCN0000056185 |
| EFHD1 pLKO.1 puro | Sigma-Aldrich | sh4-EFHD1, TRCN0000056186 |
| EFHD1 pLKO.1 puro | Sigma-Aldrich | sh5-EFHD1, TRCN0000056187 |
| EFHD1-WT plx304 | Generated for this study | |
| EFHD1-Mut plx304 | Generated for this study | |
| **Antibodies** | | |
| HA | BioLegend | MMS-101R |
| V5 | Thermo Fisher Scientific | R96025 |
| ACTIN | Sigma-Aldrich | A2228 |
| MAIP1 | König et al, 2016 | – |
| ATP5A | Abcam | MS507 |
| Lamin | Santa Cruz | sc-6217 |
| VDAC | Abcam | ab14734 |
| EFHD1 | Sigma-Aldrich | HPA056959 |
| HSP60 | R&D Systems | MAB1800 |
| SOD2 | Antibody Verify Inc. | AAS29585C |
| TIM23 | BD Bioscience | 611222 |

| Reagent/resource | Reference or source | Identifier or catalog number |
|---|---|---|
| CyPD | Abcam | ab110324 |
| GRP75 | Santa Cruz | sc-133137 |
| MCU | Sigma-Aldrich | HPA016480 |
| EMRE | Santa Cruz | sc-86337 |
| MCUB | Santa Cruz | sc-163985 |
| MICU1 | Sigma-Aldrich | HPA037479 |
| MICU2 | Sigma-Aldrich | HPA045511 |
| MICU3 | Sigma-Aldrich | HPA024048 |
| TOM20 | Abcam | ab56783 |
| Goat anti-mouse | Bio-rad | 1706516 |
| Goat anti-rabbit | Bio-rad | 1706515 |
| **Oligonucleotides and other sequence-based reagents** | | |
| Scramble | Sigma-Aldrich | MISSION siRNA #1 SIC001 |
| si1-EFHD1 | Sigma-Aldrich | SASI_Hs01_00228164 |
| si2-EFHD1 | Sigma-Aldrich | SASI_Hs01_00228165 |
| si-MCUB | Sigma-Aldrich | 5'-AUACUACCAGUCACACCAU-3' |
| attB1-FW-EFHD1 | Generated for this study - Metabion | 5'-GGGGACAAGTTTGTACAAAAAAGCAGGCTTAGCCACCATGGCCAGTGAGGAGCTG-3' |
| attB2-RV-EFHD1 | Generated for this study - Metabion | 5'-GGGGACCACTTTGTACAAGAAAGCTGGGTTTGTATTGAAGTTGGCCTTGAGTTT-3' |
| **Chemicals, enzymes, and other reagents** | | |
| 2-DG | Sigma-Aldrich | D6134 |
| Acetonitrile | Sigma-Aldrich | 34851 |
| Ammonium bicarbonate | Sigma-Aldrich | A6141 |
| Anti-FLAG M2 magnetic bead resin | Sigma-Aldrich | M8823-1ML |
| Antimycin A | Sigma-Aldrich | A8674-25MG |
| ATP | Roche | 11140965001 |
| Avidin | IBA Lifesciences | 2-0204-015 |
| Pierce BCA Protein Assay kit | Thermo Fisher Scientific | 23225 |
| β-Mercaptoethanol | Sigma-Aldrich | M6250 |
| Biotin | Sigma-Aldrich | B4501-100MG |
| Blasticidin | Gibco | R210-01 |
| NativePAGE™ Running Buffer Kit | Thermo Fisher Scientific | BN2007 |
| NativePAGE™ Sample Prep Kit | Thermo Fisher Scientific | BN2008 |
| BSA (fatty acid-free) | Sigma-Aldrich | A7030-100g |
| BSA | Sigma-Aldrich | A7906-500G |
| $CaCl_2$ | CalBiochem | 208290 |
| Calcium-Green-5N | Sigma-Aldrich | C3737 |
| C2-Ceramide | Santa Cruz | sc-201375 |
| CCCP | Sigma-Aldrich | C2759 |
| 2-Chloroacetamide | Sigma-Aldrich | 22790 |
| Coelentarizne native | Biozol | BOT-10110-1 |
| Coelenterazine n | Biozol | BOT-10115-1 |
| CyQUANT® Cell Proliferation Assay Kit | Life Technologies | C7026 |
| DDM | Thermo Fisher Scientific | BN2005 |
| Digitonin | Sigma-Aldrich | D141-500MG |
| DMEM | Sigma-Aldrich | D6429 |
| DTT | Omnilab | D9779-10G |
| ECL Prime WB Detection Reagent | Amersham | RPN2232 |
| EDTA | Sigma-Aldrich | E5134 |

| Reagent/resource | Reference or source | Identifier or catalog number |
|---|---|---|
| EGTA | Sigma-Aldrich | E3889 |
| FBS | Sigma-Aldrich | F7524-500ML |
| Formic acid | Sigma-Aldrich | 5.43804 |
| Fura2AM | Merck | 47989-1MG-F |
| Glucose | Merck | 137048 |
| Glycerol | Sigma-Aldrich | G5516 |
| Hematoxylin | Sigma-Aldrich | 1.04302 |
| HCl | VWR | 20257-296 |
| HEPES | Carl Roth | 9105.3 |
| Hygromicin B | Invitrogen | 10687010 |
| Iodoacetamide | Sigma-Aldrich | I6125 |
| Isopropanol | VWR | ACRO444250050 |
| $K_2HPO_4$ | Sigma-Aldrich | P8281 |
| KCl | VWR | 1.04936.1000 |
| $KH_2PO_4$ | Merck | 104873 |
| KOH | Sigma-Aldrich | P1767 |
| 4X Laemmli Sample Buffer | Bio-Rad | 1610747 |
| L-Glutamine | Sigma-Aldrich | G3126 |
| Lipofectamine™ RNAiMAX | Thermo Fisher Scientific | 13778075 |
| LysC | Promega | VA1170 |
| D-Mannitol | Sigma-Aldrich | M4125-1KG |
| Methanol | Sigma-Aldrich | 34860 |
| $MgCl_2$ | Carl Roth | KK36.1 |
| MgSO4 | Sigma-Aldrich | 230391 |
| $Na_2CO_3$ | Sigma-Aldrich | 451614 |
| NaCl | Sigma-Aldrich | S3014 |
| $NaHCO_3$ | Sigma-Aldrich | S6297 |
| NaOH | Sigma-Aldrich | 1.06498 |
| $NH_4OH$ | Sigma-Aldrich | 5.43830 |
| Pierce™ High Capacity Ni-IMAC MagBeads | Thermo Fisher Scientific | A50589 |
| Nitrocellulose membrane | Bio-Rad | 1620112 |
| Nonidet® P 40 Substitute (NP-40) | VWR | PIER85124 |
| Nupage 4%-12% bis-tris 2D | Thermo Fisher Scientific | NP0326BO |
| Oligomycin A | Sigma-Aldrich | 75351-5MG |
| Paclitaxel | Abcam | ab120143 |
| PBS 1X | Life Technologies | 10010015 |
| Pertex mounting medium | VWR | LEIC811 |
| Pluronic F-127 | Thermo Fisher Scientific | P3000MP |
| PMSF | Sigma-Aldrich | 10837091001 |
| Poly-D lysine | Sigma-Aldrich | A3890401 |
| Polyacrylamide | Sigma-Aldrich | A3699-5X100ML |
| Protease Inhibitor Cocktail | Sigma-Aldrich | 5056489001 |
| Proteinase K | Biozym | 351100902 |
| Puromycin | Life Technologies | A1113803 |
| PVDF membrane | Bio-Rad | 1620177 |
| Sodium Pyruvate | Sigma-Aldrich | S8636 |
| Rapigest | VWR | WATE186001861 |
| Resazurin sodium salt | Santa Cruz | sc-206037 |
| Rneasy Mini Kit | Qiagen | 74104 |
| Rotenone | Sigma-Aldrich | R8875 |

| Reagent/resource | Reference or source | Identifier or catalog number |
|---|---|---|
| RPMI1640 | Thermo Fisher Scientific | A1049101 |
| Ru360 | Sigma-Aldrich | 557440 |
| SDC | Sigma-Aldrich | 30970 |
| SDS | Sigma-Aldrich | L3771 |
| D-Sucrose | Carl Roth | 4621.1 |
| Sulfinpyrazone | Sigma-Aldrich | PHR3244 |
| SuperScript™ III SuperMix | Invitrogen | 18080400 |
| TCEP | Sigma-Aldrich | 75259 |
| Tetracycline | Sigma-Aldrich | T7660 |
| TFA | Sigma-Aldrich | 302031 |
| Thapsigargin | VWR | 586005 |
| Thiourea | Sigma-Aldrich | T7875 |
| NuPAGE™ Transfer Buffer (20X) | Thermo Fisher Scientific | NP0006 |
| Tris | Carl Roth | 4855.2 |
| Triton X-100 | Sigma-Aldrich | T9284 |
| Trypsin | Sigma-Aldrich | T4049-100ML |
| Tween-20 | Sigma-Aldrich | P1379 |
| Urea | Sigma-Aldrich | U5128 |
| Milk (Powder) | Carl Roth | T145.2 |
| X-tremeGENE™ HP DNA | Merck | 6366244001 |
| Zeocin | Thermo Fisher Scientific | R25001 |
| **Software** | | |
| Cytoscape 3.10.0 | Shannon et al, 2003 | https://cytoscape.org/ |
| DeepTMHMM | Hallgren et al, 2022 | https://dtu.biolib.com/DeepTMHMM |
| Depmap portal | Ghandi et al, 2019 | https://depmap.org/ |
| DisGeNET | Piñero et al, 2021 | https://www.disgenet.org/ |
| GEPIA2 | Tang et al, 2019 | http://gepia2.cancer-pku.cn/ |
| g:Profiler | Raudvere et al, 2019 | https://biit.cs.ut.ee/gprofiler/gost |
| Prism 10.0 | Graphpad | https://www.graphpad.com/ |
| GTEx | Lonsdale et al, 2013 | https://gtexportal.org/home/ |
| ImageJ | Schneider et al, 2012 | https://imagej.net/ij/ |
| MaxQuant 1.5.5.2 | Cox and Mann, 2008 | https://www.maxquant.org/ |
| MitoCarta3.0 | Rath et al, 2021 | https://www.broadinstitute.org/mitocarta |
| Perseus 1.6.15.0 | Tyanova et al, 2016 | https://www.maxquant.org/ |
| R 4.1.2 release | The R Foundation | https://www.r-project.org/ |
| ROC Plotter | Fekete and Győrffy, 2019 | https://rocplot.org/ |
| TNMplot | Bartha and Győrffy, 2021 | https://tnmplot.com/ |
| Uniprot | Bateman et al, 2015 | https://www.uniprot.org/ |
| **Other** | | |
| PowerPac HC Power Supply | Bio-Rad | |
| Mini-PROTEAN Tetra Cell electrophoresis chamber | Bio-Rad | |
| Trans-Blot® Turbo™ Transfer System | Bio-Rad | |
| CLARIOstar Plus multiplate reader | BMG Labtech | |
| Bioruptor® Plus sonication device | Diagenode | |
| EASY-nLC 1000 HPLC | Thermo Fisher Scientific | |
| ORCA-Flash 4.0v3 sCMOS camera | Hamamatsu | |
| 12-Tube Magnetic Separation Rack | New England Biolabs | |
| BX43 Microscope | Olympus | |
| IX81 Microscope | Olympus | |

| Reagent/resource | Reference or source | Identifier or catalog number |
|---|---|---|
| UAPO 340 Objective | Olympus | |
| 4639 Cell Disruption Vessel by nitrogen | Parr Instrument Company | |
| MicroBeta2 LumiJET™ Detector | PerkinElmer | |
| Q Exactive HF mass spectrometer | Thermo Fisher Scientific | |
| Seahorse XFe96/XF96 Analyzer | Agilent | |
| Tissue grinders Potter-Elvehjem | VWR | |

the addition of tetracycline (1 μg/ml) to a growth medium lacking hygromycin and blasticidin, 24 h prior to harvest.

Human cell lines stably expressing shRNAs for gene-specific knockdown (KD) were generated as previously described (Perocchi et al, 2010). The following shRNA constructs were used for MCU (sh-MCU, TRCN0000133861); EMRE (sh-EMRE, TRCN0000145067); MICU1 (sh-MICU1, TRCN0000053370); MICU2 (sh-MICU2 TRCN0000055848); MICU3 (sh1-MICU3, TRCN0000056083; sh2-MICU3, TRCN0000056084; sh3-MICU3, TRCN0000056085; sh4-MICU3, TRCN0000056086; sh5-MICU3, TRCN0000056087); EFHD1 (sh1-EFHD1, TRCN0000056183; sh2-EFHD1, TRCN0000056184; sh3-EFHD1, TRCN0000056185; sh4-EFHD1, TRCN0000056186; sh5-EFHD1, TRCN0000056187); and MCUB (sh-MCUB, TRCN0000128550). Infected HeLa cells were selected with 2 μg/mL puromycin. EFM19 and HCC1500 cells were infected with pLKO and sh5-EFHD1 viruses and selected with 1 μg/mL puromycin. The same number of cells for each condition (50,000 for EFM19; 25,000 for HCC1500) were seeded in a 3 cm culture plate and imaged 10 days after seeding, with media changes every 48 h. For siRNA-mediated KD, 50 nM of negative control scramble or targeting siRNAs (Scr: MISSION siRNA #1 SIC001; si1-EFHD1: SASI_Hs01_00228164; si2-EFHD1: SASI_Hs01_00228165, si-MCUB: 5'-AUACUACCAGUCACAC-CAU-3', si-PRELID1: 5'-CCCGAAUCCCUAUAGCAAA-3', si-MICU1: 5'-UCUGAAGGGAAAGCUGACAAU-3') were transfected using Lipofectamine RNAiMAX (Thermo Fisher Scientific, 13778075). Assays were normally carried out 48 h after transfection, unless noted otherwise. For the exogenous expression of MICU3 and EFHD1 ORFs in human cells, MICU3 (HsCD00296366) and EFHD1 (HsCD00719312) WT clones without a STOP codon in pDONR221 were purchased from DNASU, whereas MICU3 and EFHD1 EF-hand mutants were synthesized in the pUC57 vector (Thermo Fisher Scientific, SD0171) and cloned into a pDONR221 vector with the following primers: attB1-FW-MICU3, GGGGACAAGTTTGTACAAAAAAGCAGGCTTAGCC ACCATGGTGGCTCGAGGGCT; attB2-RV-MICU3, GGGGACC ACTTTGTACAAGAAAGCTGGGTTTCTGCTGTGAAGTTCTTT CTTCAGG; attB1-FW-EFHD1, GGGGACAAGTTTGTACAA AAAAGCAGGCTTAGCCACCATGGCCAGTGAGGAGCTG; att B2-RV-EFHD1, GGGGACCACTTTGTACAAGAAAGCTGGGT TTGTATTGAAGTTGGCCTTGAGTTT. Constructs were then cloned either into pcDNA-DEST40 Vector (Thermo Fisher Scientific, 12274015) or pLX304 for expression in mammalian cells in frame with a C-terminal V5-His$_6$ and V5 tag, respectively, by gateway cloning according to manufacturer instructions. Lentivirus production and infection were performed according to guidelines from the Broad RNAi Consortium and infected cell lines were selected 48 h post-transduction with the respective selection markers. Transient protein expression was performed using X-tremeGENE™ HP DNA transfection reagent following the manufacturer's instructions for a 1:3 DNA/reagent ratio. Assays were normally carried out 48 h after transfection, unless noted otherwise.

## Immunoblot analysis

To monitor endogenous and overexpressed proteins, cells or isolated mitochondria were lysated in RIPA buffer (150 mM NaCl, 50 mM Tris, 1 mM EGTA, 1% NP-40, 0.1% SDS); after 30 min of incubation on ice, the lysates were centrifuged at 15,000×*g* for 10 min to remove debris. 20 μg of total proteins were loaded, according to BCA quantification (Thermo Fisher Scientific, 23225) for each lane. Proteins were reduced with Laemmli buffer (Bio-Rad, 1610747) supplemented with 2.75 mM β-mercaptoethanol (Thermo Fisher Scientific, 21985-023) and denatured for 5 min at 90 °C, unless otherwise specified. To visualize MICU1-MICU2 or MICU1-MICU3 heterodimers, protein samples were denatured at 70 °C for 10 min in LDS Sample Buffer (Thermo Fisher Scientific, NP0007) with or without 100 mM Dithiothreitol (+DTT and -DTT, respectively), as previously performed (Patron et al, 2014). Proteins were separated by SDS-PAGE electrophoresis in 12% or 14% acrylamide gels, and transferred on nitrocellulose membranes (Amersham, 10600021) by semi-dry electrophoretic transfer (Bio-Rad). Accordingly with primary antibody datasheets, blots were blocked 1 h at room temperature (RT) with 5% non-fat dry milk (Carl Roth, T145.2) or 5% bovine serum albumin (BSA, Sigma-Aldrich, A7906) in TBS-Tween (0.5 M Tris, 1.5 M NaCl, 0.01% Tween) solution and incubated overnight at 4 °C with primary antibodies. Horseradish peroxidase-conjugated secondary antibodies (Bio-Rad, 1706515 or 1706516), diluted in TBS-Tween containing 0.5% BSA, were incubated for 1 h at RT followed by detection by chemiluminescence (Amersham, RPN2236). The expression level of specific proteins was detected by immunoblot analysis with the following antibodies: HA (BioLegend, MMS-101R), V5 (Thermo Fisher Scientific R96025), ACTIN (Sigma-Aldrich, A2228), MAIP1 (König et al, 2016), ATP5A (Abcam MS507), Lamin (Santa Cruz, sc-6217), VDAC (Abcam, ab14734), EFHD1 (Sigma-Aldrich HPA056959), HSP60 (R&D System MAB1800), SOD2 (Antibody Verify Inc. AAS29585C), TIM23 (BD bioscience, 611222), CyPD (Abcam, ab110324), GRP75 (Santa Cruz, sc-133137), MCU (Sigma-Aldrich, HPA016480), EMRE (Santa Cruz, sc-86337), MCUB (Santa Cruz, sc-163985), MICU1 (Sigma-Aldrich, HPA037479), MICU2 (Sigma-Aldrich, HPA045511), MICU3 (Sigma-Aldrich, HPA024048), TOM20 (Abcam, ab56783). Densitometry analysis of protein bands was performed with ImageJ by subtracting background signal and normalizing the area of each peak intensity to actin.

## Tandem affinity purification

Isogenic, parental Flp-In T-REx HEK293 cells stably expressing tagged MCU, MCUB, or EMRE as baits, as well as Flp-In T-REx HEK293 cells expressing tagged MCU with stable knockdown of either MICU1, MICU2, EMRE, or MCUB were expanded into two expression plates (245 × 245 mm, NUNC, 9407400) to obtain roughly 250 million cells. For TAP, each cell pellet was resuspended in 2 mL lysis buffer (50 mM HEPES-KOH pH 7.4, 150 mM KCL, 5 mM EGTA, 5% Glycerol, 3% Digitonin) and incubated at 4 °C in rotation for 30 min. Lysates were centrifuged at 10,000×$g$ for 5 min at 4 °C and the supernatants were incubated for 20 min at RT on a rolling shaker with 50 µL of a 500 mM HEPES solution at pH 8.0 containing avidin (10 µM final concentration). In the meantime, 150 µL of streptavidin resin was washed three times with 500 µL washing buffer (50 mM HEPES-KOH pH 7.4, 150 mM KCl, 5 mM EGTA, 5% Glycerol, 0.02% Digitonin), combined with the supernatant, and incubated 45 min at 4 °C in rotation. Resins were then washed three times, before eluting bound proteins with 250 µL of elution buffer (50 mM HEPES-KOH pH 8, 150 mM KCl, 5 mM EGTA, 5% glycerol, 10 mM Biotin), incubating on a rolling shaker at 900 rpm for 5 min at RT. The elution step was repeated 4 times and a total of 1 mL eluate was collected. Next, 50 µL of Ni-NTA resin was washed three times with 500 µL of washing buffer and incubated with the eluate for 45 min at 4 °C in rotation. The resin was then collected by quick spin and washed three times with 500 µL of washing buffer followed by two washing steps in washing buffer without detergent. Purified complexes were eluted through three steps of incubation with 35 µL of 1% RapiGest on a rolling shaker for 10 min at 900 rpm at RT. Eluates were pulled and subjected to acetone precipitation overnight. The next day, samples were centrifuged at 20,000×$g$ for 30 min at 4 °C, the supernatant was removed, and the pellet was stored at −80 °C for LC-MS/MS analysis.

## Sample preparation for LC-MS/MS analysis

Eluates were resuspended in 50 µL denaturation buffer (6 M urea, 2 M thiourea, 10 mM HEPES pH 8.0, 10 mM DTT) for 30 min at RT before adding alkylation agent (55 mM iodoacetamide) and incubating at RT in the dark for 20 min. Proteins were digested at RT for 3 h by adding LysC at 1:100 ratio of enzyme:protein, before diluting the sample with 50 mM ammonium bicarbonate to reach a urea concentration of 2 M. Subsequently, samples were digested at RT overnight by adding Trypsin at 1:100 ratio of enzyme:protein. The next day, the digestion was stopped by acidifying the sample with 10 µL of 10% trifluoroacetic acid and the final peptides were cleaned up using SDB-RPS StageTips as described (Kulak et al, 2014). MS analysis was performed using Q Exactive HF mass spectrometers (Thermo Fisher Scientific, Bremen, Germany) coupled online to a nanoflow ultra-high-performance liquid chromatography instrument (Easy1000 nLC, Thermo Fisher Scientific). Peptides were separated on a 50 cm long (75 µm inner diameter) column packed in-house with ReproSil-Pur C18-AQ 1.9-µm resin (Dr. Maisch GmbH, Ammerbuch, Germany). Column temperature was kept at 50 °C. Peptides were loaded with buffer A (0.1% (v/v) formic acid) and eluted with a nonlinear gradient of 5–60% buffer B (0.1% (v/v) formic acid, 80% (v/v) acetonitrile) at a flow rate of 250 nL/min. Peptide separation was achieved by 120 min gradients. The survey scans (300–1650 $m/z$, target value = 3E6, maximum ion injection times = 20 ms) were acquired at a resolution of 60,000 followed by higher-energy collisional dissociation (HCD) based fragmentation (normalized collision energy = 27) of up to 15 dynamically chosen most abundant precursor ions. The MS/MS scans were acquired at a resolution of 15,000 (target value = 1E5, maximum ion injection times = 60 ms). Repeated sequencing of peptides was minimized by excluding the selected peptide candidates for 20 s.

## MS data processing and visualization

Data analysis was carried out using the MaxQuant software package 1.5.5.2. The false discovery rate (FDR) cutoff was set to 1% for protein and peptide spectrum matches. Peptides were required to have a minimum length of 7 amino acids and a maximum mass of 4600 Da. MaxQuant was used to score fragmentation scans for identification based on a search with an initial allowed mass deviation of the precursor ion of a maximum of 4.5 ppm after time-dependent mass calibration. The allowed fragment mass deviation was 20 ppm. Fragmentation spectra were identified using the UniprotKB *Homo sapiens* database, combined with 245 common contaminants by the integrated Andromeda search engine. Enzyme specificity was set as C-terminus to arginine and lysine, also allowing cleavage before proline, and a maximum of two missed cleavages. Carbamidomethylation of cysteine was set as fixed modification and N-terminal protein acetylation as well as methionine oxidation as variable modifications. Both "label-free quantification (MaxLFQ)" with a minimum ratio count of 1 and "match between runs" with standard settings were enabled. Protein copy number estimates were calculated using the iBAQ algorithm, in which the sum of all tryptic peptides intensities for each protein is divided by the number of theoretically observable peptides. The mass spectrometric data have been deposited via PRIDE (Vizcaíno et al, 2013) to the ProteomeXchange Consortium under the accession number PXD040893.

Basic data handling, normalization, statistics, and annotation enrichment analysis was performed with the Perseus software package (1.6.15.0 release) (Tyanova et al, 2016), R (4.1.2 release) and GraphPad Prism (10.0 release). The label-free quantification (LFQ) module of the MaxQuant software (Cox and Mann, 2008; Cox et al, 2014) was used to define specific proteins that were quantitatively enriched with a given bait over all measured samples. Protein groups were filtered removing hits to the reverse decoy database and proteins only identified by modified peptides. Mitochondrial proteins were filtered using a curated list of 1276 proteins that integrates information from MitoCarta3.0 (Rath et al, 2021), Uniprot annotated mitochondria proteins, and IMPI (Smith and Robinson, 2019). It was required that each protein was quantified in at least two biological replicates from TAPs of each cell line to be considered for analysis. Protein LFQ intensities were log-transformed and missing values were imputed by values sampled from a normal distribution shifted 1.8 standard deviation and with a width 0.3 standard deviations from the distribution of all protein intensities within each sample as the background. Protein interactors were identified by volcano plot and protein correlation analyses. In each volcano plot, a quantitative bait-prey co-enrichment analysis was performed based on a two-tailed Welch's $t$ test using the multiple volcano (Hawaii) plot option of

Perseus (version 1.6.2.0) and a permutation-based FDR cutoff of 0.05 (Class A) and 0.10 (Class B) and S0 = 1 (Hein et al, 2015). However, proteins that do not cross any FDR threshold can still represent true positive interactors. Therefore, to be comprehensive, we used the correlation coefficients of interacting proteins as an additional qualifier to the FDR-controlled confidence of the volcano plot. In protein–protein interaction studies, true positive interacting proteins often exhibit a good correlation of their intensity profiles across different samples, as demonstrated in our previous studies (Hein et al, 2015; Michaelis et al, 2023). Protein intensity profiles across each pair of bait and control TAPs were used to calculate a Pearson's correlation coefficient between baits and preys (local correlation). A protein was defined as a specific interactor when having a correlation coefficient higher or equal than 0.6 based on the mean correlation of Class A interactors identified by volcano analysis (0.53) (Class C). Similarly, specific interactors were identified based on a Pearson's correlation analysis across control, MCU-, EMRE- and MCUB-TAPs (global correlation) using a permutation-based FDR of 0.05 (Class D). For the remodeling of MCUC interaction network upon genetic perturbation, Class A, B, and C parameters were used to define MCU interactors. For sh-MCUB a permutation-based FDR cutoff of 0.025 and 0.05 for class A and B, respectively, and a correlation coefficient higher or equal to 0.8 for class C, were used.

## Protein–protein interaction network

MCU, MCUB, and EMRE protein–protein interactions were visualized using Cytoscape 3.10.0 (Shannon et al, 2003). For each MCUC interactor, biological function, the number of transmembrane domains, submitochondrial localization, and gene-disease associations were retrieved based on MitoCop (Morgenstern et al, 2021), literature-based manual curation, DeepTMHMM (Hallgren et al, 2022), MitoCarta3.0 (Rath et al, 2021), and DisGeNET (Piñero et al, 2021). By uncharacterized proteins, we refer to proteins that lack single gene-based experimental evidence but whose function is inferred simply through large-scale analysis or sequence similarity. Protein expression level of MCUC interactors across 201 samples from 32 different tissue types of normal human individuals was obtained from GTEx (Jiang et al, 2020). The tissue numbers were reduced by removing some samples (Artery–Coronary, Esophagus–Mucosa, Minor Salivary Gland, Nerve–Tibial, Pituitary, and Skin) and by averaging tissues belonging to the same organ (arteries, colon segments, esophagus segments, and heart compartments), obtaining 21 tissues in total. Orthologs of MCUC interactors were identified across 120 eukaryotic species and one prokaryotic species (E. coli) used as outgroup and are common to CLIME (Li et al, 2014) and ProtPhylo (Cheng and Perocchi, 2015). Orthologs were defined using OBH (one-way best hit) as in (Cheng and Perocchi, 2015). The NCBI taxonomy database and the R package taxize were used to build the species tree. Hierarchical clustering was performed using the Euclidean distance with complete linkage. Significant interactors were analyzed using gProfiler's GOSt tool (Raudvere et al, 2019). Significance was established using the algorithm gSCS as a multiple-testing correction method with a significance threshold of 0.05. Only GO biological process driver terms significant in at least one condition were taken in consideration.

## Measurements of Ca²⁺ transients

Measurement of extracellular $Ca^{2+}$ clearance by mitochondria from digitonin-permeabilized HEK293 cells was performed using the membrane-impermeable $Ca^{2+}$ indicator Calcium-Green-5N (Life technologies, C3737). Cells were harvested at a density of 500,000 cells/mL in growth medium supplemented with 20 mM HEPES (pH 7.2/NaOH). Cells were collected by centrifugation at $300 \times g$ for 3 min at RT, resuspended in extracellular-like media (145 mM NaCl, 5 mM KCl, 1 mM $MgCl_2$, 10 mM Glucose, 10 mM HEPES, pH 7.4) containing 200 nM thapsigargin and incubated for 10 min under constant agitation at RT. Cells were then resuspended in intracellular-like buffer (140 mM KCl, 1 mM $KH_2PO_4/K_2HPO_4$, 1 mM $MgCl_2$, 20 mM HEPES, 100 µM EGTA, pH 7.2/KOH), supplemented with 1 mM $Na^+$-pyruvate, 1 mM of equimolar solution of $ATP/MgCl_2$ and 2 mM $Na^+$-succinate at a density of $2.5 \times 10^6$ cells/mL and the plasma membrane was permeabilized by incubation with 60 µM digitonin for 5 min at RT under constant agitation. 100 µL of the cell suspension was seeded into a black 96-well plate (PerkinElmer) and Calcium-Green-5N fluorescence (excitation 506 nm, emission 531 nm) was monitored every 2 s at RT using a CLARIOstar microplate reader (BMG Labtech) after injection of 40 µM $CaCl_2$. The MCU inhibitor Ru360 (10 µM) was used as a positive control.

Measurements of $Ca^{2+}$ transients in HeLa cells were performed using the luminescence $Ca^{2+}$ indicator aequorin as previously described (Arduino et al, 2017). Briefly, HeLa mt-AEQ cells infected with lentivirus carrying the specific shRNA were seeded in white 96-well plates (PerkinElmer, 6005181) at 25,000 cells/well. After 24 h, aequorin was reconstituted with 2 µM native coelenterazine for 1–2 h at 37 °C. For measurements of $Ca^{2+}$ kinetics upon siRNA-mediated EFHD1 knockdown, cells were transfected using a transfection mix including Lipofectamine RNAiMAX transfection reagent (Thermo Fisher Scientific, 13778075), 0.5 µg of mt-AEQ (Rizzuto et al, 1992) or cytosolic aequorin (Brini et al, 1995) cDNAs, and either a final concentration of 50 nM siRNA. Mt-AEQ-based measurements of $Ca^{2+}$-dependent light kinetics were performed upon 100 µM histamine stimulation. Light emission was measured either using the luminescence counter MicroBeta2 LumiJET Microplate Counter (PerkinElmer) or the PerkinElmer Envision plate reader at 469 nm every 0.1 and 1 s, respectively. For measurements of $Ca^{2+}$ kinetics upon EFHD1-WT or EFHD1$_{(EF1+EF2)}$ overexpression, cells were transfected using TransIT-2020 Transfection Reagent (Mirus Bio) with 1 ug of EFHD1 cDNA and 0.5 µg of mt-AEQ or cytosolic aequorin. $Ca^{2+}$ kinetics upon store-operated $Ca^{2+}$ entry (SOCE) were measured upon pre-treatment with the irreversible SERCA inhibitor thapsigargin (100 nM) for 10 min in $Ca^{2+}$-free modified Krebs-Ringer Buffer (135 mM NaCl, 5 mM KCl, 1 mM $MgCl_2$, 0.4 mM $KH_2PO_4$, 1 mM $MgSO_4$, 20 mM HEPES, 600 µM EGTA, 10 mM glucose, pH 7.4 at 37 °C) followed by perfusion with the same medium without thapsigargin but supplemented with 1.5 mM $CaCl_2$. For the EFHD1 rescue expression experiment, HeLa cells were transiently transfected using a calcium phosphate transfection protocol. Briefly, for each well of a 6-well plate, a total of 8 µg of plasmid DNA (2 µg of mt-AEQ and 6 µg split between the other constructs) was diluted in 90 µL of water and 10 µL of 2.5 M $CaCl_2$ and mixed with an equal volume of 2xHBS (50 mM HEPES, 280 mM NaCl, 1.5 mM $Na_2HPO_4$, pH 7.06) to induce the formation of $Ca^{2+}$ phosphate-DNA precipitates. The prepared transfection mix was applied to the cells for 8 h. Following this incubation, cells were washed

three times with PBS to remove any residual precipitates. Twenty-four hours before the mitochondrial $Ca^{2+}$ measurement, the transfected HeLa cells were trypsinized, and 25,000 cells were seeded into each well of a 96-well plate. The measurement of mt-$Ca^{2+}$ uptake in digitonin-permeabilized mt-AEQ HeLa cells was performed as previously described (Wettmarshausen et al, 2018). Briefly, HeLa cells stably expressing mt-AEQ were harvested at a density of 500,000 cells/mL in growth medium supplemented with 20 mM HEPES (pH 7.4/NaOH) and the aequorin was reconstituted by incubation with 2 μM native coelenterazine $n$ (Biotium, BOT-10115-1) for 2.5 h at RT. Cells were then centrifuged at 300×$g$ for 3 min and the pellet was resuspended in an extracellular-like buffer (145 mM NaCl, 5 mM KCl, 1 mM $MgCl_2$, 10 mM Glucose, 10 mM HEPES, pH 7.4) containing 200 nM thapsigargin and incubated for 20 min under constant agitation at RT. The cells were collected by centrifugation at 300×$g$ for 3 min and the pellet was resuspended in an intracellular-like buffer supplemented with 1 mM $Na^+$-pyruvate, 1 mM of equimolar solution of ATP/$MgCl_2$ and 2 mM $Na^+$-succinate. The cells were permeabilized for 5 min with 60 μM digitonin, collected by centrifugation for 3 min at 300×$g$ and resuspended in intracellular-like buffer at a density of 900 cells/μL. Then, 90 μL of cell suspension was dispensed into a white 96-well plate (PerkinElmer). Cells were incubated for 5 min at RT and light signal was recorded at 469 nm every 0.1 s using a luminescence counter (MicroBeta2 LumiJET Microplate Counter, PerkinElmer) after injection of $CaCl_2$ to achieve a free $Ca^{2+}$ concentration of 5 μM in solution. Ru360 (10 μM) was used as a positive control. All the luminescence signals were converted in [$Ca^{2+}$] values according to the $Ca^{2+}$ response curve of aequorin, as previously performed (Brini et al, 1999).

For single-cell measurement of $Ca^{2+}$ transient, HeLa cells were treated with siRNA for 48 h before the experiment. Cells were transiently transfected with mitochondrial matrix-targeted RCaMP to measure changes in [$Ca^{2+}$]$_{mt}$ then plated on Poly-D-Lysin coated 25 mm coverslips (Thermo Fisher Scientific, 25CIR-1.5). To measure changes in the [$Ca^{2+}$]$_{cyt}$ cells were loaded with 2 μM Fura2AM (Moleculer probes, F-1221) in 2% BSA containing extracellular medium (ECM, 121 mM NaCl, 5 mM $NaHCO_3$, 10 mM Na-HEPES, 4.7 mM KCl, 1.2 mM $KH_2PO_4$, 1.2 mM $MgSO_4$, 2 mM $CaCl_2$, and 10 mM glucose, pH 7.4) in the presence of 0.003% Pluronic F-127 (Thermo Fisher Scientific, P6867) and 150 μM sulfinpyrazone (Sigma-Aldrich, S9509) for 15 min at 35 °C. After dye-loading, cells were washed with fresh 0.25% BSA containing ECM with or without $Ca^{2+}$ and transferred to the temperature-controlled stage (37 °C) of an Olympus IX81 motorized inverted epifluorescence microscope fitted with a Hamamatsu ORCA-Flash 4.0v3 sCMOS camera, high-speed excitation switching by Sutter Lambda 421 LED illuminator and UV-optimized Olympus UAPO/340 ×40/1.35NA oil immersion objective. For simultaneous measurements of [$Ca^{2+}$]$_{cyt}$ and [$Ca^{2+}$]$_{mt}$ Fura-2 fluorescence was recorded at 340 and 380 nm, and RCaMP at 577 nm excitations, using dual-band Chroma 59022bs dichroic and 59022 m emission filter. Image triplets were acquired every 0.33 s.

## Animals

C57BL/6n WT mice or C57BL/6n MCUB KO mice (Feno et al, 2021) were housed in a pathogen-free, temperature- and humidity-controlled animal facility on a 12:12 h light-dark cycle. Diet consisted of standard laboratory chow and double-distilled water. All animal procedures were in accordance with the European Community Council Directive for the Care and Use of Laboratory Animals (86/609/ECC) and German Law for Protection of Animals and were approved by the local authorities. All experiments were performed with female mice that were at least 3 months old.

## Isolation of functional mitochondria from cultured cells and mouse tissues

Mitochondria were isolated by nitrogen cavitation as previously described (Wettmarshausen and Perocchi, 2017). Briefly, HeLa and HEK293 cells were grown to confluency in 600 cm² cell culture plates. Culture medium was removed, and cells were rinsed with 30 mL PBS, scraped down and resuspended in 5 mL PBS. After 5 min of centrifugation at 600×$g$ at 4 °C, the cell pellet was resuspended in ice-cold isolation buffer (IB; 220 mM mannitol, 70 mM sucrose, 5 mM HEPES-KOH pH 7.4, 1 mM EGTA-KOH, pH 7.4), supplemented with protease inhibitor (Sigma-Aldrich, 5056489001). Cell suspension was immediately subjected to nitrogen cavitation at 800 psi for 10 min at 4 °C. Nuclei and intact cells were pelleted by centrifugation at 600×$g$ for 10 min at 4 °C. Supernatants were transferred into new tubes and centrifuged at 8000×$g$ for 10 min at 4 °C. The resulting pellet containing crude mitochondria was resuspended in 50-200 μL IB for further analyses. For mouse brain mitochondria isolation, the tissue was homogenized with two strokes at 300 rpm using a loose-fitting Teflon homogenizer followed by nitrogen cavitation at 800 psi for 10 min in ice-cold isolation buffer supplemented with 0.5% fatty acid-free BSA (Sigma-Aldrich, A7030) and protease inhibitor (Sigma-Aldrich, 5056489001). Nuclei and intact cells were pelleted by centrifugation at 600×$g$ for 10 min at 4 °C. Supernatants were transferred into new tubes and centrifuged at 12,000×$g$ for 10 min at 4 °C. The centrifugation step was repeated, changing the buffer with IB without BSA. The final pellet was resuspended in IB without BSA and stored on ice for further use. During the isolation, whole cell lysate (WCL), as well as cytosolic (C), nuclear (N), and crude mitochondrial (M) fractions, were collected for immunoblot analysis.

## Topology analysis of mitochondrial proteins

Alkaline carbonate extractions at pH 10, pH 11, or pH 12 from crude mitochondria were performed as previously described (Wettmarshausen et al, 2018) to analyze membrane association and submitochondrial localization of proteins. Briefly, 100 μg of mitochondria were centrifuged at 8000×$g$ for 10 min at 4 °C and then resuspended in 0.1 M $Na_2CO_3$ at pH of 10, 11, or 12 and incubated for 30 min on ice. Afterward, the samples were centrifuged at 45,000×$g$ for 10 min at 4 °C. Pellets resulting from this process were resuspended in 100 μL of 2× Laemmli buffer, boiled at 98 °C for 5 min, and stored at −80 °C for subsequent use (referred to as the membrane sample). Supernatants were combined with 40 μL of 100% trichloroacetic acid and left to incubate overnight at −20 °C. The next day, the supernatants were centrifuged at 16,000×$g$ for 25 min at 4 °C. Pellets were washed twice with cold acetone, air-dried for 20-30 min at RT, resuspended in 100 μL of 2× Laemmli buffer, and heated to 98 °C for 5 min (referred to as the soluble sample). SDS-PAGE analysis was performed on 25 μL of both the soluble and membrane samples. Antibodies against MICU1 (soluble, membrane-associated), ATP5a

(soluble, membrane-associated), MCU (integral transmembrane protein) were used as positive controls. To determine the submitochondrial localization of MICU3 and EFHD1, 30 µg of crude mitochondria were exposed to increasing concentrations of digitonin or 1% Triton X-100, which sequentially permeabilize the outer and inner membranes. This was conducted in the presence of 5 mM membrane-impermeable maleimide functionalized polyethylene glycol (mPEG, Sigma-Aldrich, 63187). This compound attaches a polyethylene glycol polymer chain of approximately 5 kDa to free thiol groups of proteins. The reaction was carried out at RT for 30 min and was quenched with 100 µM cysteine on ice for 10 min. Immunoblot analysis was performed on the samples, with the intermembrane space protein MICU1, and the matrix soluble proteins HSP60 and SOD2, serving as controls. The proteinase K protection assay was performed by incubating 30 µg of mitochondria in 30 µL of isolation buffer in the presence of 100 µg/mL proteinase K with increasing concentrations of digitonin or 1% Triton X-100 to sequentially permeabilize outer and inner membranes. The reaction was carried out at RT for 15 min and was stopped by the addition of 5 mM phenylmethylsulfonyl fluoride (PMSF), followed by incubation on ice for 10 min. Immunoblot analysis was used to examine the samples, with TOM20 and TIM23 (integral outer and inner membrane proteins, respectively), along with cyclophilin D (CyPD) and HSP60 (soluble matrix proteins) as controls.

## Blue native page analysis of mitochondrial protein complexes

Blue-Native (BN) PAGE analysis was performed as described by Witting et al (Wittig et al, 2006) with some adaptations. Briefly, equal amounts of mitochondria (10 µg per lane, unless noted otherwise) were diluted at least 1:100 in ice-cold miliQ water with proteinase inhibitors (Sigma-Aldrich, 5056489001), the pellet was then collected at 20,000×$g$ for 10 min at 4 °C, resuspended in ice-cold 1× BN Sample Buffer (Thermo Fisher Scientific, BN2003) with 1% (w/v) digitonin and incubated on ice for 15 min. Afterward, the sample was centrifuged at 20,000×$g$ at 4 °C for 30 min and the supernatant was transferred into a new pre-chilled tube with 0.25% G-250 (Thermo Fisher Scientific, BN2004). BN-PAGE was performed at 4 °C on NativePAGE Novex 3-12% Bis-Tris Protein gels (Thermo Fisher Scientific, BN1001) according to manufacturer's instructions, followed by overnight wet blot transfer at 30 V and 4 °C onto a 0.2 µM pore size PVDF membrane (Amersham, GE10600021). Immunoblotting was performed according to the standard protocol. Second dimension (2D) analysis was performed as described by Na Ayutthaya et al (Na Ayutthaya et al, 2020) with some modifications. Following the above described first dimension (1D) BN-PAGE, sample lanes for 2D were excised, incubated in 1× SDS sample buffer (Bio-Rad, 1610747) for 10 min and boiled shortly, followed by incubation in hot 1× SDS sample buffer for 15 min. As a control, one well was loaded with 5 µg of input mitochondria previously diluted in 5 µL of 2× LDS dye (Thermo Fisher Scientific, 84788) with 2.75 mM β-mercaptoethanol (Thermo Fisher Scientific, 21985-023) and RIPA buffer (total volume 10 µL) and boiled for 5 min at 95 °C. The 2D well of the pre-cast NuPAGE 4%-12% Bis-tris 2D well gel (Thermo Fisher Scientific, NP0326BOX) was washed and filled with 1× MOPS running buffer (Thermo Fisher Scientific, NP0001). Lanes were then fitted onto the 2D and overlaid with 1× SDS sample buffer, followed by electrophoresis according to the

manufacturer's instructions. The proteins were then transferred by wet blot transfer on 0.2 µM pore size PVDF membrane (Amersham, GE10600021) and immunoblotted using MCU (Sigma-Aldrich, HPA016480) according to the standard protocol.

## RNA extraction and quantitative real-time PCR

RNA was isolated with the RNeasy Mini kit (Qiagen, 74104), according to manufacturer instructions. An equal amount of RNA from each sample was used to generate complementary DNA (cDNA) with the SuperScript™ III First-Strand Synthesis SuperMix kit (Thermo Fisher Scientific, 18080400). The resulting cDNA was diluted 1:8 in nuclease-free water and analyzed by real-time PCR using the following TaqMan assays: Hs00368816 (EFHD1) and Hs01003267 (HPRT1, used as control).

## Mitochondrial bioenergetics

HeLa sh5-EFHD1 and pLKO mt-AEQ cells were seeded at a density of 20000 cells/well in a Seahorse XFe96 Cell Culture Microplate 24 h before the experiment in DMEM with 10% FBS. On the day of the experiment, medium was removed, and cells were washed twice with 200 µL of the respective assay medium for the mito-stress test (Agilent Base Medium supplemented with 1 mM pyruvate, 2 mM L-glutamine, 10 mM glucose, pH 7.4) or glycolysis (Agilent Base Medium supplemented with 2 mM L-glutamine, pH 7.4). Next, 180 µL of the assay medium for mito-stress test or glycolysis were added to each well and the plate was incubated for 60 min in a non-$CO_2$ incubator at 37 °C. For the mito-stress test, 10× port solution of the following compounds were injected to reach the specified final concentration: oligomycin (1.5 µM), carbonyl cyanide m-chlorophenyl hydrazone (CCCP; 1.5 µM), and antimycin A/rotenone (4 µM/2 µM). For the glycolysis assay, glucose (10 mM), oligomycin (1.5 µM), and 2-DG (100 mM). The CyQUANT® Cell Proliferation Assay Kit (Thermo Fisher Scientific, C7026) was used to normalize differences in cell density between wells. Briefly, growth media was removed, and the plate was frozen at −80 °C. Cells were thawed and lysed with 200 µL of dye/lysis buffer. After 5 min, fluorescence was measured at 480ex/520em using a CLARIOstar plate reader (BMG Labtech). The raw fluorescence of each well was divided by the plate average and the resulting factor was used to normalize the Seahorse data.

## Mitochondrial membrane potential measurement

Mitochondrial membrane potential was measured as previously described (Vecellio Reane et al, 2016). Briefly, HeLa cells were incubated with 20 nM tetramethyl rhodamine methyl ester (TMRM, Thermo Fisher Scientific) for 30 min at 37 °C in Krebs-Ringer modified buffer (135 mM NaCl, 5 mM KCl, 1 mM $MgCl_2$, 0.4 mM $KH_2PO_4$, 1 mM $MgSO_4$, 20 mM HEPES, 10 mM glucose, 1 mM $CaCl_2$, pH 7.4 at 37 °C). Images were taken on an inverted microscope (Zeiss Observer.Z1) equipped with an Apochromat ×40/1.1 NA water immersion objective, an AxioCam HRm, and a Zeiss HXP 120 C Fluorescence Light Source. TMRM excitation and emission were performed through the Zeiss filter set 43 HE (BP550/25 FT570 BP605/70). Images were taken every 5 s with a fixed 100 milliseconds (msec) of exposure time. In all conditions, 10 µM

CCCP (carbonyl cyanide p-trichloro-methoxyphenylhydrazone), was added after ten acquisitions to completely collapse the electrical gradient across the IMM. After background correction, the fluorescence values after CCCP addition (i.e., TMRM fluorescence not due to membrane potential) was subtracted for each cell.

## Basal [Ca$^{2+}$]$_{mt}$ measurements

For the measurement of [Ca$^{2+}$]$_{mt}$ in resting conditions, sh5-EFHD1 HeLa cells were transfected with 4mtGCaMP6f. 48 h later, coverslips were transferred to an imaging chamber and incubated in 1 mL of modified KRB buffer supplemented with 5.5 M glucose and 1 mM CaCl$_2$. Imaging was performed on an inverted microscope (Zeiss Axiovert 200), equipped with a 40×/1.3 N.A. PlanFluor objective. Excitation was performed with a Deltaram V high-speed mono-chromator (Photon Technology International) equipped with a 75 W Xenon Arc lamp. Images were captured using an Evolve 512 Delta EMCCD (Photometrics), and the system was controlled by Meta-Morph 7.5 (Molecular Devices), assembled by Crisel Instruments. As previously performed (Butera et al, 2021), we used the fluorescence measurement at the Ca$^{2+}$-independent isosbestic point at 410 nm to normalize the differences in the expression of the probes and in the specimen focus plane. For this reason, the cells were alternatively illuminated every second at 485 and 410 nm, and fluorescence was collected through a 525/50 filter (Chroma) mounted on an OptoSpin25 wheel (Cairn research). The exposure time was 100 msec. Analysis was performed with the Fiji distribution of ImageJ. Data are presented as fluorescence ratio (485/410 nm) after frame-by-frame background correction.

## Cell viability

In total, 12,500 cells/well from HeLa pLKO and HeLa sh5-EFHD1 were seeded into 96-well plates. Resazurin sodium salt (Santa Cruz Biotechnology, sc-206037) was used to assess cell viability. For untreated cells, culture media was changed 48 h after seeding to 0.004% resazurin in DMEM 10% FBS medium and incubated for 4 h at 37 °C after which the fluorescence was measured at 540ex/590em using a CLARIOstar Plate Reader (BMG Labtech). For treatments, 24 h after seeding, both cell lines were treated with 0.5 mM H$_2$O$_2$ (Sigma-Aldrich, 516813), 500 nM thapsigargin (Sigma-Aldrich, 586005), 40 µM C2-Ceramide (Santa Cruz Biotechnology, sc-201375) or 50 nM Paclitaxel (Abcam, ab120143-10mg), and 48 h after treatment cell viability was assessed as described above.

## Cancer data sources

RNA-Seq and proteomics data were obtained from the DepMap portal (https://depmap.org/), RNA-Seq gene expression TPM values from DepMap Public 23Q2 release (Ghandi et al, 2019), and proteomics from the Nusinow et al, 2020 data release (Nusinow et al, 2020). Primary breast tumor and adjacent normal tissue gene expression pair data was analyzed with TNMplot.com (Bartha and Győrffy, 2021) using the RNA-Seq data as source. Kaplan–Meier plot to assess overall breast cancer patient survival was generated using GEPIA2 (Tang et al, 2019), selecting the BRCA dataset for analysis and setting the median expression value as cutoff for determining the high and low expressing EFHD1 cohorts. Gene

expression data for chemotherapy responders and non-responders was analyzed from rocplot.org (Fekete and Győrffy, 2019) selecting the pathological complete response cohort, any chemotherapy, and JetSet only data as parameters.

## Cell migration assay (Boyden chamber assay)

Migration assays were performed using 24-well plates with uncoated polycarbonate membrane inserts (BD Biosciences, 353097). In total, 2 × 10$^5$ cells were allowed to migrate for 24 h. Membranes were fixed with Methanol (Merck, 106009) stained by Hematoxylin solution (Sigma, 51275) and mounted on glass slides with Pertex mounting medium (#41-4010-00, MEDITE GmbH). Images were taken with an Olympus BX43 microscope.

# Data availability

The mass spectrometric data have been deposited via PRIDE to the ProteomeXchange Consortium under the accession number PXD040893.

The source data of this paper are collected in the following database record: biostudies:S-SCDT-10_1038-S44318-024-00219-w.

# Peer review information

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

## Acknowledgements

The authors thank Tullio Pozzan for his substantial contribution to this study, both in the design of the experiments and interpretation of the results, until his passing in October 2022. The authors thank Marcus Conrad, Toshitaka Nakamura, Suresh Joseph and David Weaver for critical reading of the paper and helpful discussions; Tim König and Daniela M. Arduino for experimental advice. FP, HCD, SW, NPD, MF, YC, and MC were supported by the Munich Center for Systems Neurology (SyNergy EXC 2145; Project ID 390857198) and the ExNet-0041-Phase2-3 ('SyNergy-HMGU') through the Initiative and Network Fund of the Helmholtz Association. DVR was supported by the European Union funding program Horizon Europe (HORIZON-MSCA-2021-PF Project ID 101065790). JW, AL, and MG were supported by the German Research Foundation (DFG) under the Emmy Noether Programme (PE 2053/1-1) and the Bavarian Ministry of Sciences, Research and the Arts in the framework of the Bavarian Molecular Biosystems Research Network (D2–F5121.2–10c/4822) MK and GH were supported by an NIH grant (RO1-HL142271) to GH The authors acknowledge Euro-BioImaging (www.eurobioimaging.eu) for providing access to imaging technologies and services via the ALM Node (Padua, Italy).

## Author contributions

**Hilda Delgado de la Herran**: Validation; Investigation; Visualization; Methodology; Writing—original draft; Writing—review and editing. **Denis Vecellio Reane**: Validation; Investigation; Methodology; Writing—original draft; Writing—review and editing. **Yiming Cheng**: Data curation; Formal analysis; Visualization. **Máté Katona**: Investigation. **Fabian Hosp**: Formal analysis; Investigation. **Elisa Greotti**: Investigation. **Jennifer Wettmarshausen**: Investigation; Visualization. **Maria Patron**: Investigation. **Hermine Mohr**: Investigation. **Natalia Prudente de Mello**: Investigation. **Margarita Chudenkova**: Investigation. **Matteo Gorza**: Investigation. **Safal Walia**: Formal analysis; Validation. **Michael Sheng-Fu Feng**: Investigation. **Anja Leimpek**: Investigation. **Dirk Mielenz**: Resources. **Natalia S Pellegata**: Resources. **Thomas Langer**: Resources. **György Hajnóczky**: Resources. **Matthias Mann**: Resources. **Marta Murgia**: Conceptualization; Formal analysis; Supervision; Investigation; Methodology. **Fabiana Perocchi**: Conceptualization; Supervision; Funding acquisition; Methodology; Writing—original draft; Writing—review and editing.

Source data underlying figure panels in this paper may have individual authorship assigned. Where available, figure panel/source data authorship is listed in the following database record: biostudies:S-SCDT-10_1038-S44318-024-00219-w.

## Funding

## Disclosure and competing interests statement

The authors declare no competing interests.

# Expanded View Figures

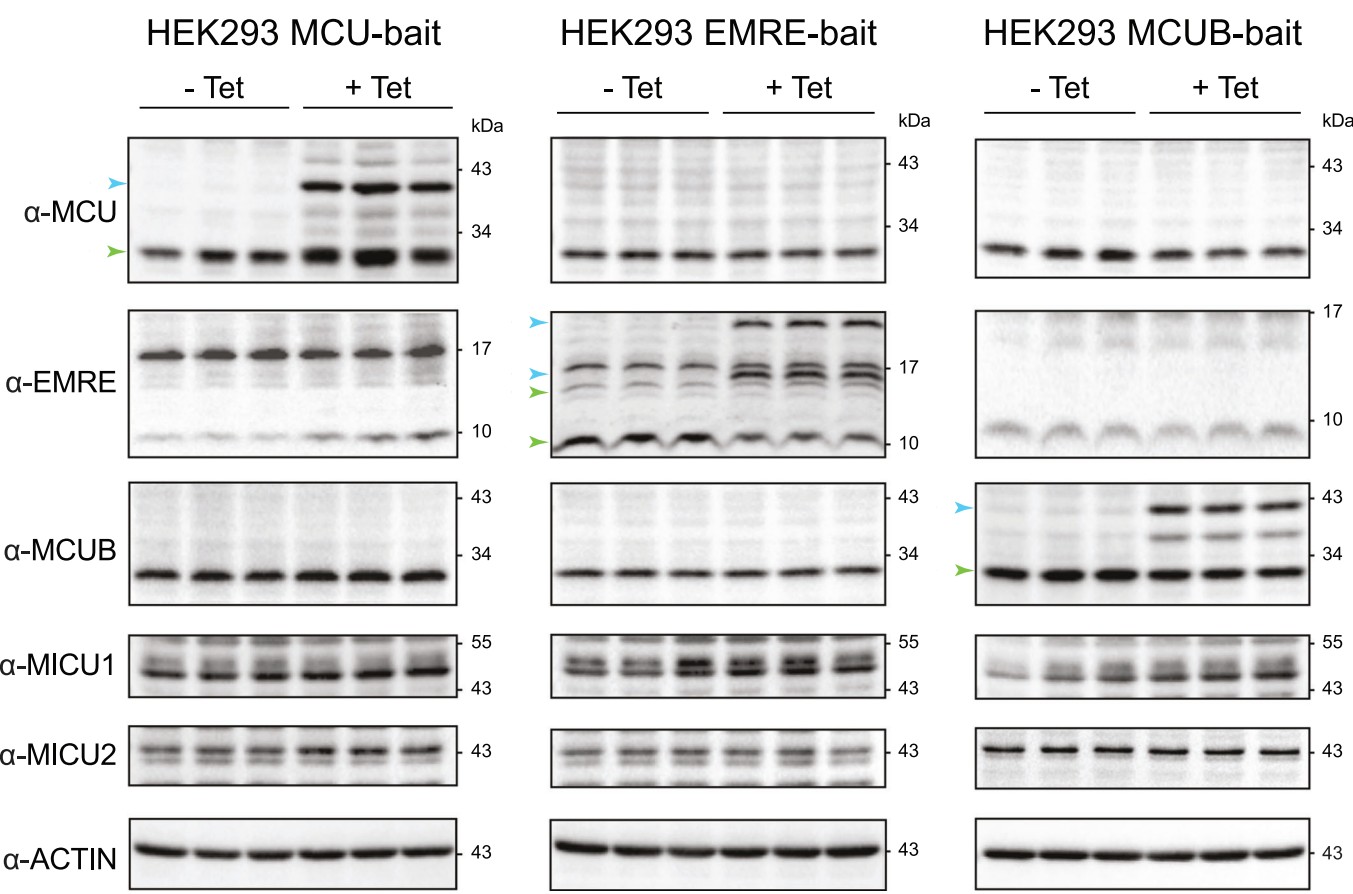

**Figure EV1. Expression of MCUC components before and after bait induction.**

Immunoblot analysis of MCU, EMRE, MCUB, MICU1, MICU2 and ACTIN (loading control) in whole cell lysates from Flp-In T-REx HEK293 cell lines before (-Tet) and after ( + Tet) tetracycline-driven expression of each bait. Immunoblots of MCU from MCU-bait cells, EMRE from EMRE-bait cells and MCUB from MCUB-bait cells were re-used from Fig. 2A. Refer to quantification in Fig. 2A.

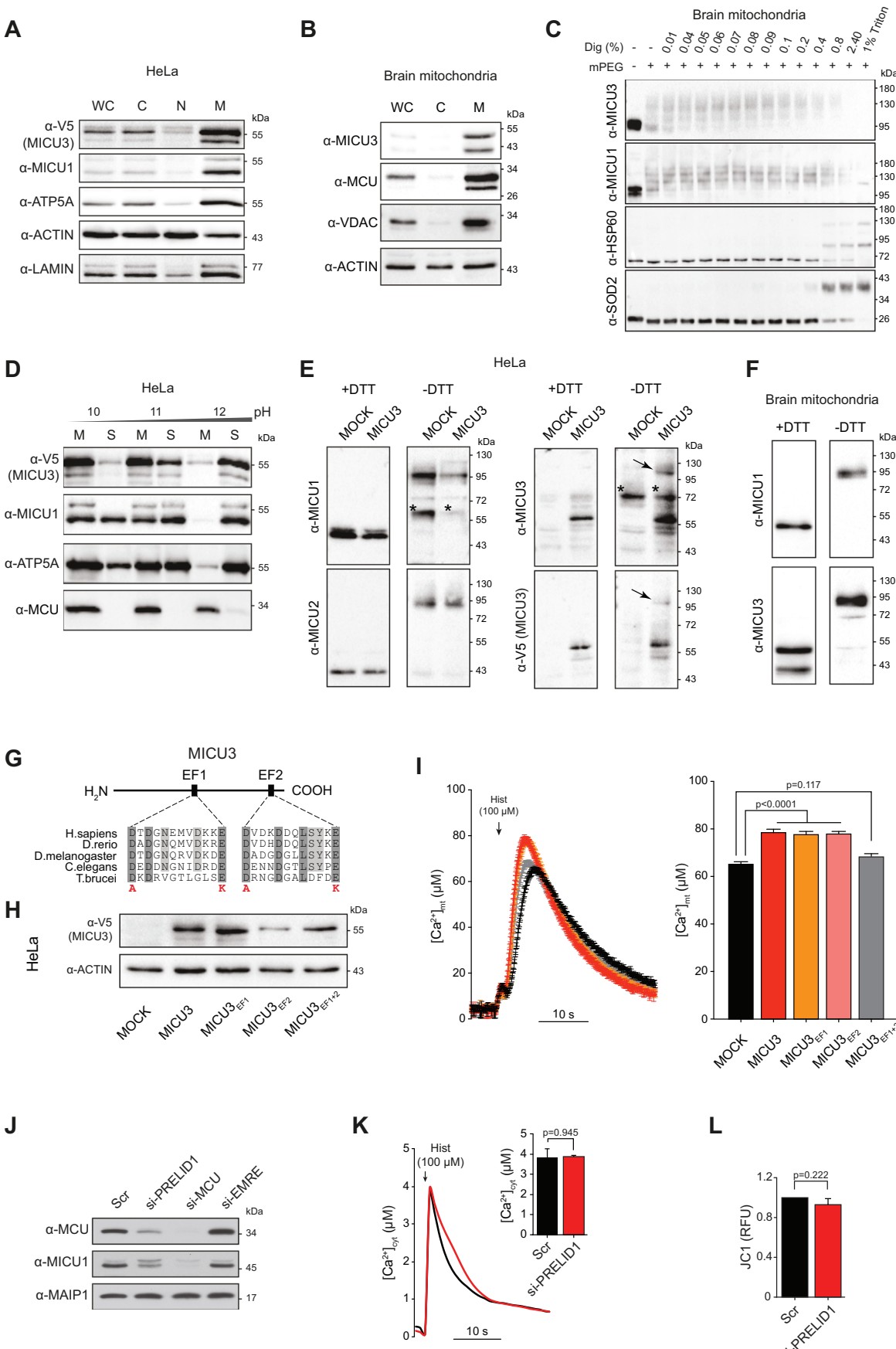

◀ **Figure EV2.   MICU3 positively regulates MCU-dependent mitochondrial Ca²⁺ uptake and PRELID1 is required for MCUC stability.**

(A, B) MICU3 is enriched in mitochondria from (A) HeLa cells overexpressing MICU3-V5 and (B) mouse brain. WC whole cell lysate, C cytosol, N nuclei, M mitochondria. MICU1, ATP5A, MCU, and VDAC are used as markers of mitochondrial proteins, while ACTIN and LAMIN as markers of cytosolic and nuclear proteins, respectively. (C) Immunoblot analysis of mitochondria isolated from mouse brain in the presence of increasing concentrations of the membrane-impermeable sulfhydryl group reactive PEG derivative (mPEG, maleimide functionalized polyethylene glycol). MICU1 is used as positive control for IMS proteins, whereas HSP60 and SOD2 for mitochondrial matrix proteins. (D) Immunoblot analysis of mitochondrial soluble (S) and membrane (M) fractions isolated from HeLa cells overexpressing MICU3-V5 by alkaline carbonate extraction at pH 10, pH 11, and pH 12. MICU1 and ATP5A (soluble and membrane-associated proteins, respectively), and MCU (integral transmembrane protein) are used as positive controls. (E, F) MICU3 and MICU1 dimerize through a disulfide bond in mitochondria of (E) HeLa cells overexpressing MICU3-V5 compared to control (MOCK) and of (F) mouse brain. Immunoblot analysis was performed in both reducing ($+$DTT) and non-reducing (-DTT) conditions. *Indicates non-specific bands. MICU1/MICU3 dimers are indicated by an arrow. (G) Domain structure of MICU3. EF1 and EF2 refer to two evolutionarily conserved EF-hand domains. Amino acid substitution used to generate MICU3 EF-hand mutants are indicated in red (EF1$_{mut}$, D245A and E256K; EF2$_{mut}$, D483A and E494K). (H) Immunoblot analysis of exogenous MICU3 detected with an anti-V5 antibody and ACTIN (loading control) in whole cell lysates from HeLa mt-AEQ cells expressing either WT MICU3 (MICU3) or MICU3 mutants in the first (MICU3$_{EF1}$), the second (MICU3$_{EF2}$) or both (MICU3$_{EF1+2}$) EF-hands fused to a C-terminal V5 tag and compared to untransfected control cells (MOCK). (I) Average traces and quantification of [Ca²⁺]$_{mt}$ transients in HeLa mt-AEQ cells expressing either WT or MICU3 mutants in response to histamine (Hist) and compared to control (MOCK). Data represent mean ± SEM ($n = 4$ biological replicates); one-way ANOVA with Dunnett's multiple comparison test. (J) Immunoblot analysis of MCU, MICU1 and MAIP1 (loading control) in whole cell lysate from HeLa cells transfected with si-PRELID1 and compared to negative control (Scr). si-MCU and si-EMRE are used as positive controls for MCUC expression and stability. (K), Representative traces and quantification of [Ca²⁺]$_{cyt}$ transients upon histamine (Hist) stimulation in si-PRELID1 and Scr HeLa cells expressing cytosolic aequorin (mean ± SEM; $n = 3$ biological replicates); Student's $t$ test. (L) JC1-based quantification of mitochondrial membrane potential upon PRELID1 knockdown in HeLa cells (RFU, relative fluorescence unit), (mean ± SEM; $n = 3$ biological replicates); Student's $t$ test. Refer to Fig. 4.

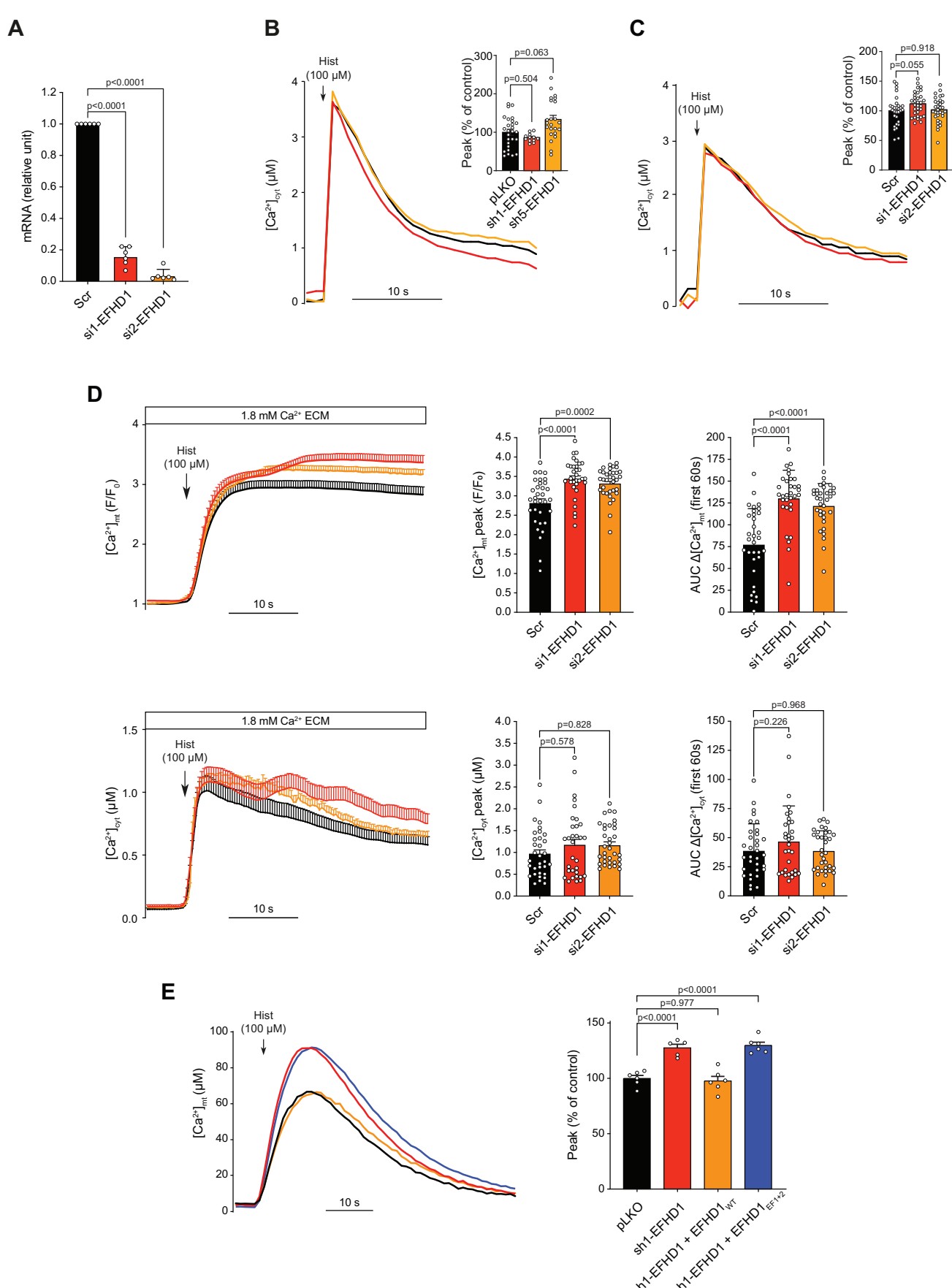

◀ **Figure EV3. EFHD1 inhibits MCU-dependent uptake of Ca$^{2+}$ in mitochondria without affecting [Ca$^{2+}$]$_{cyt}$ transients.**

(A) Quantification of EFHD1 KD by real-time PCR (mean ± SEM; $n = 6$ biological replicates); one-way ANOVA with Dunnett's multiple comparisons test. (B, C) Representative traces and quantification of [Ca$^{2+}$]$_{cyt}$ transients upon histamine (Hist) stimulation in (B) sh-EFHD1 and (C) si-EFHD HeLa cells expressing cytosolic aequorin (mean ± SEM; $n \geq 12$ biological replicates); one-way ANOVA with Dunnett's multiple comparisons test. (D) Quantification of [Ca$^{2+}$]$_{mt}$ (upper panel) and [Ca$^{2+}$]$_{cyt}$ (lower panel) responses in control (Scr) and si-EFHD1 treated HeLa cells upon histamine-induced ER Ca$^{2+}$ release in presence of 1.8 mM Ca$^{2+}$ in the extracellular medium (ECM). Peak and area under the curve (AUC) are calculated for the first 60 s of histamine (Hist) stimulation (mean ± SEM from 3 independent experiments ($n \geq 30$ cells from 2 independent experiments); Student's $t$ test. (E) Representative traces and quantification of [Ca$^{2+}$]$_{mt}$ transients upon histamine (Hist) stimulation in HeLa cells either expressing EFHD1-targeting shRNA alone or with EFHD1 expression rescue using the EFHD1$_{WT}$ and EFHD1$_{EF1+2}$ constructs (mean ± SEM; $n \geq 5$ biological replicates); one-way ANOVA with Dunnett's multiple comparisons test. Refer to Fig. 4.

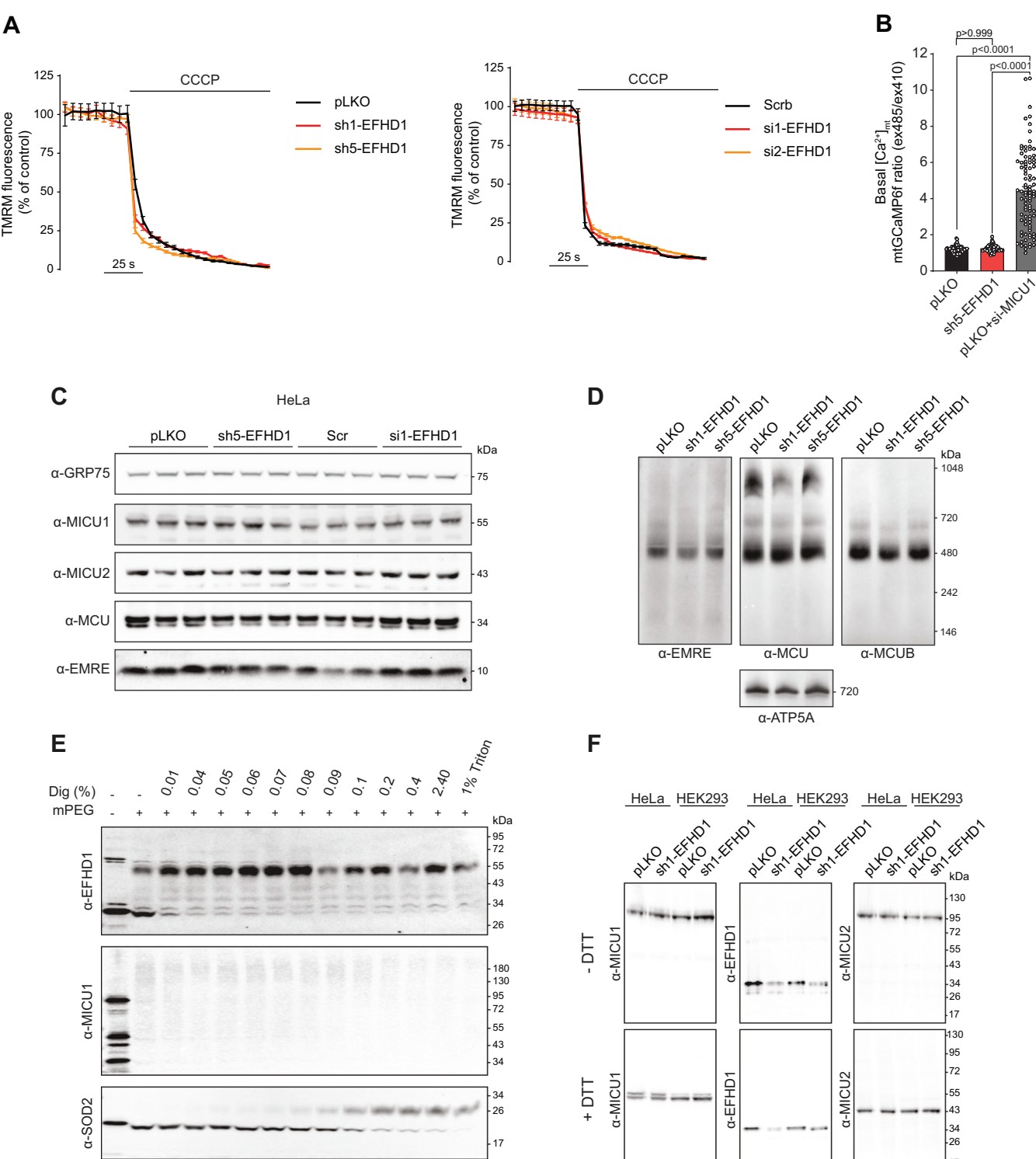

◄ **Figure EV4.   Effect of EFHD1 knockdown on mitochondrial membrane potential, basal Ca²⁺ level, and MCUC assembly.**

(A) TMRM fluorescence in HeLa cells upon stable (sh-EFHD1) or transient (si-EFHD1) EFHD1 silencing. Data are expressed as a percentage of the control (mean ± SEM; n ≥ 53 cells). (B) Resting $[Ca^{2+}]_{mt}$ in HeLa cells upon stable EFHD1 silencing. Data are expressed as the ratio between mt-GCaMP6f fluorescence upon excitation at 485 and 410 nm (mean ± SEM; *n* ≥ 59 cells); one-way ANOVA with Tukey's multiple comparisons test. si-MICU1 is used as positive control. (C) Immunoblot analysis of MCUC protein level in whole cell lysates from HeLa cells upon stable (sh-EFHD1) or transient (si-EFHD1) EFHD1 silencing. GRP75 is used as a loading control. (D) BN-PAGE analysis of MCUC assembly in mitochondria isolated from sh-EFHD1 HeLa cells. ATP5A is used as a loading control. (E) Immunoblot analysis of EFHD1 in isolated mitochondria from HeLa cells treated with increasing concentrations of digitonin and maleimide functionalized polyethylene glycol (mPEG). MICU1 and SOD2 are used as positive controls for IMS and matrix proteins, respectively. (F) Immunoblot analysis of mitochondria from HeLa and HEK293 cells expressing either an empty vector (pLKO) or shRNA against EFHD1. Samples were analyzed in reducing (+ DTT, dithiotreitol) and non-reducing (-DTT) conditions to detect disulfide-mediated oligomerization. Refer to Fig. 5.

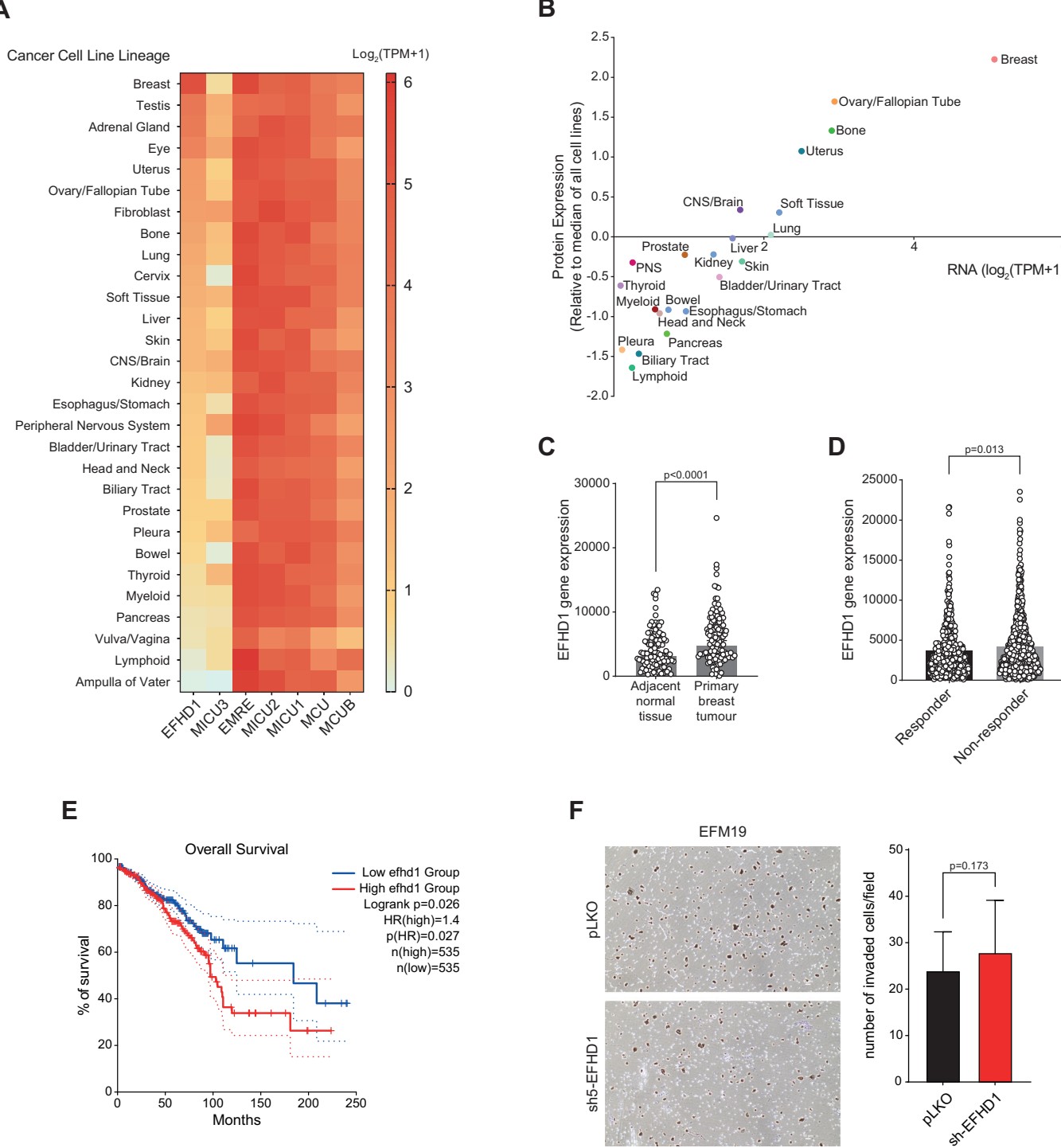

**Figure EV5.  Assessment of EFHD1 as a potential target in cancer.**

(A) Heatmap of gene expression level (TPM, transcripts per million) for the known MCUC components and EFHD1 retrieved from the DepMap Public 23Q2 release (Ghandi et al, 2019) and grouped according to cell line lineage ($n = 1450$ cell lines; 29 lineages). Averaged values are inferred from RNA-sequencing data using the RSEM tool and $\log_2$ transformed, using a pseudo-count of 1 ($\log_2$(TPM + 1)). (B) Correlation between protein and RNA levels of EFHD1 in 1019 different cancer cell lines, grouped based on cell lineage average expression. RNA-sequencing data were retrieved from DepMap Public 23Q2 release (Ghandi et al, 2019) whereas normalized protein expression data were taken from (Nusinow et al, 2020); RNA-protein expression Pearson correlation $r^2 = 0.85$. (C) Median EFHD1 expression in pairs of primary breast tumor and their adjacent normal tissue ($n = 112$ paired samples). Data were extracted from TNMplot.com median ± 95% confidence interval (CI) and Wilcoxon match-paired two-tailed test. (D) Median EFHD1 expression in neoadjuvant chemotherapy in responder ($n = 532$ independent samples) and non-responder ($n = 1100$ independent samples) breast cancer patients analyzed using ROCplot.org from GEO/Array express data median ± 95% CI; Mann–Whitney two-tailed test. (E) Survival of breast cancer patients exhibiting high and low EFHD1 expression. Data were retrieved from TCGA-BRCA; Kaplan–Meier plot and the median expression was used as the cohort cutoff, respectively. HR hazard ratio. Dotted lines indicate 95% confidence interval (CI 95%). (F) Boyden Chamber migration assay on EFM19 pLKO and sh-EFHD1 cells. Representative image (left) and quantification of migrated cells per field (right). Mean ± SEM from 3 independent experiments ($n \geq 30$ biological replicates); Student's *t* test. Refer to Fig. 5.

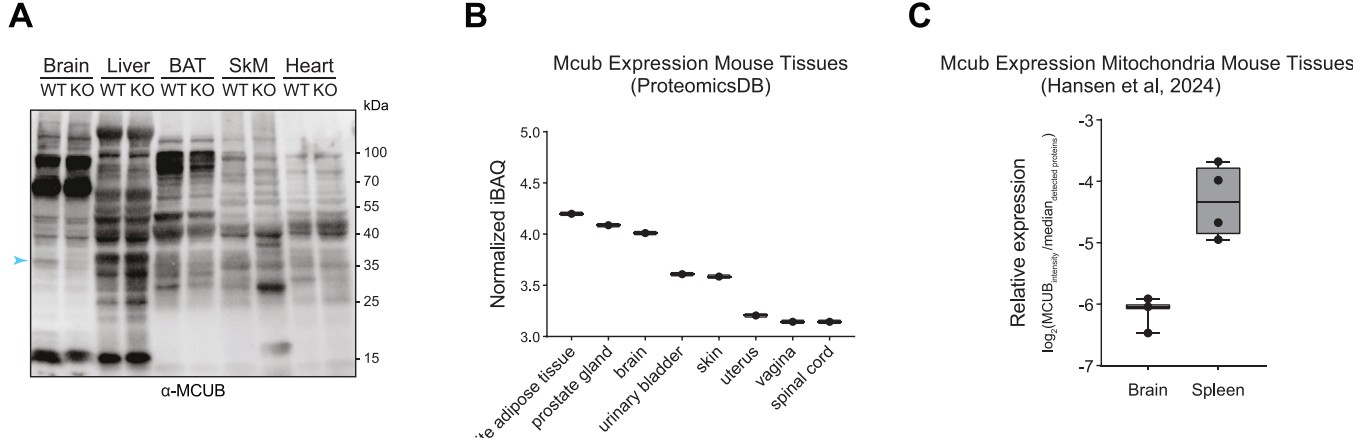

**Figure EV6.  Validation of MCUB as an inhibitor of MCU protein–protein interactions.**

(**A**) Immunoblot analysis of MCUB in whole tissue lysates from MCUB KO and wild-type mice. Blue arrow indicates MCUB protein (BAT, brown adipose tissue; SkM, skeletal muscle). (**B**) Relative MCUB protein expression in mouse tissues from the ProteomicsDB database (white adipose tissue $n = 1$, prostate gland $n = 1$, brain $n = 4$, skin $n = 1$, uterus $n = 1$, vagina $n = 1$, urinary bladder $n = 1$, spinal cord $n = 1$). The line in the middle of the box is plotted at the mean normalized iBAQ, the boxes extend from the minimum normalized iBAQ to the maximum normalized iBAQ values. (**C**) Relative MCUB protein expression in pure mitochondrial proteomes extracted from Hansen et al, 2024. MCUB was detected only in spleen ($n = 4$ biological replicates) and brain ($n = 3$ biological replicates). The line in the middle of the box is plotted at the median, the boxes extend from the 25th to the 75th percentile, and the whiskers extend to the minimum and maximum values. Refer to Fig. 6.

