## [Peer Review File · The EMBO Journal]

Systematic mapping of mitochondrial calcium uniporter channel (MCUC)-mediated calcium signaling networks

Fabiana Perocchi, Hilda Delgado de la Herran, Denis Vecellio Reane, Yiming Cheng, Mate Katona, Fabian Hosp, Elisa Greotti, Jennifer Wettmarshausen, Maria Patron, Hermine Mohr, Natalia Prudente de Mello, Margarita Chudenkova, Matteo Gorza, Safal Walia, Michael Feng, Anja Leimpek, Dirk Mielenz, Natalia Pellegata, Thomas Langer, György Hajnóczky, Matthias Mann, and Marta Murgia

Corresponding author(s): Fabiana Perocchi (fabiana.perocchi@helmholtz-muenchen.de) , Marta Murgia (mmurgia@biochem.mpg.de)

Review Timeline:

Submission Date:	20th Feb 24
Editorial Decision:	27th Mar 24
Revision Received:	22nd Jun 24
Editorial Decision:	25th Jul 24
Revision Received:	8th Aug 24
Accepted:	15th Aug 24

Editor: William Teale

Transaction Report:

Dear Dr. Perocchi,

Thank you again for the submission of your manuscript entitled "Systematic mapping of MCU-mediated mitochondrial calcium signaling networks" and for your patience during the review process. We have now received the reports from the referees, which I copy below.

As you can see from their comments, while the referees need a more careful explanation of the rationale behind aspects of your experimental design and more careful controls in some of the experiments you present, all of them point out the potential value of your work to the scientific community.

Based on the overall interest expressed in these reports, therefore, I would like to invite you to address the comments of all referees in a revised version of the manuscript. I should add that it is The EMBO Journal policy to allow only a single major round of revision and that it is therefore important to resolve the main concerns at this stage. I believe the concerns of the referees are reasonable and addressable, but please contact me if you have any questions, need further input on the referee comments or if you anticipate any problems in addressing any of their points. I am happy to arrange a Zoom call after Easter once you have had a chance to digest the reports. Please, follow the instructions below when preparing your manuscript for resubmission.

I would also like to point out that as a matter of policy, competing manuscripts published during this period will not be taken into consideration in our assessment of the novelty presented by your study ("scooping" protection). We have extended this 'scooping protection policy' beyond the usual 3 month revision timeline to cover the period required for a full revision to address the essential experimental issues. Please contact me if you see a paper with related content published elsewhere to discuss the appropriate course of action.

Again, please contact me at any time during revision if you need any help or have further questions.

Thank you very much again for the opportunity to consider your work for publication. I look forward to your revision.

Best regards,

William

William Teale, Ph.D.
Editor
The EMBO Journal

When submitting your revised manuscript, please carefully review the instructions below and include the following items:

- 1) a .docx formatted version of the manuscript text (including legends for main figures, EV figures and tables). Please make sure that the changes are highlighted to be clearly visible.
- 2) individual production quality figure files as .eps, .tif, .jpg (one file per figure).
- 3) a .docx formatted letter INCLUDING the reviewers' reports and your detailed point-by-point response to their comments. As part of the EMBO Press transparent editorial process, the point-by-point response is part of the Review Process File (RPF), which will be published alongside your paper.
- 4) a complete author checklist, which you can download from our author guidelines ([https://wol-prod-cdn.literatumonline.com/pb-assets/embo-site/Author Checklist%20-%20EMBO%20J-1561436015657.xlsx](https://wol-prod-cdn.literatumonline.com/pb-assets/embo-site/Author%20Checklist%20-%20EMBO%20J-1561436015657.xlsx)). Please insert information in the checklist that is also reflected in the manuscript. The completed author checklist will also be part of the RPF.
- 5) Please note that all corresponding authors are required to supply an ORCID ID for their name upon submission of a revised manuscript.
- 6) We require a 'Data Availability' section after the Materials and Methods. Before submitting your revision, primary datasets produced in this study need to be deposited in an appropriate public database, and the accession numbers and database listed

under 'Data Availability'. Please remember to provide a reviewer password if the datasets are not yet public (see <https://www.embopress.org/page/journal/14602075/authorguide#datadeposition>). If no data deposition in external databases is needed for this paper, please then state in this section: This study includes no data deposited in external repositories. Note that the Data Availability Section is restricted to new primary data that are part of this study.

Note - All links should resolve to a page where the data can be accessed.

8) For data quantification: please specify the name of the statistical test used to generate error bars and P values, the number (n) of independent experiments (specify technical or biological replicates) underlying each data point and the test used to calculate p-values in each figure legend. The figure legends should contain a basic description of n, P and the test applied. Graphs must include a description of the bars and the error bars (s.d., s.e.m.).

9) We would also encourage you to include the source data for figure panels that show essential data. Numerical data can be provided as individual .xls or .csv files (including a tab describing the data). For 'blots' or microscopy, uncropped images should be submitted (using a zip archive or a single pdf per main figure if multiple images need to be supplied for one panel). Additional information on source data and instruction on how to label the files are available at .

10) We replaced Supplementary Information with Expanded View (EV) Figures and Tables that are collapsible/expandable online (see examples in <https://www.embopress.org/doi/10.15252/embj.201695874>). A maximum of 5 EV Figures can be typeset. EV Figures should be cited as 'Figure EV1, Figure EV2" etc. in the text and their respective legends should be included in the main text after the legends of regular figures.

12) Our journal encourages inclusion of *data citations in the reference list* to directly cite datasets that were re-used and obtained from public databases. Data citations in the article text are distinct from normal bibliographical citations and should directly link to the database records from which the data can be accessed. In the main text, data citations are formatted as follows: "Data ref: Smith et al, 2001" or "Data ref: NCBI Sequence Read Archive PRJNA342805, 2017". In the Reference list, data citations must be labeled with "[DATASET]". A data reference must provide the database name, accession number/identifiers and a resolvable link to the landing page from which the data can be accessed at the end of the reference. Further instructions are available at .

Further instructions for preparing your revised manuscript:

At EMBO Press we ask authors to provide source data for the main manuscript figures. Our source data coordinator will contact you to discuss which figure panels we would need source data for and will also provide you with helpful tips on how to upload

and organize the files.

We realize that it is difficult to revise to a specific deadline. In the interest of protecting the conceptual advance provided by the work, we recommend a revision within 3 months (25th Jun 2024). Please discuss the revision progress ahead of this time with the editor if you require more time to complete the revisions. Use the link below to submit your revision:

Referee #1:

In the study by Delgado et al. entitled 'Systematic mapping of MCU-mediated mitochondrial calcium signaling networks', the authors employed an unbiased and quantitative proteomic approach to delineate the MCUC interactome in HEK293 cells under physiological conditions and conditions involving chronic loss or gain of mitochondrial Ca²⁺ uptake. They successfully expanded the "holo-complex" by identifying 139 statistically significant protein-protein interactions (PPIs) between known members of the uniporter and an additional 89 mitochondrial proteins localized in all four sub-mitochondrial compartments, associated with metabolic, neurological, and immunological diseases. The authors also confirmed EFHD1 as a binding partner of MCU, EMRE, and MCUB, demonstrating a MICU1-dependent inhibitory effect on Ca²⁺ uptake. Finally, their results suggest that EFHD1 functions as an inhibitor of MCU-mediated [Ca²⁺]_m uptake.

Comments:

MCU complex and mitochondria are an interesting field of research in several branches of science, both in physiology and pathology. This is due to the roles which they cover during life. As a consequence this manuscript could attract multiple readers and can be potentially useful for further research.

However, as it appears in this current view, it seems a mixture of many things put together. The biochemical method to detect interactors is one of the gold standards and is truly appreciated; nevertheless, it has not been fully exploited. The manuscript reports many information poorly characterized: from the choice and the rationale of the interactors to be further validated to their validation which consist of few experiments: only one panel for MICU3 and PRELID1, one figure for EFHD1 for which either the reviewer or the reader expects a complete characterization by using in vitro and in vivo approaches. At least to better support the conclusions written by the authors.

Also, the choice of the cell lines in which the authors performed the experiments is confusing. The validation process should be done in more than one cell line (possibly encompassing different origin types) and not each candidate in a different cell type without explaining the rationale.

In addition, the authors claimed the experimental plan of the MCUC interactome to be done under physiological conditions and with relevance for human diseases. In my opinion, the MCUC interactome has not been detected in physiological conditions (immortalized cells in which the expression of tagged proteins has been obtained) and few data are linked to the claimed human diseases.

Other general comments:

- how is it possible for MCU to interact with OMM proteins? Maybe it would have been interesting to examine and validate these interactions rather than those reported.
- the authors refer to the interactors as any protein that stably or dynamically binds MCUC. Is it possible to experimentally evaluate the nature of this binding?
- Fig.3C: it is strange that MCU/EMRE/MCUB interactors are more involved in other pathways (i.e., protein homeostasis,

translation) than calcium homeostasis. Can you further discuss this?

- in 2020 it has been reported the interaction between MCU and c subunit of ATP synthase (32184243). How can you discuss this data? Have you found a match through your biochemical approach?
- in explaining their data, the authors meticulously contextualize their findings in HEK293 cells. Can these findings be reproducible also in other contexts?

Other specific comments:

- Fig.4, 5 and supplementary: I really appreciated the wide range of methods applied to measure calcium fluxes. However, why the alternate use of permeabilized and intact cells to assess mitochondrial calcium?
- Fig.4D,E: When PRELID1 is silenced, MCU disappears. In this context, why is mitochondrial calcium uptake reduced by only 20-25%?
- If I am not wrong, in vitro validation of PRELID1 interaction with MCUC (Co-IP) has not been provided.
- Fig.4: what happens if EFHD1 becomes overexpressed? What is its basal expression in these cells?
- Given the purpose of the western blots in Fig. 4B and 4G is the quantification of proteins localized to the mitochondria, it would be advisable to also show a mitochondrial marker (e.g., Tom20).
- Page 8, line 290: in my opinion the statement reported about PRELID1 is not completely supported by the data shown.
- Fig.5A: why in your opinion, the red and black kinetics are so different?
- Page 8, lines 298-303: also mitochondrial calcium affects OXPHOS and ATP production and this seems to not vary among conditions, also, mitochondrial membrane potential is mentioned but not measured.
- Page 9, lines 335: what is the expression of EFHD1 in HeLa cells compared to other one (normal), is it overexpressed? Also, can Paclitaxel alone move calcium inside cells? Ceramide alone does not increase cell death? Why? Fig.5, data about the involvement of EFHD1 in cell viability in cancer cells are poor.
- To complete the panel it would be intriguing to investigate whether reverting the phenotype yields similar effects to those observed in the control group. Specifically, reintroducing wild-type or mutant forms of EFHD1 into cells from which endogenous EFHD1 was consistently depleted using short hairpin RNA (shRNA). Particularly, examining the effects of its reintroduction on viability, migration, and invasion in HeLa cells would be of interest.

Minor comments:

- Lines 306-307: There is an excessive space before the reference
- Line 654 sentence: "MCU, MCUB, and EMRE protein-protein interactions (PPIs)" on page 17, add the acronym at line 158 and remove it by page 17.
- Please standardize Figure S5E: Change "percent" to "%" to match the format used in other graphs.

Referee #2:

In this manuscript, the Perocchi lab utilized sophisticated mass spec techniques and analyses to explore the protein interactome of the mitochondrial uniporter Ca²⁺ channel complex. They employed affinity purification of various components of the complex using tagged protein expressed at close to endogenous levels. In addition, they explored the consequences of deleting components of the channel complex on the interactome. They identify a variety of interacting proteins, including all of those predicted based on previous published work. The interactome they have identified will provide a resource for functional studies for the Ca²⁺ signaling field. As proof of concept, they focused on the EFHD1 as one of the interacting proteins to demonstrate its functional role in Ca uptake by the channel complex.

The analyses are complicated for this reviewer, so I will take the authors at face value. I have relatively few suggestions for minor revision.

1. There is a tendency in several instances of the authors equating physical interactions they detect with Ca²⁺ signaling. The authors detect interactions, which may or may not be due to or regulated by Ca. Lines 189, 430, 432, in which they use identified protein interactions to infer that certain processes are regulated by mito Ca²⁺. The authors should change the wording.
2. The analyses were limited to known mitochondrial proteins, which begs that question of how many non-mitochondrial proteins were detected and eliminated from the analyses. This goes to the issue of specificity and promiscuity of the approaches, which the authors otherwise address, but I think that this should be commented upon.
3. I was surprised to learn of a database, GTE_x, that quantifies protein expression levels across tissues. Is this database based on quantitative mass spec analyses, or on what, and what degree of confidence do the authors have in it? Maybe a comment here.
4. It seems an omission to have not measured mitochondrial inner membrane potential in the analyses of the effects of EFHD1 expression manipulations. As the major driving force for Ca²⁺ uptake, this would seem to be necessary to make the conclusions that the authors have come to.
5. The authors comment on the interactions with PDH, but in Fig 3D, it seems that the interactions are limited to MCUB, and with dashed lines. Dashed lines represent local and global correlations. It is unclear what this all means..on the one hand it is

emphasized in the text, on the other it seems like an correlative interaction with a single complex component. It would be very helpful for the non-experts if these types of correlations could be explained in a bit of detail so the reader can appreciate better what kind of interaction is implied.

6. Methods: what is ATP/MgCl₂?

7. Methods; what is the free Ca²⁺ concentration in the media in the permeabilized cell experiments?

8. Figure 4A. What is the meaning of the numbers?

9. Fig 2B. After induction?

Referee #3:

This manuscript report on the relationships between the MCU complex and the complexes and pathways with which it communicates that both regulate its function and are potentially regulated by the MCU complex through mitochondrial Ca²⁺ or indirectly by mitochondrial Ca²⁺ independent manner. The approach taken by the authors is isolation of the protein interaction landscape of the various components of the MCU channel complex (MCUC) at conditions close to physiological and when several components are eliminated to determine their potential role in assembly and function of the MCUC and few cellular activities, in particular cell growth and toxicity. Finally, the validity and mitochondrial function of a few interactors, most notably of MICU3 and EFHD1, is examined in detail.

Overall, the findings provide a very useful tool for mitochondrial interactors that is determined with high confidence and should be welcomed by the field and will likely lead to new studies and perhaps new research directions. The findings with MICU3 and in particular EFHD1 are also important and provide interesting new information. I have a few fairly minor comments that are listed below chronically rather than importance.

1. Page 2: Please cite PMID: 7634331 together with (Tsai et al, 2022; Fecher et al, 2019; Fieni et al, 2012; Paillard et al, 2017) and PMID: 20693986 together with (Perocchi et al, 2010).

2. Pages 4-5: The authors indicate that "Among the MCUC interactors, ... more than 50% of all preys were bait-specific, suggesting their involvement in the selective maturation and regulation of MCU, EMRE, or MCUB" This raised the question can or whether any of these proteins found in complexes different from MCU? One way to validate this and perhaps get a clue for their significance is to isolate the PPIs from cells transfected with all baits. This should also provide information on complexes that are strictly dependent on the intact complex formed by the tagged proteins.

3. Figure 5: It is not clear why changing mitochondrial Ca²⁺ by the sh-EFHD1 had no effect on mitochondrial oxygen consumption given that changes in mitochondrial Ca²⁺ has profound role in Oxphos and oxygen consumption. Can the authors explain and comment on.

4. In general, fonts and labels are difficult to see on laptops and when printed. I had to use a very large screen to clearly see the figures. More contrasting colors and larger fonts can help readers.

Response to Reviewers

Dear Reviewers,

We extend our gratitude for taking the time to review our manuscript and providing constructive feedback. We have performed additional experiments and revised the text to fully address your concerns and suggestions. We believe the revised manuscript has gained clarity and hope you will find it suitable for publication at the *EMBO Journal*.

Reviewer #1

MCU complex and mitochondria are an interesting field of research in several branches of science, both in physiology and pathology. This is due to the roles which they cover during life. As a consequence this manuscript could attract multiple readers and can be potentially useful for further research.

Response: We appreciate the positive comments of the Reviewer on the relevance of our study and findings.

However, as it appears in this current view, it seems a mixture of many things put together. The biochemical method to detect interactors is one of the gold standards and is truly appreciated; nevertheless, it has not been fully exploited. The manuscript reports many information poorly characterized: from the choice and the rationale of the interactors to be further validated to their validation which consist of few experiments: only one panel for MICU3 and PRELID1, one figure for EFHD1 for which either the reviewer or the reader expects a complete characterization by using in vitro and in vivo approaches.

Response: We thank the reviewer for raising this point and would like to take the opportunity to clarify the rationale behind our approach. The goal of our study is to provide the scientific community with a resource and discovery tool for MCUC interactors, which will likely open new research directions. What the Reviewer defines as a “*mixture of many things put together*” is instead a survey of different properties of the MCUC protein interaction network, from its link to mitochondrial functions to its tissue distribution, disease connection, and remodelling upon loss and gain-of-function. Therefore, the novelty offered by our study is to be found in the dataset itself rather than in the validation of a few candidates. PRELID1, MICU3, EFHD1 and MCUB are here used as a proof-of-concept of how our analysis can recapitulate what was known before but also go beyond. We like to think that each candidate in our interactome could represent a novel research direction and for this reason we have submitted our manuscript to the *EMBO Journal* as a Resource Article.

We acknowledge the challenge of fully exploring every hypothesis arising from the analysis of our dataset. We have made extensive efforts to validate several findings and to demonstrate the robustness of our discoveries. Indeed, the validation of MICU3's role in the regulation of mitochondrial calcium homeostasis in HEK293 cells is supported by three panels (A-C) presented in the Figure 4, and in eight panels (A-I) in Supplementary Figure S2, addressing mitochondrial localization, dimerization and role of EF-hand domains. Those findings serve as a proof of concept for the sensitivity of our approach, given that MICU3 was missed as a MCU interactor in HEK293 cells in previous analyses (Sancak *et al*, 2013; Austin *et al*, 2022; Antonicka *et al*, 2020). Similarly, for PRELID1, in addition to panels D and E in Figure 4, we offer three additional panels (J-L) in Supplementary Figure S2, addressing its effect on mitochondrial calcium uptake, cytosolic calcium uptake, mitochondrial membrane potential, and MCU assembly and stability. Likewise, our finding that EFHD1 is a novel regulator of mitochondrial calcium uptake is supported by seven panels (F-L) in Figure 4, eleven panels in Figure 5, and two Supplementary Figures (S3 and S4). Altogether, these panels address EFHD1 localization, its role in the regulation of mitochondrial calcium signalling, as well as insights into the pathological relevance of EFHD1 in breast cancer cells.

Also, the choice of the cell lines in which the authors performed the experiments is confusing. The validation process should be done in more than one cell line (possibly encompassing different origin types) and not each candidate in a different cell type without explaining the rationale.

Response: We have consistently used both HEK293 and HeLa cells to characterize and validate the MCUC protein-protein interaction network. Those are the most widely used cell lines in the biological research community and recognized as the gold standard for biochemical analyses and for investigating mitochondrial calcium signalling.

In addition, the authors claimed the experimental plan of the MCUC interactome to be done under physiological conditions and with relevance for human diseases. In my opinion, the MCUC interactome has not been detected in physiological conditions (immortalized cells in which the expression of tagged proteins has been obtained) and few data are linked to the claimed human diseases.

Response:

We thank the Reviewer for the opportunity to clarify this point. We refer to “physiological conditions” as those conditions devoid of any perturbation (e.g., genetic manipulations of the MCUC complex) in the following text:

Line 44-45, “...*Here, we map the MCUC interactome under physiological conditions and upon chronic loss or gain of mitochondrial calcium uptake. ...”.*

Line 94-95: “...*Here, we devised a biochemical strategy to characterize the MCUC interactome in human cells with high confidence and resolution under physiological conditions and following genetic perturbations.*...”.

Line 116-117: “...*We devised a biochemical workflow to achieve an unbiased and comprehensive characterization of MCUC-specific PPIs in human cells under near-physiological conditions.*...”.

Line 353: “...*Mapping of the MCUC network at physiological conditions clearly highlighted that the molecular composition of MCUC and the functional associations between mitochondria and Ca²⁺ are far more complex than previously anticipated (Sancak et al., 2013).*...”.

Line 482: “...*In summary, our MCUC interaction map under physiological conditions and genetic perturbation of the different MCUC components,*...”.

We also use the wording “near-physiological conditions” to distinguish our strategy from the one of others interrogating MCUC interactions upon non-physiological overexpression of the bait (Sancak et al, 2013) as in the following text:

Line 120-121: “...*We devised a biochemical workflow to achieve an unbiased and comprehensive characterization of MCUC specific PPIs in human cells under near-physiological conditions.*...”.

To address the concerns of the Reviewer we have now replaced “physiological conditions” with “resting conditions” to better reflect our experimental context.

Other general comments:

- how is it possible for MCU to interact with OMM proteins? Maybe it would have been interesting to examine and validate these interactions rather than those reported.

Response: Thank you for raising this important point. Several OMM proteins contain domains that are exposed to the IMS. Therefore, it is possible that MCUC components that are either localized or exposing a domain in the IMS to engage in PPIs with OMM proteins. Another possibility for members of the MCU complex to interact with OMM proteins is through direct engagement at the IMM-OMM contact sites. As an example, MICU1 was shown to interact and regulate the MICOS complex (Tomar et al, 2023), which is known to

bind to OMM complexes such as the sorting and assembly machinery (SAM) (Tang *et al*, 2020). Given the existence of structural and functional bridges facilitating the interaction between IMM and OMM, it is possible to speculate that direct interactions between MCU and OMM proteins could form to regulate for example mitochondrial morphology, dynamics, and inter-organelles contacts. While our study validates other interactions, follow-up analyses assessing potential links between MCU and OMM proteins could be certainly of great interest.

- the authors refer to the interactors as any protein that stably or dynamically binds MCUC. Is it possible to experimentally evaluate the nature of this binding?

Response: The Reviewer is correct that our current experimental design does not allow evaluating the nature of protein-protein interactions. Our definition of “dynamic interactors” refers to those not fulfilling the main criteria used to define “stable binding”, namely high fold enrichment and low *p* values, which apply for example to some core members of the MCUC. We have used the same two criteria in our recent analysis of the yeast protein interactome, where over 4000 pull-downs generated a highly structured network of 3,927 proteins connected by 31,004 interactions (Michaelis *et al*, 2023). While it was not our intention to discriminate between “dynamic” and “stable” interactions, we observed from the analysis of our results that we were able to detect both. For instance, the interactions of EMRE and MCU with the mAAA proteases and the subunits of the TOM/TIM complexes exemplify “dynamic” interactions. Instead, the interactions between components of MCU complex most likely represent “stable” interactions. While there are methodologies designed to assess specifically the nature of protein binding, those fall beyond the scope of this study. To address the Reviewer’s concern, we have now replaced the word “dynamically” with “transiently”.

- Fig.3C: it is strange that MCU/EMRE/MCUB interactors are more involved in other pathways (i.e., protein homeostasis, translation) than calcium homeostasis. Can you further discuss this?

Response: We believe the presence of interactors involved in mitochondrial pathways such as protein homeostasis and translation, in addition to calcium homeostasis, reflects the multifaceted nature of the MCU complex, which we discuss at page 3, Line 80-93. By using components of the MCU complex as baits for TAPs from cell lysate, it is possible to capture interactions between baits and pathways/complexes involved for example in their import, maturation, assembly, etc. This does not imply that MCU, EMRE, and MCUB are more engaged in interactions with other pathways than calcium homeostasis, but shows that our network can unbiasedly and comprehensively capture them. Our findings are consistent with the emerging view that mitochondrial calcium signalling is intricately linked with broader cellular functions.

- in 2020 it has been reported the interaction between MCU and c subunit of ATP synthase (32184243). How can you discuss this data? Have you found a match through your biochemical approach?

Response: Thank you for bringing up this interesting finding. Indeed, the interaction between MCU and the c subunit of ATP synthase provides valuable insights into the potential crosstalk between mitochondrial calcium uptake and ATP synthesis. Our biochemical approach detects the J subunit of the ATP synthase complex (ATP5MPL) and the ATP synthase assembly factor, TMEM70, which was recently described to interact with

the ATP synthase subunit c (Kovalčíkova *et al*, 2019; Bahri *et al*, 2021). Therefore, we now cite those findings in the revised manuscript as follows:

Page 6, Line 199: "...Furthermore, our biochemical approach detected interactions between MCU, the J subunit of the ATP synthase complex (ATP5MPL), and the ATP synthase assembly factor, TMEM70. The latter was recently described to interact with the ATP synthase subunit c (Kovalčíkova *et al*, 2019; Bahri *et al*, 2021), which was previously shown to bind MCU in trypanosomes and human cells (Huang & Docampo, 2020)...".

- in explaining their data, the authors meticulously contextualize their findings in HEK293 cells. Can these findings be reproducible also in other contexts?

Response: We used HEK293 cells for TAPs because they are easy to genetically manipulate, widely used for biochemical analyses and recognized as the gold standard for investigating mitochondrial calcium signalling. Regarding the reproducibility of our results, several of our interactions have already been validated by others and we thoroughly discuss those findings in our manuscript. We believe that the experimental validation of several interactors performed in our study as well as evidence from previous publications provide a strong proof-of-concept that novel PPIs identified in our MCUC protein network can be reproducible and worth investigating.

Other specific comments:

- Fig.4, 5 and supplementary: I really appreciated the wide range of methods applied to measure calcium fluxes. However, why the alternate use of permeabilized and intact cells to assess mitochondrial calcium?

Response: We measure calcium kinetics sequentially (not alternating) in intact HeLa cells and then in permeabilized cells to demonstrate that the mitochondrial calcium phenotype observed in response to genetic manipulations of candidate genes is not due to changes in signalling pathways upstream MCU-mediated mitochondrial calcium uptake. If a defect in mitochondrial calcium uptake is consistently observed in response to agonist-induced stimulation of ER-to-Mitochondria calcium transfer and CRAC-dependent entry of calcium in intact cells, as well as in response to a bolus of calcium after permeabilization of the plasma membrane, it is valid to conclude that the observed phenotype is due to changes in mitochondrial calcium uptake.

- Fig.4D,E: When PRELID1 is silenced, MCU disappears. In this context, why is mitochondrial calcium uptake reduced by only 20-25%?

Response: We thank the Reviewer for his/her insightful question. While it is true that the Blue-Native PAGE shows a dramatic impairment in the assembly of the MCUC upon PRELID1 silencing, the SDS-PAGE analysis in Fig. S2J provides a more nuanced picture. The SDS-PAGE results indicate that the overall protein expression level of MCU is reduced by about 50%. In this scenario, a reduction of approximately 40% in mitochondrial calcium uptake (from 85.8 to 55.9 μ M) is consistent with the observed decrease in MCU expression levels.

- If I am not wrong, in vitro validation of PRELID1 interaction with MCUC (Co-IP) has not been provided.

Response: We did not perform western blot-based co-immunoprecipitation experiments in cells overexpressing both PRELID1 and MCUC components, as correctly pointed out by the Reviewer. Our choice was dictated by the consideration that: i) our approach is less prone to overexpression artifacts due to near-endogenous bait expression; ii) quantitative mass spectrometry-based proteomics confers higher sensitivity, accuracy and reproducibility

across replicates than other approaches employing relative quantification of proteins; iii) we already identified and quantified PRELID1-MCUC interaction in 5 biological replicates and using two different MCUC components as baits.

- Fig.4: what happens if EFHD1 becomes overexpressed? What is its basal expression in these cells?

Response: The effect of EFHD1 overexpression on mitochondrial calcium uptake was analyzed and reported in Figure 4L. Based on this result, the overexpression of EFHD1 in HeLa cells did not affect mitochondrial calcium uptake, possibly due to the high endogenous level of EFHD1 in HeLa cells, as already discussed at Page 12 lines 464-470:

“Importantly, the heterogeneous expression pattern of EFHD1 among human tissues and cancer cell lines could also explain why EFHD1 loss- or gain-of-function would affect $[Ca^{2+}]_{mt}$ homeostasis to different extents in different cell types. Accordingly, in HeLa cells that already express EFHD1 at a high level, we failed to observe an alteration in $[Ca^{2+}]_{mt}$ upon EFHD1 overexpression. Instead, its overexpression in cells like ccRCC that show 30-fold lower level of endogenous EFHD1 compared to HeLa cells was found to significantly reduce $[Ca^{2+}]_{mt}$ (Meng et al, 2023). t.”.

- Given the purpose of the western blots in Fig. 4B and 4G is the quantification of proteins localized to the mitochondria, it would be advisable to also show a mitochondrial marker (e.g., Tom20).

Response: We thank the Reviewer for this suggestion. We have now probed for the mitochondrial protein GRP75 in Fig. 4B and 4G.

- Page 8, line 290: in my opinion the statement reported about PRELID1 is not completely supported by the data shown.

Response: We hope the Reviewer will find the following statement more appropriate:

“...We spotlight PRELID1 as a candidate protein linking membrane phospholipid metabolism, MCUC complex stability and $[Ca^{2+}]_{mt}$ homeostasis...”.

- Fig.5A: why in your opinion, the red and black kinetics are so different?

Response: We are not sure how to interpret the Reviewer's question. Is the Reviewer asking why the mitochondrial calcium uptake kinetics of intact cells stimulated with histamine in Figure 4H look so different from the kinetics of permeabilized cells in Figure 5A? In both experimental conditions we observe a profound effect of EFHD1 knockdown on mitochondrial calcium uptake. The measurements of calcium uptake in permeabilized cells are designed to accentuate differences in uptake kinetics by exposing mitochondria of permeabilized cells to a relatively high dose of calcium in the milieu.

- Page 8, lines 298-303: also mitochondrial calcium affects OXPHOS and ATP production and this seems to not vary among conditions, also, mitochondrial membrane potential is mentioned but not measured.

Response: We thank the Reviewer for his/her suggestions. Those points were also raised by Reviewer 2 and 3 and therefore we have performed additional experiments to validate our initial findings and to measure membrane potential upon loss of EFHD1 function. As an agonist-induced increase in mitochondrial Ca^{2+} uptake can stimulate aerobic metabolism, it is plausible that upon histamine stimulation mitochondrial bioenergetics would

be affected in cells with reduced EFHD1 expression. However, the measurements of oxygen consumption rate (OCR) reported in Figure 5B were performed in cells at “resting conditions” (not stimulated by histamine or other calcium agonists). In these conditions, we would not expect profound alterations in OXPHOS and ATP production, unless EFHD1 knockdown affects basal mitochondrial calcium levels. Therefore, to address the Reviewer’s concern we measured basal mitochondrial Ca^{2+} levels using the high-affinity calcium probe mt-GCaMP6 in EFHD1 knockdown, control (pLKO), and MICU1 knockdown HeLa cells. The latter is used as a positive control given its loss-of-function is known to increase mitochondrial Ca^{2+} levels in resting conditions. As shown in the figure below, basal mitochondrial calcium levels remained unaffected by EFHD1 LOF. We have included these new data in the Supplementary Figure 4 (panel B).

We have also measured the mitochondrial membrane potential with the potentiometric probe TMRM under both shRNA (left) and stable siRNA-mediated silencing (right) of EFHD1. Our results showed no significant differences in mitochondrial membrane potential across these conditions. We would like to emphasize that these experiments have the purpose to control for factors affecting mitochondrial calcium uptake rather than to study the physiological role of EFHD1 in the regulation of mitochondrial energy metabolism. These new results should help explaining why we could not detect differences in OCR between control and shEFHD1 cells in unstimulated cells and confirm that the mitochondrial calcium phenotype observed upon EFHD1 knockdown is not due to differences in the driving force for calcium uptake. We have included these new data in the Supplementary Figure 4 (panel A).

- Page 9, lines 335: what is the expression of EFHD1 in HeLa cells compared to other one (normal), is it overexpressed?

Response: Comparing the expression of EFHD1 between a cancer cell line like HeLa and “normal” cells in a tissue or organ can be challenging. This is partly due to the lack of comprehensive databases that facilitate direct comparisons and a clear definition of what can be really considered a “normal” cell line. Instead, it is possible to compare the expression of EFHD1 between cancer cell lines, as we do in our study (page 12, line 464):

“...Importantly, the heterogeneous expression pattern of EFHD1 among human tissues and cancer cell lines could also explain why EFHD1 loss- or gain-of-function would affect $[Ca^{2+}]_{mt}$ homeostasis to different extents in different cell types. Accordingly, in HeLa cells that already express EFHD1 at a high level, we failed to observe an alteration in $[Ca^{2+}]_{mt}$ upon EFHD1 overexpression. Instead, its overexpression in cells like ccRCC that show 30-fold lower level of endogenous EFHD1 compared to HeLa cells was found to significantly reduce $[Ca^{2+}]_{mt}$ (Meng et al, 2023)...”.

We also show in Supplementary Figure 5 that EFHD1 gene expression is highly variable across 1450 cancer cell lines, with an expression in HeLa cells that was higher than in clear cell renal cell carcinoma (ccRCC) cells, but lower than in breast cancer cell lines.

Also, can Paclitaxel alone move calcium inside cells? Ceramide alone does not increase cell death? Why?

Response: Paclitaxel has been shown to influence calcium signalling in various cell lines, such as neuroblastoma and pancreatic acinar cells (Boehmerle *et al*, 2006; Kidd *et al*, 2002). However, to the best of our knowledge, there are no specific reports about the effect of paclitaxel on calcium signalling in HeLa cells. The aim of the experiment reported in Figure 5I was to test for a synergistic effect of EFHD1 loss-of-function and cell death activators, rather than to study the role of C2-ceramide and paclitaxel in regulating calcium signalling. Therefore, we chose a sub-lethal dose of those compounds at which, no significant cell death should be observed. As hypothesized, we found that EFHD1 knockdown led to an increased susceptibility to cell death in cells treated with either drugs. To clarify the rationale behind our experimental design, we have changed the text as follow:

“As shown in Fig. 5H, the KD of EFHD1 in HeLa cells significantly decreased cell viability and sensitized cells to sub-lethal doses of apoptotic inducers such as C2-ceramide and paclitaxel (Fig. 5I).”

Fig.5, data about the involvement of EFHD1 in cell viability in cancer cells are poor. To complete the panel it would be intriguing to investigate whether reverting the phenotype yields similar effects to those observed in the control group. Specifically, reintroducing wild-type or mutant forms of EFHD1 into cells from which endogenous EFHD1 was consistently depleted using short hairpin RNA (shRNA). Particularly, examining the effects of its reintroduction on viability, migration, and invasion in HeLa cells would be of interest.

Response: We thank the Reviewer for suggesting this key experiment, which is necessary

to demonstrate that the mitochondrial calcium phenotype observed upon knockdown of EFHD1 is due to EFHD1 loss-of-function. We have included the results in the Supplementary Figure 3 (panel E). We find that the overexpression of WT EFHD1 significantly rescued the mitochondrial calcium

phenotype triggered by EFHD1 knockdown. In contrast, an EFHD1 mutant in the calcium binding domains was unable to rescue the increase in mitochondrial calcium level observed upon loss of EFHD1 function. This also demonstrates that the ability to sense Ca^{2+} is essential for EFHD1 to function and is in line with the observation that the overexpression of an EF-hands mutant mimics a loss-of-function phenotype (Figure 4L).

Minor comments:

-Lines 306-307: There is an excessive space before the reference

Response: We have fixed the formatting issue.

-Line 654 sentence: "MCU, MCUB, and EMRE protein-protein interactions (PPIs)" on page 17, add the acronym at line 158 and remove it by page 17.

Response: We have edited the text accordingly.

-Please standardize Figure S5E: Change "percent" to "%" to match the format used in other graphs.

Response: Thank you for pointing this out.

Response to Reviewer #2:

1. There is a tendency in several instances of the authors equating physical interactions they detect with Ca²⁺ signaling. The authors detect interactions, which may or may not be due to or regulated by Ca. Lines 189, 430, 432, in which they use identified protein interactions to infer that certain processes are regulated by mito Ca²⁺. The authors should change the wording.

Response: We thank the Reviewer for raising this important suggestion. We have now changed the wording as follows:

- *"...In addition, our results raise the hypothesis that members of the MCUC could participate in the regulation of contact sites between inner and outer mitochondrial membranes..."*.
- *"...As MCUC interactors, we also identified proteins involved in mt-DNA maintenance and replication, mRNA transcription and protein translation, as well as enzymes of the beta-oxidation pathway (e.g. ECHS1, ACACB, ACOT7). Interestingly, the impairment of mt-Ca²⁺ uptake was already associated with the rewiring of energy production from glycolysis to fatty acid oxidation in the skeletal muscle (Kwong et al, 2018; Gherardi et al, 2018; Huo et al, 2023), but the molecular mechanisms underlying such metabolic flexibility remain unknown..."*.

By revising these sentences, we aim to clarify that the detected protein interactions do not necessarily imply a direct regulation by calcium.

2. The analyses were limited to known mitochondrial proteins, which begs that question of how many non-mitochondrial proteins were detected and eliminated from the analyses. This goes to the issue of specificity and promiscuity of the approaches, which the authors otherwise address, but I think that this should be commented upon.

Response: We would like to thank the reviewer for raising this key point. Indeed, it is plausible that our baits could interact with non-mitochondrial proteins, even if they exert their function in mitochondria. Those interactions would be captured by our approach, and should not necessarily be considered as false positives. Indeed, our TAPs are performed using

whole cell lysates as starting material (instead of isolated mitochondria). Moreover, all three baits are encoded by the nuclear genome, translated in the cytosol, and imported in their unfolded states before being processed and assembled into their functional states in mitochondria.

Given our interest was mainly to characterize the interaction partners of MCUC within mitochondria, we have focused our analyses on mitochondrial preys. Therefore, we aim to clarify that we have not eliminated non-mitochondrial proteins from the analysis to make the results of our TAPs look cleaner. Indeed, our dataset can also be mined to search for interactions between MCU, EMRE and MCUB and non-mitochondrial proteins using the raw files deposited in ProteomeXchange Consortium under the accession number PXD040893. Those interactions could potentially point to cellular processes involved for example in translational regulation of the baits or their targeting to mitochondria. As suggested by the Reviewer, we show below the result of our analysis without filtering for mitochondrial proteins. As expected, we identify several non-mitochondrial proteins as putative MCU interactors. However, known MCUC components such as EMRE, MCUB, MICU1, and MICU2 still remain among the most enriched preys.

We have now tried to clarify the rationale of our approach in the main text (page 4).

3. I was surprised to learn of a database, GTEx, that quantifies protein expression levels across tissues. Is this database based on quantitative mass spec analyses, or on what, and what degree of confidence do the authors have in it? Maybe a comment here.

Response: Originally, the GTEx database included primarily quantification of gene expression levels using RNA sequencing. However, in 2020, Jiang et al. published the results of quantitative mass spectrometry analyses performed on the tissue samples used for RNA-sequencing. We consider the GTEx database highly reliable due to the extensive use of high-throughput proteomic and transcriptomic techniques, cross-validation with RNA-sequencing data, and rigorous statistical analyses. Cell type-resolved proteomics atlases of the human body are being generated using recently developed spatial proteomics techniques and could represent a major reference point in the future.

4. It seems an omission to have not measured mitochondrial inner membrane potential in the analyses of the effects of EFHD1 expression manipulations. As the major driving force for Ca²⁺ uptake, this would seem to be necessary to make the conclusions that the authors have come to.

Response: We thank the Reviewer for his/her suggestion. We measured the mitochondrial membrane potential using the potentiometric probe TMRM (Tetramethylrhodamine, Methyl Ester) under both acute (siRNA) and stable (shRNA) silencing of EFHD1. As shown below,

we did not detect significant differences in the mitochondrial membrane potential of control and EFHD1 knockdown cells.

We have included these new data in our revised manuscript (Supplementary Figure 4A).

5. The authors comment on the interactions with PDH, but in Fig 3D, it seems that the interactions are limited to MCUB, and with dashed lines. Dashed lines represent local and global correlations. it is unclear what this all means..on the one hand it is emphasized in the text, on the other it seems like an correlative interaction with a single complex component. It

would be very helpful for the non-experts if these types of correlations could be explained in a bit of detail so the reader can appreciate better what kind of interaction is implied.

Response:

In a Volcano plot a quantitative bait-prey co-enrichment analysis is performed based on a two-tailed Welch's t-test, a within-group variance (s_0) of 1, and a permutation-based false discovery rate (FDR) of either 0.05 ("high-confidence") or 0.10 ("medium-confidence"). However, proteins that do not cross any FDR threshold can still represent true positive interactors. Therefore, to be comprehensive, we used the correlation coefficients of interacting proteins as an additional qualifier to the FDR-controlled confidence of the Volcano plot. In protein-protein interaction studies, true positive interacting proteins often exhibit a good correlation of their intensity profiles across different samples, as demonstrated in our previous studies (Hein *et al*, 2015; Michaelis *et al*, 2023) and

shown below for the MCUC components in our network.

Protein correlation profiling is often used to find stable complexes and to assign proteins to subcellular structure (as an example please look at (Andersen *et al*, 2003; Havugimana *et al*, 2012; Kristensen *et al*, 2012). Independent affinity purification experiments can serve the same purpose if the proteins are quantified across many samples and we have previously demonstrated the usefulness of profile correlation as an additional determinant for gauging protein-protein interactions (Hein *et al*, 2015; Michaelis *et al*, 2023). In our analysis, local correlation is used as a less stringent criteria to identify PPIs, because it only considers the correlation between the expression profile of the bait and any other single protein across biological replicates of TAPs from control and bait expressing cells. Instead, global profile

correlation analyses use all the experimental conditions (19 in our case: 4 TAPs from Control cells, 5 TAPs from MCU-tag, 5 TAPs from EMRE-tag, 5 TAPs from MCUB-tag) to infer PPIs. We hope this clarifies the Reviewer's concerns. We have now attempted to clarify this point also in the Material and Methods section of the revised manuscript (Page 17).

6. Methods: what is ATP/MgCl₂?

Response: ATP/MgCl₂ refers to an equimolar solution of ATP and magnesium chloride. In biochemical experiments, magnesium ions (Mg²⁺) are often required as a cofactor for ATP, as they facilitate the proper functioning and stability of ATP in enzymatic reactions and other cellular processes. Therefore, ATP/MgCl₂ is used to ensure that ATP is available in its active, magnesium-bound form.

7. Methods; what is the free Ca²⁺ concentration in the media in the permeabilized cell experiments?

Response: The experiment is designed to achieve a free Ca²⁺ concentration of about 5 μM in solution. We have corrected the methods section to accurately reflect this information.

8. Figure 4A. What is the meaning of the numbers?

Response: The numbers in the heat map represent the relative expression levels of MCUC components over the median intensity of all detected proteins in the specific proteome of each cell line. To enhance clarity, we have integrated this information into the figure legend.

9. Fig 2B. After induction?

Response: We edited the figure legend as follows:

"Bait enrichment after TAP from whole cell lysates (WCL) of HEK293 cells after tetracycline-driven expression of the bait".

Reviewer #3:

Overall, the findings provide a very useful tool for mitochondrial interactors that is determined with high confidence and should be welcomed by the field and will likely lead to new studies and perhaps new research directions. The findings with MICU3 and in particular EFHD1 are also important and provide interesting new information. I have a few fairly minor comments that are listed below chronically rather than importance.

Response: We thank the Reviewer for his/her positive feedback.

1. Page 2: Please cite PMID: 7634331 together with (Tsai et al, 2022; Fecher et al, 2019; Fieni et al, 2012; Paillard et al, 2017) and PMID: 20693986 together with (Perocchi et al, 2010).

Response: We have followed the Reviewer's suggestion and cited the study of Hajnóczky et al., Cell 1995 (PMID: 7634331) at Page 2, Line 66. However, the second PMID suggested by the Reviewer (20693986) corresponds to the study by Perocchi et al., Nature 2010, which we already cite at Page 2, Line 74. We would be happy to cite other relevant studies that we may have overlooked.

2. Pages 4-5: The authors indicate that "Among the MCUC interactors, ... more than 50% of all preys were bait-specific, suggesting their involvement in the selective maturation and

regulation of MCU, EMRE, or MCUB" This raised the question can or whether any of these proteins found in complexes different from MCU? One way to validate this and perhaps get a clue for their significance is to isolate the PPIs from cells transfected with all baits. This should also provide information on complexes that are strictly dependent on the intact complex formed by the tagged proteins.

Response: We thank the Reviewer for this insightful comment. The possibility that MCU, EMRE, or MCUB can have more interaction partners than just members of the uniporter complex is certainly valid. As an example, we detected interactions between baits and proteins involved in mitochondrial proteostasis and protein import, which likely occur during maturation and sorting of baits into mitochondrial subcompartments. In addition, recent studies suggest that MICU1 is also involved in the regulation of the MICOS complex independently of its function as MCUC gatekeeper. This finding shows that core components of the MCUC can have other functions in mitochondrial physiology, as correctly pointed out by the Reviewer. Indeed, our experimental approach captures all interactions formed by the three baits. Particularly for the interactions that are not shared among the three baits, it does not discriminate whether these interactions occur solely within the MCUC or whether they are promiscuous with other complexes. This would still hold true if we transfected the cells with all three baits at a time, which, in addition, would presumably alter the stoichiometry between MCUC subunits. Proximity labelling approaches combined with affinity purification

might shed light on the heterogeneous nature of these interactions in the future. Notably, the existence of shared interactions between protein complexes is not limited to members of the uniporter complex, but rather an emerging feature of large-scale interactomes, as recently shown in our study yeast, where the vast majority of proteins resulted highly connected, with an average of 16 interactors (Michaelis et al., Nature 2023).

3. Figure 5: It is not clear why changing mitochondrial Ca²⁺ by the sh-EFHD1 had no effect on mitochondrial oxygen consumption given that changes in mitochondrial Ca²⁺ has profound role in Oxphos and oxygen consumption. Can the authors explain and comment on.

Response: We thank the Reviewer for raising this important point. Indeed, it is well established that agonist-induced increase in mitochondrial Ca²⁺ uptake stimulates aerobic metabolism. It is therefore plausible that upon histamine stimulation mitochondrial bioenergetics of cells with reduced EFHD1 expression could be affected. We hypothesize that the reason why changing mitochondrial Ca²⁺ level by EFHD1 knockdown did not alter mitochondrial oxygen consumption is that the measurements of oxygen consumption rate (OCR) were performed in cells at “resting conditions” (not stimulated by histamine or other calcium agonists). This would hold true only if EFHD1 loss-of-function would not affect basal mitochondrial calcium levels. Therefore, to address the Reviewer’s concern, we measured basal mitochondrial Ca²⁺ levels using the high-affinity calcium probe mt-GCaMP6 in EFHD1 knockdown and control HeLa cells, as well as in response to the knockdown of MICU1, which is known to increase mitochondrial Ca²⁺ levels in resting conditions. As shown in the figure, basal mitochondrial calcium levels remained unaffected by EFHD1 LOF. This result should also help explaining why we cannot detect differences in oxygen consumption rates between control and shEFHD1 cells in unstimulated cells. We have included these new data in the Supplementary Figure 4 (panel B).

4. In general, fonts and labels are difficult to see on laptops and when printed. I had to use a very large screen to clearly see the figures. More contrasting colors and larger fonts can help readers.

Response: We appreciate the Reviewer's suggestion on how to make our figure more highly readable. We have made efforts to address this issue by increasing the font size and using more contrasting colours when possible.

References

- Andersen JS, Wilkinson CJ, Mayor T, Mortensen P, Nigg EA & Mann M (2003) Proteomic characterization of the human centrosome by protein correlation profiling. *Nature* 426: 570–574
- Antonicka H, Lin ZY, Janer A, Aaltonen MJ, Weraarpachai W, Gingras AC & Shoubridge EA (2020) A High-Density Human Mitochondrial Proximity Interaction Network. *Cell Metab* 32: 479-497.e9
- Austin S, Mekis R, Mohammed SEM, Scalise M, Wang W-A, Galluccio M, Pfeiffer C, Borovec T, Parapatics K, Vitko D, *et al* (2022) TMBIM5 is the Ca²⁺/H⁺ antiporter of mammalian mitochondria. *EMBO Rep* 23: e54978
- Bahri H, Buratto J, Rojo M, Dompierre JP, Salin B, Blancard C, Cuvellier S, Rose M, Ben Ammar Elgaaied A, Tetaud E, *et al* (2021) TMEM70 forms oligomeric scaffolds within mitochondrial cristae promoting in situ assembly of mammalian ATP synthase proton channel. *Biochim Biophys Acta Mol Cell Res* 1868: 118942
- Boehmerle W, Splittgerber U, Lazarus MB, McKenzie KM, Johnston DG, Austin DJ & Ehrlich BE (2006) Paclitaxel induces calcium oscillations via an inositol 1,4,5-trisphosphate receptor and neuronal calcium sensor 1-dependent mechanism. *Proc Natl Acad Sci U S A* 103: 18356–18361
- Gherardi G, Nogara L, Ciciliot S, Fadini GP, Blaauw B, Braghetta P, Bonaldo P, De Stefani D, Rizzuto R & Mammucari C (2018) Loss of mitochondrial calcium uniporter rewires skeletal muscle metabolism and substrate preference. *Cell Death Differ* 2018 262 26: 362–381
- Havugimana PC, Hart GT, Nepusz T, Yang H, Turinsky AL, Li Z, Wang PI, Boutz DR, Fong V, Phanse S, *et al* (2012) A census of human soluble protein complexes. *Cell* 150: 1068–1081
- Hein MY, Hubner NC, Poser I, Cox J, Nagaraj N, Toyoda Y, Gak IA, Weisswange I, Mansfeld J, Buchholz F, *et al* (2015) A Human Interactome in Three Quantitative Dimensions Organized by Stoichiometries and Abundances. *Cell* 163: 712–723
- Huang G & Docampo R (2020) The mitochondrial calcium uniporter interacts with subunit c of the ATP synthase of trypanosomes and humans. *MBio* 11
- Huo J, Prasad V, Grimes KM, Vanhoutte D, Blair NS, Lin S-C, Broun MJ, Bers DM & Molkentin JD (2023) MCUB is an inducible regulator of calcium-dependent mitochondrial metabolism and substrate utilization in muscle. *Cell Rep* 42: 113465
- Kidd JF, Pilkington MF, Schell MJ, Fogarty KE, Skepper JN, Taylor CW & Thorn P (2002) Paclitaxel affects cytosolic calcium signals by opening the mitochondrial permeability transition pore. *J Biol Chem* 277: 6504–6510

- Kovalčíkova J, Vrbacký M, Pecina P, Tauchmannová K, Nůsková H, Kaplanová V, Brázdová A, Alán L, Eliáš J, Čunátová K, *et al* (2019) TMEM70 facilitates biogenesis of mammalian ATP synthase by promoting subunit c incorporation into the rotor structure of the enzyme. *FASEB J* 33: 14103–14117
- Kristensen AR, Gsponer J & Foster LJ (2012) A high-throughput approach for measuring temporal changes in the interactome. *Nat Methods* 9: 907–909
- Kwong JQ, Huo J, Brourd MJ, Boyer JG, Schwanekamp JA, Ghazal N, Maxwell JT, Jang YC, Khuchua Z, Shi K, *et al* (2018) The mitochondrial calcium uniporter underlies metabolic fuel preference in skeletal muscle. *JCI Insight* 3
- Meng K, Hu Y, Wang D, Li Y, Shi F, Lu J, Wang Y, Cao Y, Chris |, Zhang Z, *et al* (2023) EFHD1, a novel mitochondrial regulator of tumor metastasis in clear cell renal cell carcinoma. *Cancer Sci* 00: 1–12
- Michaelis AC, Brunner AD, Zwiebel M, Meier F, Strauss MT, Bludau I & Mann M (2023) The social and structural architecture of the yeast protein interactome. *Nature* 624: 192–200
- Sancak Y, Markhard AL, Kitami T, Kovács-Bogdán E, Kamer KJ, Udeshi ND, Carr SA, Chaudhuri D, Clapham DE, Li AA, *et al* (2013) EMRE is an essential component of the mitochondrial calcium uniporter complex. *Science* 342: 1379–82
- Tang J, Zhang K, Dong J, Yan C, Hu C, Ji H, Chen L, Chen S, Zhao H & Song Z (2020) Sam50–Mic19–Mic60 axis determines mitochondrial cristae architecture by mediating mitochondrial outer and inner membrane contact. *Cell Death Differ* 27: 146–160
- Tomar D, Thomas M, Garbincius JF, Kolmetzky DW, Salik O, Jadiya P, Joseph SK, Carpenter AC, Hajnóczky G & Elrod JW (2023) MICU1 regulates mitochondrial cristae structure and function independently of the mitochondrial Ca²⁺ uniporter channel. *Sci Signal* 16: eabi8948

Dear Fabiana,

We have now received re-review reports from two referees, which I have included below. As you will see, you have addressed their concerns satisfactorily. In the interests of time, I have decided to proceed without additional input from referee #1. Before I can finally accept the manuscript, there are some remaining editorial points which need to be addressed. In this regard would you please:

- ensure funding from the German Research Foundation (DFG) under the Emmy Noether Programme (PE 2053/1-1) and the Bavarian Ministry of Sciences, Research and the Arts in the framework of the Bavarian Molecular Biosystems Research Network (D2-F5121.2-10c/4822) is acknowledged in our online submission system,
- rename the 'Conflict of Interest Statement' the 'Disclosure and competing interests statement',
- remove the author credit section from the manuscript text,
- rename Supplementary tables S1-S5 as Dataset EV1-EV5 with the corresponding callouts (legends are correct; nomenclature should be Dataset EV1-EV5),
- state in the legend of Appendix Fig S1 that blots have been re-used from Figure 2A; the Fig S1 legend should be updated to reference not just the quantification but also the blots; consider referring to quantification in Fig. 2A,
- provide exact p values in the legends of figures 2a, c; 4e, h-l; 5a, f, h-i, supplementary figures 2i; 3a, d-e; 4b; 5c-d, indicate the statistical test used for data analysis in the legends of figures 3d; 5g; 6c, f,
- in figures 4h-i, correct the mismatch between the annotated p values in the figure legend and the annotated p values in the figure file,
- define box plots in terms of minima, maxima, centre, bounds of box and whiskers, and percentile in the legends of figures 2c, supplementary figures 6b-c,
- define n in the legends of supplementary figures 6b-c,
- describe the nature of entity for 'n' in the legends of figures 2a; 4c, e, h-l; 5a-c, f, h-i, supplementary figures 2i, k-l; 3a-c, e; 5c,
- change the nomenclature for Supplemental Figures 1-6 to Figure EV1-EV6 with the appropriate callouts,
- remove Supplemental table legends from ms file, and
- provide a valid email address for author Maria Patron; emails to maria.patron@age.mpg.de bounced.

We include a synopsis of the paper (see <http://emboj.embopress.org/>). Please provide me with a general summary image, a two sentence summary statement and 3-5 bullet points that capture the key findings of the paper.

I am looking forward to receiving your revised manuscript.

EMBO Press is an editorially independent publishing platform for the development of EMBO scientific publications.

Best wishes,

William

William Teale, PhD
Editor
The EMBO Journal
w.teale@embojournal.org

- a point-by-point response to the referees' comments, with a detailed description of the changes made (as a word file).
- a word file of the manuscript text.
- individual production quality figure files (one file per figure)
- a complete author checklist, which you can download from our author guidelines (<https://www.embopress.org/page/journal/14602075/authorguide>).

- Expanded View files (replacing Supplementary Information)

We realize that it is difficult to revise to a specific deadline. In the interest of protecting the conceptual advance provided by the work, we recommend a revision within 3 months (23rd Oct 2024). Please discuss the revision progress ahead of this time with the editor if you require more time to complete the revisions. Use the link below to submit your revision:

Referee #2:

The authors have responded well and satisfactorily to my comment.

Referee #3:

The authors addressed all my concerns and I support publication of the manuscript in EMBO at its current form.

Point-by-point response

- ensure funding from the German Research Foundation (DFG) under the Emmy Noether Programme (PE 2053/1-1) and the Bavarian Ministry of Sciences, Research and the Arts in the framework of the Bavarian Molecular Biosystems Research Network (D2-F5121.2-10c/4822) is acknowledged in our online submission system,

We have updated the acknowledgment in the online submission system to include the specified funding sources.

- rename the 'Conflict of Interest Statement' the 'Disclosure and competing interests statement',

We have renamed the section as requested.

- remove the author credit section from the manuscript text,

The author credit section has been removed from the manuscript text.

- rename Supplementary tables S1-S5 as Dataset EV1-EV5 with the corresponding callouts (legends are correct; nomenclature should be Dataset EV1-EV5),

We have edited the names of the supplementary tables and the corresponding callouts to Dataset EV1-EV5.

- state in the legend of Appendix Fig S1 that blots have been re-used from Figure 2A; the Fig S1 legend should be updated to reference not just the quantification but also the blots; consider referring to quantification in Fig. 2A,

We have updated the legend of Appendix Fig S1 as follows:

“Immunoblot analysis of MCU, EMRE, MCUB, MICU1, MICU2 and ACTIN (loading control) in whole cell lysates from Flp-In T-REx HEK293 cell lines before (-Tet) and after (+Tet) tetracycline-driven expression of each bait. Immunoblots of MCU from MCU-bait cells, EMRE from EMRE-bait cells and MCUB from MCUB-bait cells were re-used from Figure 2A. Refer to quantification in **Figure 2A**.”

- provide exact p values in the legends of figures 2a, c; 4e, h-l; 5a, f, h-i, supplementary figures 2i; 3a, d-e; 4b; 5c-d,

We have added the exact p-values to the panels of the specified figures.

- indicate the statistical test used for data analysis in the legends of figures 3d; 5g; 6c, f,

We have indicated the statistical tests used for data analysis in the legends of the specified figures.

- in figures 4h-i, correct the mismatch between the annotated p values in the figure legend and the annotated p values in the figure file,

We have corrected the mismatch between the annotated p-values in the figure panel.

- define box plots in terms of minima, maxima, centre, bounds of box and whiskers, and percentile in the legends of figures 2c, supplementary figures 6b-c,

We have defined the center, bounds of the box, whiskers, and percentiles in the legends of the specified figures.

- define n in the legends of supplementary figures 6b-c,

We have defined the sample size (n) in the legends of the specified figures.

- describe the nature of entity for 'n' in the legends of figures 2a; 4c, e, h-l; 5a-c, f, h-i, supplementary figures 2i, k-l; 3a-c, e; 5c,

We have described the nature of the entity for 'n' in the legends of the specified figures.

- change the nomenclature for Supplemental Figures 1-6 to Figure EV1-EV6 with the appropriate callouts,

We have aligned the nomenclature of the supplementary figures as requested.

- remove Supplemental table legends from ms file, and

We have removed the supplementary table legends from the manuscript file.

- provide a valid email address for author Maria Patron; emails to maria.patron@age.mpg.de bounced.

We have provided a valid email address for Maria Patron in the submission system.

We include a synopsis of the paper (see <http://emboj.embopress.org/>). Please provide me with a general summary image, a two sentence summary statement and 3-5 bullet points that capture the key findings of the paper.

We have prepared and submitted the general summary image, the two-sentence summary statement, and 3-5 bullet points capturing the key findings of the paper as requested.

Dear Fabiana,

I am pleased to inform you that your manuscript has been accepted for publication in the EMBO Journal.

Congratulations! I'm sure this article will prove very useful to many labs.

Best wishes,

William

William Teale, PhD
Editor
The EMBO Journal
w.teale@embojournal.org
